# Learning low-rank latent mesoscale structures in networks

Hanbaek Lyu [1] ✉, Yacoub H. Kureh[2], Joshua Vendrow[3] & Mason A. Porter [2,4,5]

Researchers in many fields use networks to represent interactions between entities in complex systems. To study the large-scale behavior of complex systems, it is useful to examine mesoscale structures in networks as building blocks that influence such behavior. In this paper, we present an approach to describe low-rank mesoscale structures in networks. We find that many real-world networks possess a small set of latent motifs that effectively approximate most subgraphs at a fixed mesoscale. Such low-rank mesoscale structures allow one to reconstruct networks by approximating subgraphs of a network using combinations of latent motifs. Employing subgraph sampling and nonnegative matrix factorization enables the discovery of these latent motifs. The ability to encode and reconstruct networks using a small set of latent motifs has many applications in network analysis, including network comparison, network denoising, and edge inference.

It is often insightful to examine structures in networks[1] at intermediate scales (i.e., mesoscales) that lie between the microscale of nodes and edges and the macroscale distributions of local network properties. Researchers have considered subgraph patterns (i.e., the connection patterns of subsets of nodes) as building blocks of network structure at various mesoscales[2]. In many studies of networks, researchers identify *motifs* as $k$-node subgraph patterns (where $k$ is typically between 3 and 5) of a network that are unexpectedly more common in that network than in some random-graph null model[3]. In the past two decades, the study of motifs has been important for the analysis of networked systems in many areas, including biology[4–8], sociology[9,10], and economics[11,12]. However, to the best of our knowledge, researchers have not examined how to use such motifs (or related mesoscale structures), after their discovery, as building blocks to reconstruct a network. In the present paper, we provide this missing computational framework to bridge inferred subgraph-based mesoscale structures and the global structure of networks. To do this, we propose (1) an algorithm for network dictionary learning (NDL) that learns *latent motifs* from samples of certain random $k$-node subgraphs and (2) a complementary algorithm for network denoising and reconstruction (NDR) that approximates a given network as well as possible using the

learned latent motifs. We also provide a rigorous theoretical analysis of the proposed algorithms. This analysis includes a proof that one can accurately reconstruct a network if one has a dictionary of latent motifs that can accurately approximate mesoscale structures of the network. We compare our approach to related prior work[13] in the Methods section and in our Supplementary Information (SI).

Using our approach, we find that various real-world networks (such as Facebook friendship networks, coronavirus and *Homo sapiens* protein–protein interaction (PPI) networks, and an arXiv collaboration network) have low-rank subgraph patterns, in the sense that one can successfully approximate their $k$-node subgraph patterns by a weighted sum of a small number of latent motifs. The latent motifs of these networks thereby reveal low-rank mesoscale structures of these networks. Our claim of the low-rank nature of such mesoscale structures concerns the space of certain subgraph patterns, rather than the embedding of an entire network into a low-dimensional Euclidean space (as considered in spectral-embedding and graph-embedding methods[14,15]). One cannot obtain such a low-dimensional graph embedding for networks with small mean degrees and large clustering coefficients[16]. Additionally, as we demonstrate in this paper, the ability to encode a network using a set of latent motifs has a wide variety of

[1]Department of Mathematics, University of Wisconsin-Madison, Madison, WI 53706, USA. [2]Department of Mathematics, University of California, Los Angeles, CA 90095, USA. [3]Department of Electrical Engineering and Computer Science, Massachusetts Institute of Technology, Cambridge, MA 02139, USA. [4]Department of Sociology, University of California, Los Angeles, CA 90095, USA. [5]Santa Fe Institute, Sante FE, NM 87501, USA. ✉e-mail: hlyu@math.wisc.edu

applications in network analysis. These applications include network comparison, network denoising, and edge inference.

## Motivating application: Anomalous-subgraph detection

A common problem in network analysis is the detection of anomalous subgraphs of a network (see Fig. 1)[17]. The connection pattern of an anomalous subgraph distinguishes it from the rest of a network. This anomalous-subgraph-detection problem has numerous high-impact applications, including in security, finance, healthcare, and law enforcement[18,19]. Various approaches, including both classical techniques[17] and modern deep-neural-network techniques[20], have been proposed to detect anomalous subgraphs.

Consider the following simple conceptual framework for anomalous-subgraph detection: We learn normal subgraph patterns in an observed network and then seek to detect subgraphs in the observed network that deviate significantly from them.

By studying low-rank mesoscale structures in networks, we can turn this high-level idea for anomalous-subgraph detection into a concrete approach, which we now briefly summarize. First, we compute latent motifs (see Fig. 1d) of an observed network (see Fig. 1a) that can successfully approximate the $k$-node subgraphs of the observed network. A key observation is that these subgraphs should also describe the normal subgraph patterns of the observed network (see Fig. 1b). The rationale that underlies this observation is that the $k$-node subgraphs of the observed network likely form a low-rank space, so we expect the latent motifs to be robust with respect to the addition of anomalous edges (see Fig. 1c). Consequently, reconstructing the observed network using its latent motifs yields a weighted network (see Fig. 1e) in which edges with positive and small weights deviate significantly from the normal subgraph patterns, which are captured by the latent motifs. Therefore, such edges are likely to be anomalous. The suspicious edges (see Fig. 1f) are the edges in the weighted reconstruction that have positive weights that are less than a threshold. One can determine the threshold using a small set of known true edges and known anomalous edges. The suspicious edges match well with the anomalous edges in Fig. 1c. See the SI for more details.

In the remainder of our paper, we carefully develop the three key components of our approach: (1) effective sampling of $k$-node subgraphs; (2) reconstructing observed networks using candidate latent motifs; and (3) computing latent motifs from observed networks. The key idea of our work is to approximate sampled subgraphs by latent motifs and then combine these approximations to construct a weighted reconstructed network. We also present a variety of supporting numerical experiments using several synthetic and real-world networks.

## Results

In this section, we describe our key ideas, key methods, and results of several numerical experiments.

### $k$-path motif sampling and latent motifs

Computing all $k$-node subgraphs of a network is computationally expensive and is the main computational bottleneck of traditional motif analysis[3]. Our approach, which bypasses this issue, is to learn latent motifs by drawing random samples of a particular class of $k$-node connected subgraphs. We consider random $k$-node subgraphs that we obtain by uniformly randomly sampling a $k$-path from a network and including all edges between the sampled nodes of the network. A sequence $\mathbf{x} = (x_1, ..., x_k)$ of $k$ (not necessarily distinct) nodes is a $k$-walk if $x_i$ and $x_{i+1}$ are adjacent for all $i \in \{1, ..., k-1\}$. A $k$-walk is a $k$-path if all nodes in the walk are distinct (see Fig. 2). Sampling a $k$-path serves two purposes: (1) it ensures that the sampled $k$-node induced subgraph is connected with the minimum number of imposed edges; and (2) it induces a natural node ordering of the $k$-node induced subgraph. (Such an ordering is important for computations that involve subgraphs.) By using a $k$-walk motif-sampling algorithm of Lyu et al.[21] in conjunction with rejection sampling, one can sample a large number of $k$-paths and obtain their associated induced subgraphs.

The $k$-node subgraphs that are induced by uniformly randomly sampling $k$-paths from a network are the mesoscale structures that we consider in the present paper. We use the term *on-chain edges* for the edges of these subgraphs between nodes $x_i$ and $x_{i+1}$ for $i \in \{1, ..., k-1\}$, and we use the term *off-chain edges* for all other edges. It is the off-chain edges that can differ across subgraphs and hence encode meaningful information about a network. For $k = 2$, the subgraphs are all isomorphic to a 2-path and hence have no off-chain edges. For $k = 3$, the subgraphs can have a single off-chain edge, so they are isomorphic either to a 2-path or to a 3-clique (i.e., a graph with three nodes and all three possible edges between them). For larger values of $k$, the subgraphs can have diverse connection patterns (see Fig. 2), depending on the architecture of the original network.

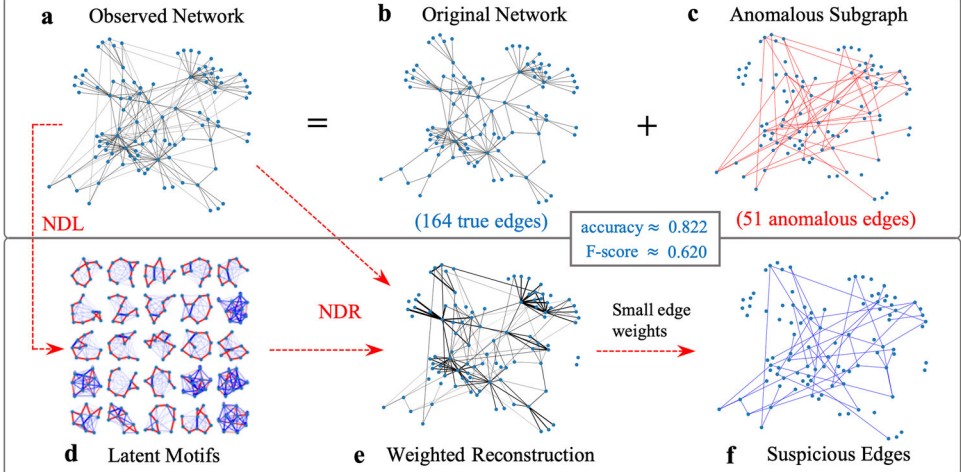

**Fig. 1 | Illustration of anomalous-subgraph detection using network reconstruction.** The (**a**) observed network consists of (**b**) the original network and (**c**) anomalous edges, and we seek to detect the anomalous edges in the observed network. In our approach, we first (**d**) determine a set of latent motifs and then (**e**) use them to reconstruct the observed network. In the (**f**) weighted reconstruction of the network, we identify the edges with positive but small weights as suspicious edges. We compute the accuracy and the F-score for inferring the anomalous edges in (**c**) as the suspicious edges in (**f**), where the F-score is the harmonic mean of the precision and recall scores.

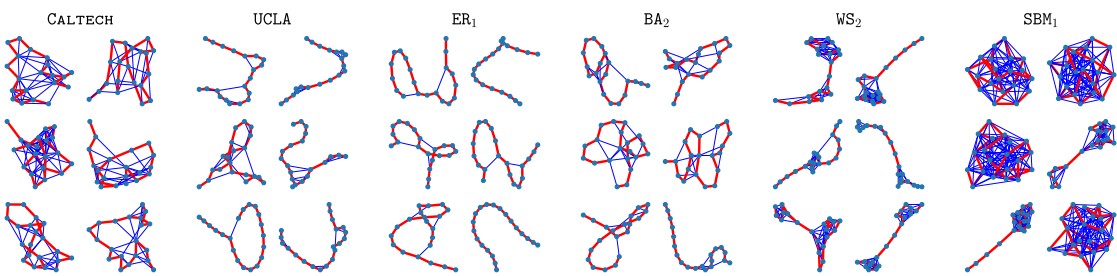

**Fig. 2 | Examples of 20-node subgraphs that are induced by uniformly sampled 20-paths from various networks.** In each subgraph, the sampled 20-path consists of the red edges. The networks are CALTECH and UCLA Facebook networks, an Erdős–Rényi (ER) random graph (ER$_1$), a Barabási–Albert (BA) random graph (BA$_2$), a Watts–Strogatz (WS) small-world network (WS$_2$), and a stochastic-block-model (SBM) network (SBM$_1$). See the Methods section for more details about these networks.

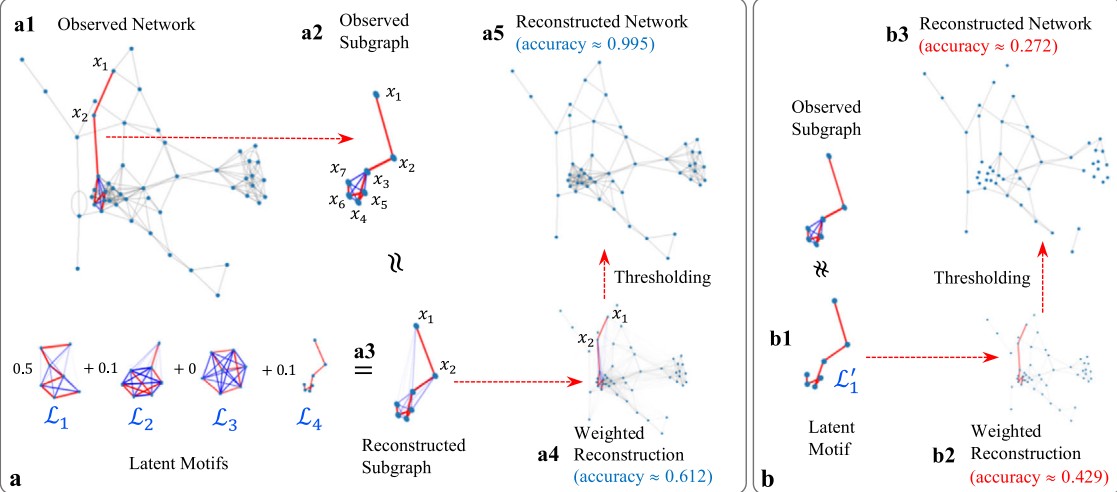

**Fig. 3 | An illustration of our low-rank network-reconstruction process using latent motifs.** Given (**a1**) an observed network and a set of latent motifs $\mathcal{L}_1, \ldots, \mathcal{L}_r$, we repeatedly sample a $k$-path and approximate (**a2**) the induced subgraph of the nodes in that path by (**a3**) a nonnegative linear combination of the latent motifs $\mathcal{L}_1, \ldots, \mathcal{L}_r$. We then compute the weighted reconstruction in (**a4**) by taking the edge weight between each unordered node pair $\{x, y\}$ to be the mean of the reconstructed weights of $\{x, y\}$ from all sampled subgraphs that include both $x$ and $y$. We measure the accuracy of the reconstruction of the weighted network by calculating the Jaccard index (i.e., 1 minus the Jaccard distance) in (31) in the SI. We then obtain (**a5**) an unweighted (i.e., binary) reconstructed network by thresholding the edge weights in (**a4**) with a threshold 0.5. That is, we retain edges whose weights are at least 0.5 and we remove all other edges. We measure the accuracy of the reconstruction of the binary network by calculating the Jaccard index between the original network's edge set and the associated reconstructed network's edge set. The same network-reconstruction process using (**b1**) a single latent motif $\mathcal{L}_1'$, which is the $k$-path, yields (**b2**) weighted and (**b3**) binary reconstructions with lower accuracies than those in (**a4**) and (**a5**), respectively.

We study the connection patterns of a random $k$-node subgraph by decomposing it as a weighted sum of more elementary subgraph patterns (possibly with continuous-valued edge weights), which we call *latent motifs* (see Fig. 3a1–a3). To study mesoscale structures in networks, we investigate several questions. How many distinct latent motifs does one need to successfully approximate all of these $k$-node subgraph patterns? What do they look like? How do these latent motifs differ for different networks?

**Low-rank network reconstruction using latent motifs**

We illustrate our procedure to reconstruct observed networks using latent motifs in Fig. 3. Suppose that we have a network $G = (V, E)$ and two collections, $W = \{\mathcal{L}_1, \ldots, \mathcal{L}_r\}$ and $W' = \{\mathcal{L}_1', \ldots, \mathcal{L}_r'\}$ of latent motifs. We refer to such collections as *network dictionaries*. How can one determine which of the network dictionaries better describes the mesoscale structure of $G$? One can sample a large number of $k$-node subgraphs $A_i$ of $G$ and, for each $A_i$, independently, determine the nonnegative linear combination of latent motifs that yields the closest approximation $\hat{A}_i$. By comparing the subgraphs $A_i$ with their corresponding approximations $\hat{A}_i$, one can demonstrate how well the latent motifs in the network

dictionary $W$ approximate the $k$-node subgraph patterns of $G$ (see Fig. 3a1–a3).

For applications such as anomalous-subgraph detection, it is helpful to construct a weighted network $G_{\text{recons}}$ with the same node set $V$ that approximates $G$ as well as possible using the network dictionary $W$. The network $G_{\text{recons}}$ is a *rank-$r$ mesoscale reconstruction* of $G$. If $G_{\text{recons}}$ is close to $G$, we conclude that the latent motifs in $W$ successfully capture the structure of $k$-node subgraphs of $G$ and that $G$ has rank-$r$ subgraph patterns that are prescribed by the latent motifs in $W$. We interpret the edge weights in $G_{\text{recons}}$ as measures of confidence in the corresponding edges of $G$ with respect to $W$. For example, we interpret the edge $e$ with the smallest weight in $G_{\text{recons}}$ as the most outlying edge with respect to the latent motifs in $W$ (see Fig. 1e, f). We can threshold the weighted edges of $G_{\text{recons}}$ at some fixed value $\theta \in [0, 1]$ to obtain an undirected reconstructed network $G_{\text{recons}}(\theta)$ with binary edge weights (which are either 0 or 1). We can then directly compare $G_{\text{recons}}(\theta)$ to the original unweighted network $G$.

Our *network denoising and reconstruction* (NDR) algorithm (see Algorithm NDR in the SI) works as follows. We seek to build a weighted network $G_{\text{recons}}$ using the node set $V$ and a weighted adjacency matrix

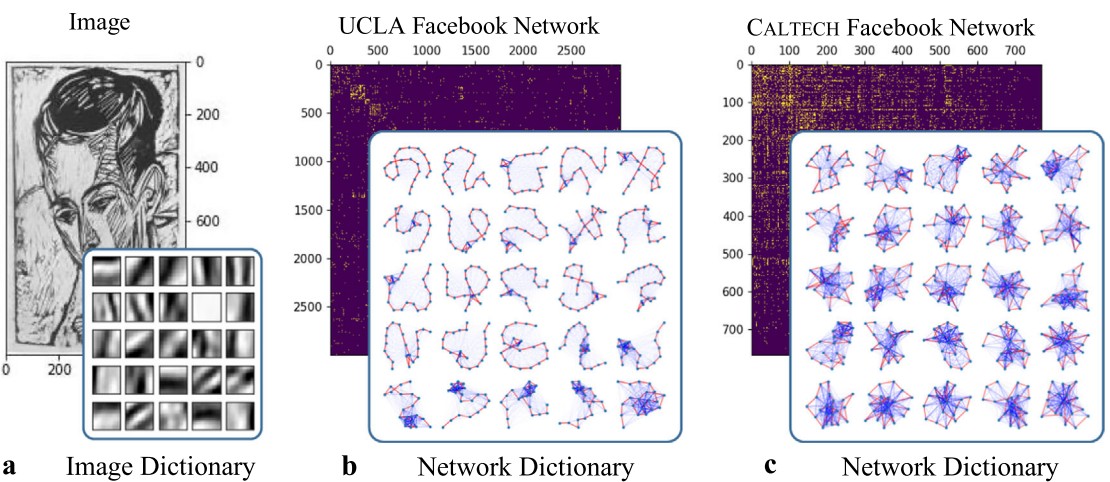

**a**   Image Dictionary          **b**   Network Dictionary          **c**   Network Dictionary

**Fig. 4 | Illustration of mesoscale structures that we learn from images and networks.** In each experiment in this figure, we form a matrix $X$ of size $d \times n$ by sampling $n$ mesoscale patches of size $d = 21 \times 21$ from the corresponding object. For the image in (**a**), the columns of $X$ are square patches with $21 \times 21$ pixels. In (**b**, **c**), we show portions of the adjacency matrices of the two networks. We take the columns of $X$ to be the $k \times k$ adjacency matrices of the connected subgraphs that are induced by a path of $k = 21$ nodes, where a $k$-node path consists of $k$ distinct nodes $x_1, \ldots, x_k$ such that $x_i$ and $x_{i+1}$ are adjacent for all $i \in \{1, \ldots, k-1\}$. Using nonnegative matrix factorization (NMF), we compute an approximate factorization $X \approx WH$ into nonnegative matrices $W$ and $H$, where $W$ is called a *network dictionary* and has $r = 25$ columns. Because of this factorization, we can approximate any sampled mesoscale patches (i.e., the columns of $X$) of an object by a nonnegative linear combination of the columns of $W$, which we interpret as latent shapes for the image and as latent motifs (i.e., subgraphs) for the networks. The columns of $H$ give the coefficients in these linear combinations. The network dictionaries of latent motifs that we learn from the (**b**) UCLA and (**c**) CALTECH Facebook networks have distinctive social structures. In the adjacency matrix of the UCLA network, we show only the first 3000 nodes (according to the node labeling in the data set). The image in (**a**) is from the collection `Die Graphik Ernst Ludwig Kirchners bis 1924, von Gustav Schiefler Band I bis 1916` (Accession Number 2007.141.9, Ernst Ludwig Kirchner, 1926). We use the image with permission from the National Gallery of Art in Washington, DC, USA.

$A_{\text{recons}} : V^2 \rightarrow \mathbb{R}$. This network best approximates the observed network $G$, given that the subgraphs of $G_{\text{recons}}$ are generated by the latent motifs in $W$. First, we uniformly randomly sample a large number $T$ of $k$-paths $\mathbf{x}_1, \ldots, \mathbf{x}_T \colon \{1, \ldots, k\} \rightarrow V$ in $G$. We then determine the $k \times k$ unweighted matrices $A_{\mathbf{x}_1}, \ldots, A_{\mathbf{x}_T}$ with entries $A_{\mathbf{x}_t}(i,j) = A(\mathbf{x}_t(i), \mathbf{x}_t(j))$, which equals 1 if nodes $\mathbf{x}_t(i)$ and $\mathbf{x}_t(j)$ are adjacent in the network and equals 0 otherwise. These are the adjacency matrices of the induced subgraphs of the nodes of the $k$-paths that we sample (see Fig. 3a2). We then approximate each $A_{\mathbf{x}_t}$ by a nonnegative linear combination $\hat{A}_{\mathbf{x}_t}$ of the latent motifs in $W$ (see Fig. 3a3). We then compute $A_{\text{recons}}(x, y)$ for each $x, y \in V$ as the mean of $\hat{A}_{\mathbf{x}_t}(a,b)$ for all $t \in \{1, \ldots, T\}$ and all $a, b \in \{1, \ldots, k\}$ such that $\mathbf{x}_t(a) = x$ and $\mathbf{x}_t(b) = y$ (see Fig. 3a4). We provide theoretical guarantees and error bounds for our NDR algorithm in the SI (see Algorithm NDR).

Consider reconstructing a network $G$ using a single latent motif $\mathcal{L}'_1$ that is a $k$-path. We begin with the case $k = 2$, such that each subgraph that we sample is a 2-path. The sampled 2-paths are approximated perfectly by $\mathcal{L}'_1$ (see Fig. 3b1). A 2-path that one chooses uniformly at random has an equal probability of sampling each edge of $G$, so $G_{\text{recons}} = G$. Therefore, we conclude that, at scale $k = 2$, one can perfectly reconstruct $G$ by using the 2-path latent motif $\mathcal{L}'_1$. However, for $k \geq 3$, the graph $G_{\text{recons}}$ can differ significantly from $G$, as approximating the observed subgraphs by a single $k$-path misses all of the off-chain edges (see Fig. 3b1–b3). To properly describe the $k$-node subgraph patterns of $G$, one may need more than one latent motif with off-chain edges (see Fig. 3a3). We give more details in Appendix D of the SI.

**Dictionary learning and latent motifs**

How does one compute latent motifs from a given network? *Dictionary-learning* algorithms are machine-learning techniques that learn interpretable latent structures of complex data sets. They are employed regularly in the data analysis of text and images[22–24]. Dictionary-learning algorithms usually consist of two steps. First, one samples a large number of structured subsets of a data set (e.g., square patches of an image or collections of a few sentences of a text); we refer to such a subset as a *mesoscale patch* of a data set. Second, one finds a set of basis elements such that taking a nonnegative linear combination of them can successfully approximate each of the sampled mesoscale patches. Such a set of basis elements is called a *dictionary*, and one can interpret each basis element as a latent structure of the data set.

As an example, consider the artwork image in Fig. 4a. We first uniformly randomly sample 10000 square patches with $21 \times 21$ pixels and vectorize them to obtain a $21^2 \times 10000$ matrix $X$. The choice of vectorization $\mathbb{R}^{k \times k} \rightarrow \mathbb{R}^{k^2}$ is arbitrary; we use the column-wise vectorization in Algorithm A4 in the SI. We then use a nonnegative matrix factorization (NMF)[25] algorithm to find an approximate factorization $X \approx WH$, where $W$ and $H$ are nonnegative matrices of sizes $21^2 \times 25$ and $25 \times 10{,}000$, respectively. Reshaping the columns of $W$ into image patches of size $21 \times 21$ yields an image dictionary that describes latent shapes of the image.

Our *network dictionary learning* (NDL) algorithm to compute a network dictionary of latent motifs is based on a similar idea. As mesoscale patches of a network, we use the $k \times k$ binary (i.e., unweighted) matrices that encode connection patterns between the nodes that form a uniformly random $k$-path. After obtaining sufficiently many mesoscale patches of a network (e.g., by using a motif-sampling algorithm[21] with rejection sampling), we apply a dictionary-learning algorithm (e.g., NMF[25]) to obtain latent motifs of the network. A latent motif is a $k$-node weighted network with nodes $\{1, \ldots, k\}$ and edges that have weights between 0 and 1. As with subgraphs more generally, we use the term *on-chain edges* for the edges of a latent motif between nodes $i$ and $i+1$ for $i \in \{1, \ldots, k-1\}$; we use the term *off-chain edges* for all other edges. We give more background about our NDL algorithm in the Methods section and provide a complete implementation of our approach in Algorithm NDL in the SI. We give theoretical guarantees for Algorithm NDL in Theorems F.4 and F.7 in the SI.

In Fig. 4, we compare 25 latent motifs with $k = 21$ nodes of Facebook friendship networks (which were collected on one day in fall

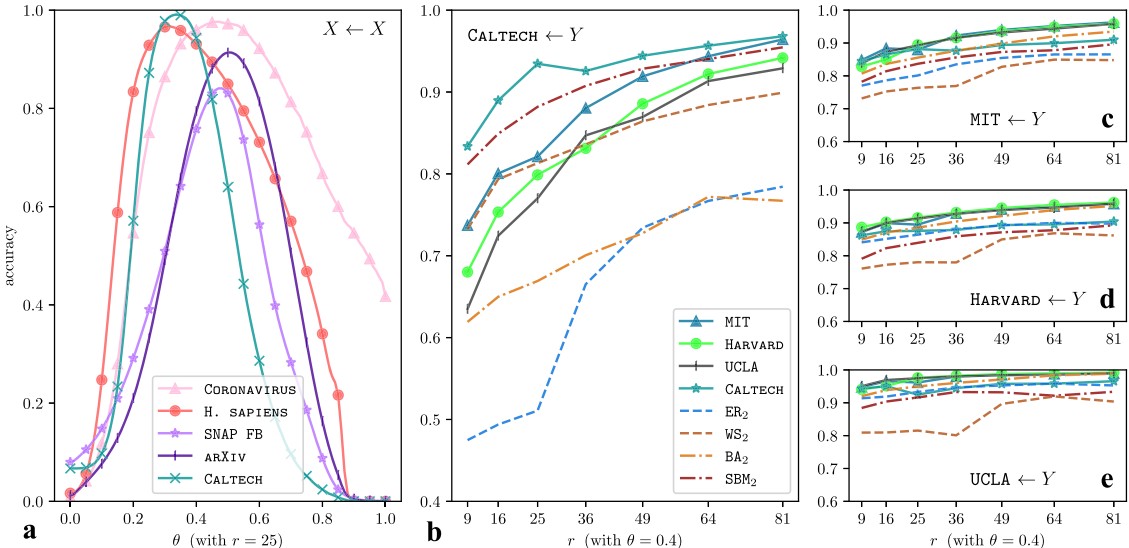

**Fig. 5 | Network-reconstruction experiments.** We show the self-reconstruction and cross-reconstruction accuracies of several real-world and synthetic networks versus the edge threshold $\theta$ and the number $r$ of latent motifs in a network dictionary. The label $X \leftarrow Y$ indicates that we reconstruct network $X$ using a network dictionary that we learn from network $Y$. The reconstruction process produces a weighted network that we turn into an unweighted network by thresholding the edge weights at a threshold value $\theta$; we keep only edges whose weights are strictly larger than $\theta$. We measure reconstruction accuracy by calculating the Jaccard index between an original network's edge set and an associated reconstructed network's edge set. In (**a**), we plot accuracies versus $\theta$ (with the number of latent motifs fixed at $r = 25$), where $X$ is one of five real-world networks (two PPI networks, two Facebook networks, and one collaboration network). In (**b**–**e**), we reconstruct each of the four Facebook networks using network dictionaries with $r \in \{9, 16, 25, 36, 64, 81, 100\}$ latent motifs that we learn from one of eight networks (with the threshold value fixed at $\theta = 0.4$).

2005) from UCLA and Caltech[26,27]. We denote the UCLA network by UCLA and the Caltech network by CALTECH. Each node of one of these networks is a Facebook account of an individual, and each edge encodes a Facebook friendship between two individuals. The latent motifs reveal striking differences between these networks in the connection patterns of the subgraphs that are induced by $k$-paths with $k = 21$. For example, the latent motifs in UCLA's dictionary (see Fig. 4b) have sparse off-chain connections with a few clusters, whereas CALTECH's dictionary (see Fig. 4c) has relatively dense off-chain connections. Most of CALTECH's latent motifs have hub nodes (which are adjacent to many other nodes in the latent motif) or communities[28,29] with six or more nodes. We give community-size statistics in Supplementary Fig. 2. An important property of $k$-node latent motifs is that any network structure (e.g., hub nodes, communities, and so on) in the latent motifs must also exist in actual $k$-node subgraphs. We observe both hubs and communities in the subgraphs of CALTECH in Fig. 2. By contrast, most of UCLA's latent motifs do not have such structures, as is also the case for the subgraphs of UCLA in Fig. 2.

Because $k$-node latent motifs encode basic connection patterns of $k$ nodes that are at most $k - 1$ edges apart, one can interpret $k$ as a scale parameter. Latent motifs that one learns from the same network for different values of $k$ reveal different mesoscale structures. For more details, see Supplementary Fig. 3.

### Example networks

We demonstrate our approach using 16 example networks; 8 of them are real-world networks and 8 of them synthetic networks. The 8 real-world networks are CORONAVIRUS PPI (for which we use the shorthand CORONAVIRUS)[30–32] and HOMO SAPIENS PPI (for which we use the shorthand H. SAPIENS)[15,30]; Facebook networks from CALTECH, UCLA, HARVARD, and MIT[26,27]; SNAP FACEBOOK (for which we use the shorthand SNAP FB)[15,33]; and ARXIV ASTRO-PH (for which we use the shorthand ARXIV)[15,34]. The first network is a protein–protein interaction (PPI) network of proteins that are related to the coronaviruses that cause Coronavirus disease 2019 (COVID-19), Severe Acute Respiratory Syndrome (SARS), and Middle Eastern Respiratory Syndrome (MERS)[31].

The second network is a PPI network of proteins that are related to *Homo sapiens*[30]. The third network is a 2012 Facebook network that was collected from participants of a survey[33]. The fourth network is a collaboration network of coauthorships of papers that were posted in the astrophysics category of the arXiv preprint server. The last four real-world networks are 2005 Facebook networks from four universities from the FACEBOOK100 data set[27]. In each Facebook network, nodes represent user accounts and edges encode Facebook friendships between these accounts.

For the eight synthetic networks, we generate two instantiations each of Erdős–Rényi (ER) $G(N, p)$ networks[35], Watts–Strogatz (WS) networks[36], Barabási–Albert (BA) networks[37], and stochastic-block-model (SBM) networks[38]. These four random-graph models are well-studied and are common choices for testing network methods and models[1]. Each of the ER networks has 5000 nodes, and we independently connect each pair of nodes with probabilities $p = 0.01$ (in the network that we call $ER_1$) and $p = 0.02$ (in $ER_2$). For the WS networks, we use rewiring probabilities $p = 0.05$ (in $WS_1$) and $p = 0.1$ (in $WS_2$) and start from a 5000-node ring network in which each node is adjacent to its 50 nearest neighbors. For the BA networks, we use $m = 25$ (in $BA_1$) and $m = 50$ (in $BA_2$), where $m$ denotes the number of edges of each new node when it connects (via linear preferential attachment) to the existing network, which we grow from an initial network of $m$ isolated nodes (i.e., none of them are adjacent to any other node) until it has 5000 nodes. The SBM networks $SBM_1$ and $SBM_2$ have three planted 1000-node communities; two nodes in the $i_0$th and the $j_0$th communities are connected by an edge independently with probability 0.5 if $i_0 = j_0$ (i.e., if they are in the same community) and with probabilities 0.001 for $SBM_1$ and 0.1 for $SBM_2$ if $i_0 \neq j_0$ (i.e., if they are in different communities). See the Methods section for more details.

### Network-reconstruction experiments

An important observation is that one can reconstruct a given network using an arbitrary network dictionary, including ones that one learns from an entirely different network. Such a cross-reconstruction allows one to quantitatively compare the learned mesoscale structures of

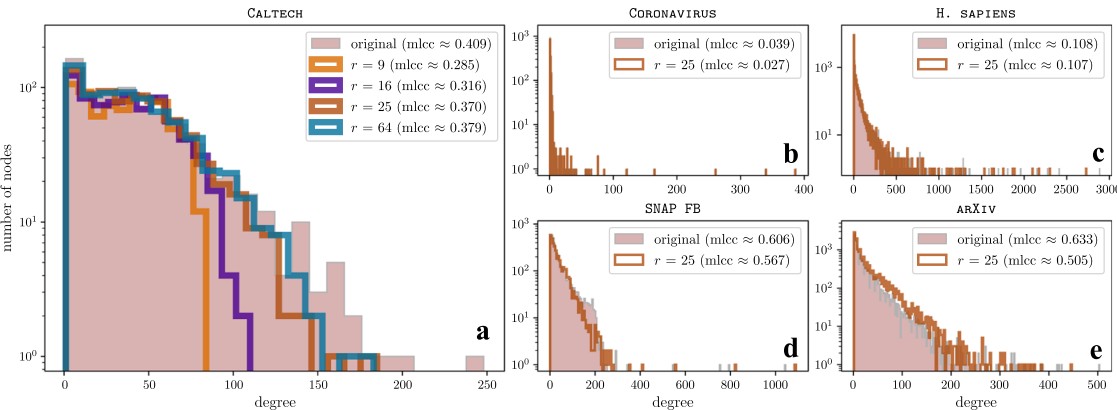

**Fig. 6 | Comparison of the degree distributions and the mean local clustering coefficients of the original and reconstructed networks.** We show the degree distributions as histograms, and we use the acronym mlcc in the legends to denote the mean local clustering coefficient. For network reconstruction, we use $r$ latent motifs at scale $k = 21$ for the five networks in Fig. 5a. In (**a**), we use the unweighted reconstructed networks for CALTECH with $r \in \{9, 16, 25, 64\}$ latent motifs that we used to compute the self-reconstruction accuracies in Fig. 5b. As we increase $r$, the

mean local clustering coefficient increases towards its value in the original network and the degree distributions of the reconstructed networks converge to that of the original network. By increasing $r$, we are able to include nodes with progressively larger degrees in the latent motifs. In (**b**–**e**), we show the mean local clustering coefficients and degree distributions of the unweighted reconstructed networks for CORONAVIRUS, H. SAPIENS, SNAP FB, and ARXIV with $r = 25$ latent motifs that we used to compute the self-reconstruction accuracies in Fig. 5b–e.

---

different networks. In Fig. 5, we show the results of several network-reconstruction experiments using a variety of real-world networks and synthetic networks. We label each subplot of Fig. 5 with $X \leftarrow Y$ to indicate that we are reconstructing network $X$ by approximating mesoscale patches of $X$ using a network dictionary that we learn from network $Y$. We perform these experiments for various values of the edge threshold $\theta \in [0, 1]$ and $r \in \{9, 16, 25, 36, 49, 81, 100\}$ latent motifs in a single dictionary. Each network dictionary in Fig. 5 has $k = 21$ nodes, for which the dimension of the space of all possible mesoscale patches (i.e., the adjacency matrices of the induced subgraphs) is $\binom{21}{2} - 20 = 190$. We measure the reconstruction accuracy by calculating the Jaccard index between the original network's edge set and the reconstructed network's edge set. That is, to measure the similarity of two edge sets, we calculate the number of edges in the intersection of these sets divided by the number of edges in the union of these sets. This gives a measure of reconstruction accuracy; if the Jaccard index equals 1, the reconstructed network is precisely the same as the original network. We obtain the same qualitative results as in Fig. 5 if we instead measure similarity using the Rand index[39].

In Fig. 5a, we plot the accuracy of the self-reconstruction $X \leftarrow X$ versus the threshold $\theta$ (with $r = 25$ latent motifs), where $X$ is one of the real-world networks CORONAVIRUS, H. SAPIENS, SNAP FB, CALTECH, and ARXIV. The accuracies for CORONAVIRUS, H. SAPIENS, and CALTECH peak above 95% when $\theta \approx 0.4$; the accuracies for ARXIV and SNAP FB peak above 88% and 70%, respectively, for $\theta \approx 0.6$. We choose $\theta = 0.4$ for the cross-reconstruction experiments for the Facebook networks CALTECH, HARVARD, MIT, and UCLA in Fig. 5b, c. These four Facebook networks have self-reconstruction accuracies above 80% for $r = 25$ motifs with the threshold $\theta = 0.4$. The total number of dimensions when using mesoscale patches at scale $k = 21$ is 190, so this result suggests that all eight of these real-world networks have low-rank mesoscale structures at scale $k = 21$.

We gain further insights into our self-reconstruction experiments by comparing the degree distributions and the mean local clustering coefficients of the original and the unweighted reconstructed networks with threshold $\theta = 0.4$ (see Fig. 6). The mean local clustering coefficients of all reconstructed networks are similar to those of the corresponding original networks. In Fig. 6a, we show that the degree distributions of the reconstructed networks for CALTECH with $r \in \{9, 16, 25, 64\}$ latent motifs converge toward that of the original network as we increase $r$.

Reconstructing CALTECH with larger values of $r$ appears to increase accuracy by including nodes with a larger degree than is possible for smaller values of $r$. In other words, low-rank reconstructions (i.e., those with small values of $r$) of CALTECH seem to recover only a small number of the edges of each node, even though it is able to achieve a large reconstruction accuracy (e.g., over 81% for $r = 9$).

We now consider cross-reconstruction accuracies $X \leftarrow Y$ in Fig. 5b–e, where $X$ is one of the Facebook networks CALTECH, HARVARD, MIT, and UCLA and $Y$ (with $Y \neq X$) is one of these four networks or one of the four synthetic networks $ER_2$, $WS_2$, $BA_2$, and $SBM_2$. From the cross-reconstruction accuracies and examination of the network structures of the latent motifs (see Appendix A.4 of the SI) in Fig. 4 (also see Supplementary Figs. 3, 8, 10, and 11), we draw a few conclusions at scale $k = 21$. First, the mesoscale structure of CALTECH is distinct from those of HARVARD, UCLA, and MIT. This is consistent with prior studies of these networks[27,40]. Second, CALTECH's mesoscale structures at scale $k = 21$ are higher-dimensional than those of the other three universities' Facebook networks. Third, CALTECH has a lot more communities with at least 10 nodes than the other three universities' Facebook networks (also see Supplementary Figs. 2 and 3). Fourth, the BA network $BA_2$ captures the mesoscale structure of MIT, HARVARD, and UCLA at scale $k = 21$ better than the synthetic networks that we generate from the ER, WS, and SBM models. However, for all $r \in \{9, 15, 25, 49\}$, the network $SBM_2$ captures the mesoscale structures of CALTECH better than all other networks in Fig. 5b except for CALTECH itself. See Appendix E.5 of the SI for further discussion.

We also comment briefly about the cross-reconstruction experiments in Fig. 5 that use latent motifs that we learn from ER networks. For instance, when reconstructing MIT, HARVARD, and UCLA using latent motifs that we learn from $ER_2$, we obtain a reconstruction accuracy of at least 72%. This may seem unreasonable at first glance because the latent motifs that we learn from $ER_2$ should not have any information about the Facebook networks. However, all of these networks are sparse (with edge densities of at most 0.02) and we are sampling subgraphs using $k$-paths. The $k$-node subgraphs that are induced by uniformly random $k$-paths in these sparse networks have only a few off-chain edges (see Fig. 2). For example, the $k$-node subgraphs that we sample from the sparse ER network $ER_2$ tend to have $k$-node paths and a few extra off-chain edges (see Fig. 2). A similarly sparse or sparser network, such as UCLA (whose edge density is about 0.0036) has similar subgraph patterns (despite the fact that, unlike the

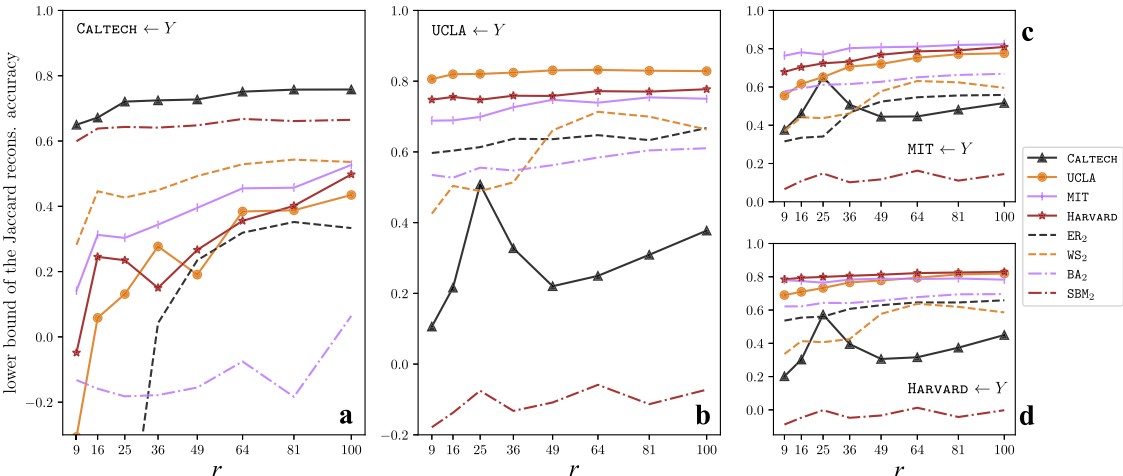

**Fig. 7 | Lower bounds on the Jaccard reconstruction accuracy for the cross-reconstruction experiments in Fig. 5.** Subtracting both sides of the inequality (1) from 1 implies that the Jaccard reconstruction accuracy $1 - \text{JD}(G, G_{\text{recons};W})$ is lower-bounded by $1 - \frac{1}{2(k-1)} \mathbb{E}_{\mathbf{x}}\left[\|A_{\mathbf{x}} - \hat{A}_{\mathbf{x};W}\|_1\right]$. This lower bound also measures the accuracy of reconstructing mesoscale patches of $G$ using latent motifs in $W$. In (**a**–**d**), we plot this lower bound for the same parameters as in Fig. 5b–e (i.e., $k = 21$ and $r \in \{6, 16, 25, 36, 49, 64, 81, 100\}$).

subgraphs of an ER network, the off-chain edges are not independent). This is the reason that we can reconstruct some networks with high accuracy by using latent motifs that we learn from a completely unrelated network.

One learns latent motifs by maximizing the accuracy of reconstructions of mesoscale patches using them, rather than by maximizing the network-reconstruction accuracy. In Fig. 7a–d, we illustrate that self-reconstructions of mesoscale patches are more accurate than cross-reconstructions of mesoscale patches. However, because network reconstruction involves taking the mean of the reconstructed weights of an edge from multiple mesoscale patches that include that edge, an accurate reconstruction of mesoscale patches need not always entail accurate network reconstruction. In Fig. 5, we see that the self-reconstruction $X \leftarrow X$ is more accurate than the cross-reconstructions $X \leftarrow Y$ for $Y \neq X$ for almost all choices of networks $X$ and $Y$ and the parameter $r$. The two exceptions are $(X, Y, r) = (\texttt{MIT}, \texttt{HARVARD}, 25)$ and $(X, Y, r) = (\texttt{MIT}, \texttt{UCLA}, 25)$, although the cross-reconstruction accuracies in these cases are at most 2% larger than the self-reconstruction accuracy.

The above discussion suggests an important question: If a network dictionary is effective at approximating the mesoscale patches of a network, what reconstruction accuracy does one expect? In the present paper, we state and prove a theorem (see Theorem 1) that answers this question. Specifically, we prove mathematically that a Jaccard reconstruction error (see (31) in the SI) of a weighted reconstructed network (i.e., without thresholding edge weights as in Fig. 5) is upper-bounded by the mean error of approximating the mesoscale patches (i.e., $k$-node subgraphs) of a network by the $k$-path latent motifs divided by $2(k-1)$.

**Theorem 1.** Consider a network $G$ and a network dictionary $W$ of $k$-node latent motifs, and let $G_{\text{recons};W}$ denote the weighted reconstructed network that we obtain using our NDR algorithm. The Jaccard distance JD between $G$ and $G_{\text{recons};W}$ satisfies the bound

$$\text{JD}(G, G_{\text{recons};W}) \leq \frac{1}{2(k-1)} \mathbb{E}_{\mathbf{x}}\left[\|A_{\mathbf{x}} - \hat{A}_{\mathbf{x};W}\|_1\right], \qquad (1)$$

where $\mathbf{x}$ is a uniformly random $k$-path of $G$, the matrix $A_{\mathbf{x}}$ is the $k \times k$ adjacency matrix of the subgraph that is induced by the node set of $\mathbf{x}$, and $\hat{A}_{\mathbf{x};W}$ is the best nonnegative linear approximation of $A_{\mathbf{x}}$ that we obtain using the latent motifs in the network dictionary $W$.

See Appendix F.3 and Theorem F.4 in the SI for precise statements of Theorem 1 and the relevant definitions.

Our NDL algorithm (see Algorithm NDL in the SI) finds a network dictionary that approximately minimizes the upper bound in the inequality (1). (See Theorem NDL in the SI.) Using an arbitrary network dictionary is likely to yield larger values of the upper bound. Therefore, according to Theorem 1, it is likely to yield a less accurate weighted reconstructed network. For instance, for a network dictionary that consists of a single $k$-path, the aforementioned upper bound is the mean number of off-chain edges in the mesoscale patches of a network divided by $k - 1$. (See the inequality (1).) For an ER network with expected edge density $p$, this upper bound equals $kp/2$ in expectation. At scale $k = 20$ for $\text{ER}_2$, this value is 0.5. Consequently, we expect a reconstruction accuracy of at least 50% when reconstructing $\text{ER}_2$ using latent motifs (such as the ones from UCLA in Fig. 4) that have large on-chain entries and small off-chain entries. Substituting the edge density of UCLA for $p$, we expect a reconstruction accuracy of at least 94%. However, according to Theorem 1, the lower bound of the reconstruction accuracy that we obtain using latent motifs of UCLA is about 80%. Therefore, when we use latent motifs from UCLA, we expect to obtain many more off-chain edges in mesoscale patches than what we expect when using latent motifs from an ER network with the same edge density. We plot the lower bound of the reconstruction accuracy in Fig. 7a–d for the parameters in Fig. 5b–e (i.e., $k = 21$ and $r \in \{6, 16, 25, 36, 49, 64, 81, 100\}$). The lower bounds for the self-reconstructions are not too far from the actual reconstruction accuracies for the unweighted reconstructed networks in Fig. 5b–e (they are within 20% for CALTECH and UCLA and within 10% for MIT and HARVARD for all $r$), but we observe much larger accuracy gaps for the cross-reconstruction experiments. (For example, there is at least a 50% difference for UCLA ← CALTECH.) This indicates that, even when using latent motifs that are not very efficient at approximating mesoscale patches, one can obtain unweighted reconstructions that are significantly more accurate than those that are guaranteed by the theoretically proven bounds.

## Network-denoising experiments

We consider the following network-denoising problem (which is closely related to the anomalous-subgraph-detection problem in Fig. 1). Suppose that we are given an observed network $G_{\text{obs}} = (V, E_{\text{obs}})$ with a node set $V$ and edge set $E_{\text{obs}}$ and that we are asked to find an unknown network $G_{\text{true}} = (V, E_{\text{true}})$ with the same node set $V$ but a possibly

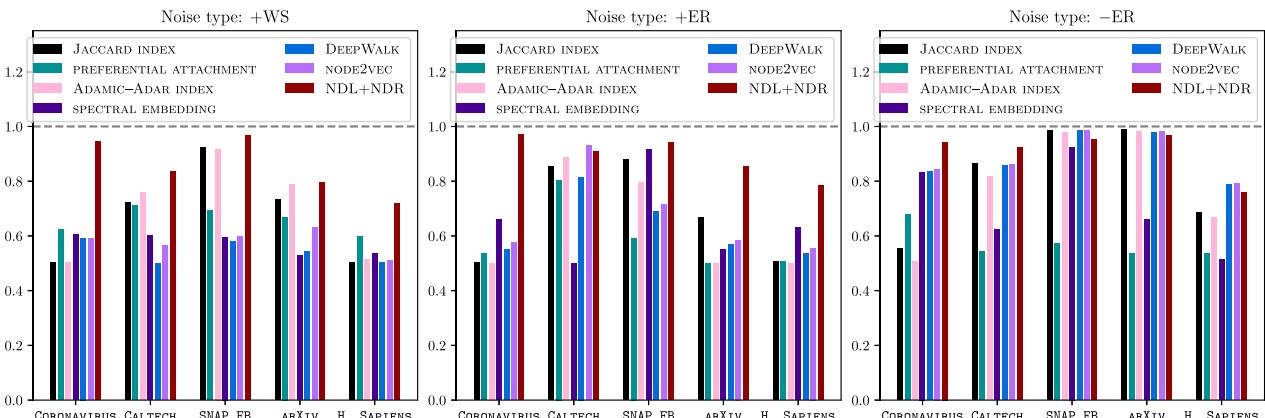

**Fig. 8 | Applications of our NDL and NDR algorithms to network denoising with additive and subtractive noise on a variety of real-world networks.** In our experiments with subtractive noise, we corrupt a network by removing 50% of its edges uniformly at random. We seek to classify the nonedges of the corrupted network as true edges (i.e., removed edges) and false edges (i.e., nonedges of the original network), respectively. In our experiments with additive noise, we corrupt a network by uniformly randomly adding 50% of the number of its edges (i.e., 1000 random edges) for all but one network (we add 30000 random edges for H. SAPIENS) that we generate using the WS model. We seek to classify the edges of the resulting corrupted network as true edges (i.e., original edges) and false edges (i.e.,

added edges). To perform classification for a network, we first use NDL to learn latent motifs from a corrupted network and then reconstruct the network using NDR to assign a confidence value to each potential edge. We then use these confidence values to infer the correct labeling of potential edges of the uncorrupted network. Importantly, we never use information from the original networks to denoise the corrupted networks. For each network, we report the areas under the curves (AUCs) of the receiver-operating characteristic (ROC) curves, which plot false-positive rates on the horizontal axis and true-positive rates on the vertical axis. See Supplementary Figs. 5–7 for the values of other binary-classification measures.

different edge set $E_{\text{true}}$. We interpret $G_{\text{obs}}$ as a corrupted version of a true network $G_{\text{true}}$ that we observe with some uncertainty. To simplify the setting, we consider two types of network denoising. In the first type of network denoising, we consider *additive noise*[18,19,41]. We suppose that $G_{\text{obs}}$ is a corrupted version of $G_{\text{true}}$ that includes false edges (i.e., $E_{\text{obs}} \supseteq E_{\text{true}}$), and we seek to classify all edges of $G_{\text{obs}}$ as positives (i.e., edges of $G_{\text{true}}$) or negatives (i.e., false edges of $G_{\text{obs}}$ or equivalently nonedges of $G_{\text{true}}$). This network-denoising setting is identical to the anomalous-subgraph-detection problem in Fig. 1, except that now we label the false edges as negatives. We interpreted them as positives when we computed the F-score (i.e., the harmonic mean of the precision and recall scores) in Fig. 1f. In the second type of network denoising, we consider *subtractive noise* (which is often called edge prediction)[42–46]. We assume that $G_{\text{obs}}$ is a partially observed version of $G_{\text{true}}$ (i.e., $E_{\text{obs}}$ is a proper subset of $E_{\text{true}}$), and we seek to classify nonedges of $G_{\text{obs}}$ into positives (i.e., nonedges of $G_{\text{true}}$) and negatives (i.e., edges of $G_{\text{true}}$). There are many more positives than negatives because $G_{\text{true}}$ is sparse (i.e., the edge density is low), so we restrict the classification task to a subset $E_{\text{nonedge}}$ of $E_{\text{obs}}$ that includes all negatives and an equal number of positives. We will shortly discuss how we choose $E_{\text{nonedge}}$.

Given a true network $G_{\text{true}} = (V, E_{\text{true}})$, we generate an observed (i.e., corrupted) network $G_{\text{obs}} = (V, E_{\text{obs}})$ as follows. In the additive-noise setting, we create two types of corrupted networks. We create the first type of corrupted network by adding false edges in a structured way by generating them using the WS model. We consider the networks CALTECH, SNAP FB, ARXIV, CORONAVIRUS, and H. SAPIENS. We select 100 nodes for four of the networks (the exception is that we use 500 nodes for H. SAPIENS) uniformly at random and generate 1000 new edges (we generate 30000 new edges for H. SAPIENS) according to the WS model. In this corrupting WS network, each node in a ring of 100 nodes is adjacent to its 20 nearest neighbors and we uniformly randomly choose 30% of the edges to rewire. When rewiring an edge, we choose one of its two ends with equal probability of each, and we attach this end to a node of the network that we choose uniformly at random. We then add these newly generated edges to the original network. We refer to this noise type as +WS. We create the

second type of corrupted network by choosing 5% of the nodes uniformly at random and adding an edge between each pair of chosen nodes with independent probability 0.3. We refer to this noise type as +ER. In the subtractive-noise setting, we obtain $G_{\text{obs}}$ from $G_{\text{true}}$ by removing half of the existing edges, which we choose uniformly at random, such that the remaining network is connected. We refer to this noise type as −ER.

For each observed network $G_{\text{obs}}$, we apply NDL at scale $k = 21$ with $r \in \{2, 25\}$ to learn a network dictionary $W_{\text{obs}}$. We construct another network dictionary $\bar{W}_{\text{obs}}$ by removing the on-chain edges from all of the latent motifs in $W_{\text{obs}}$. (For further discussion, see the Methods section.) This gives a total of four network dictionaries, corresponding to the two values of $r$ and whether or not we keep the on-chain edges of the latent motifs. With each of the network dictionaries, we use NDR to reconstruct a network $G_{\text{recons}}$ by approximating mesoscale patches of $G_{\text{obs}}$ using latent motifs in $W_{\text{obs}}$. (We compute $G_{\text{recons}}$ without using any information about $G_{\text{true}}$.) We anticipate that the reconstructed network $G_{\text{recons}}$ is similar to its corresponding original (i.e., uncorrupted) network $G_{\text{true}}$. The reconstruction algorithm outputs a weighted network $G_{\text{recons}}$, where the weight of each edge is our confidence that the edge is a true edge of that network. For denoising subtractive (respectively, additive) noise, we classify each nonedge (respectively, each edge) of a corrupted network as positive if its weight in $G_{\text{recons}}$ is strictly larger than some threshold $\theta$ and as negative otherwise. By varying $\theta$, we construct a receiver-operating characteristic (ROC) curve that consists of points whose horizontal and vertical coordinates are the false-positive rates and true-positive rates, respectively. For denoising the −ER (respectively, +ER and +WS) noise, one can also infer an optimal value of $\theta$ for a 50% training set of nonedges (respectively, edges) of $G$ with known labels and then use this value of $\theta$ to compute classification measures such as accuracy and precision.

In Fig. 8, we compare the performance of our network-denoising approach to the performance of several existing approaches using the real-world networks CALTECH, SNAP FB, ARXIV, CORONAVIRUS, and H. SAPIENS. We use four classical approaches (the JACCARD INDEX, PREFERENTIAL ATTACHMENT, the ADAMIC–ADAR INDEX, and a SPECTRAL EMBEDDING)[43,47] and two more recent methods (DEEPWALK[14] and NODE2VEC[15]) that are based

on network embeddings. Let $N(x)$ denote the set of neighbors of node $x$ of a network. For the Jaccard index, preferential attachment, and the Adamic–Adar index, the confidence score (which plays the same role as an edge weight in a reconstructed network) that the nodes $x$ and $y$ are adjacent via a true edge is $|N(x) \cap N(y)|/|N(x) \cup N(y)|$, $|N(x)| \cdot |N(y)|$, and $\sum_{z \in N(x) \cap N(y)} 1/\ln|N(z)|$, respectively.

We now discuss how we choose the set $E_{\text{nonedge}}$ of nonedges of $G_{\text{obs}}$ for our subtractive-noise experiments. First, we note that it is unlikely that many edges in the set $E_{\text{deleted}}$ of deleted edges are between two small-degree nodes. If we simply choose $E_{\text{nonedge}}$ as a uniformly random subset of the set of all nonedges of $G_{\text{obs}}$ with a given size $|E_{\text{deleted}}|$, then it is likely that we will choose many nonedges between small-degree nodes. Consequently, the resulting classification problem is easy for existing methods, such as the Jaccard index and preferential attachment, that are based on node degrees. For example, consider a star network with five leaves (i.e., degree-1 nodes). In this network, a uniformly randomly chosen nonedge is always attached to two degree-1 nodes, but a uniformly randomly chosen edge is always attached to one degree-5 node (i.e., the center node) and one degree-1 node. In our experiments, to reduce the size-biasing of node degrees, we choose each nonedge of $E_{\text{nonedge}}$ with a probability that is proportional to the product of the degrees of the two associated nodes.

We show results in the form of means of the areas under the curves (AUCs) of the ROC curves for five independent runs of each approach. In Fig. 8, we see that our approach performs competitively in all of our experiments, particularly for denoising additive noise (i.e., anomalous-subgraph detection). For example, when we add 1000 false edges that we generate from the WS model to CORONAVIRUS (which has 2463 true edges), our approach yields an AUC of 0.94. We obtain the second best AUC (it is only 0.61) using preferential attachment. For noise of type +ER, we add 804 false edges to CORONAVIRUS; our approach achieves the best AUC (it is 0.97) and spectral embedding achieves the second best AUC (it is 0.66).

In Fig. 5a, we saw that we can use a small number of latent motifs to reconstruct the social and PPI networks in our denoising experiments in Fig. 8. Because NDL learns a small number of latent motifs that are able to successfully give an approximate basis for all mesoscale patches, these motifs should not be affected significantly by false edges between the nodes of a small subset of the entire node set. Consequently, the latent motifs in $W_{\text{obs}}$ that we learn from the observed network $G_{\text{obs}}$ may still be effective at approximating mesoscale patches of the true network $G_{\text{true}}$, so the network $G_{\text{recons}}$ that we reconstruct using $G_{\text{obs}}$ and $W_{\text{obs}}$ may be similar to $G_{\text{true}}$.

## Discussion

We now highlight key conclusions, ideas, and limitations of our work. We first summarize our main results, discuss their importance, and briefly indicate relevant ideas for future studies. We then highlight key limitations and related salient points.

In the present paper, we introduced a mesoscale network structure, which we call *latent motifs*, that consists of $k$-node subgraphs that are building blocks of the connected $k$-node subgraphs of a network. In contrast to ordinary motifs[3], which refer to overrepresented $k$-node subgraphs (especially for small $k$) of a network, nonnegative linear combinations of our latent motifs approximate $k$-node subgraphs that are induced by uniformly random $k$-paths of a network. We also established algorithmically and theoretically that one can accurately approximate a network if one has a dictionary of latent motifs that can accurately approximate mesoscale structures of the network.

Our computational experiments in Fig. 4 (see also Supplementary Fig. 3) demonstrated that latent motifs can have distinctive network structures. Our computational experiments in Figs. 5 and 8 illustrated that various social, collaboration, and PPI networks have

low-rank[48] mesoscale structures, in the sense that a few latent motifs (e.g., $r = 25$ of them, but see Fig. 5 for other choices of $r$) that we learn using NDL are able to reconstruct, infer, and denoise the edges of a network using our NDR algorithm. We hypothesize that such low-rank mesoscale structures are a common feature of networks beyond the examined social, collaboration, and PPI networks. As we have illustrated in our paper, one can leverage mesoscale structures to perform important tasks, such as network denoising, so it is important in future studies to explore the level of generality of our insights.

In our work, we examined latent motifs in ordinary graphs. However, notions of motifs have been developed for several more general types of networks, including temporal networks (in which nodes, edges, and edge weights can change with time)[49] and multilayer networks (in which, e.g., nodes can be adjacent to each other via multiple types of relationships)[50]. We have not examined latent motifs in such network structures, and it is worthwhile to extend our approach and algorithms to these situations.

To help readers interpret and use our methods in a scientifically correct matter, it is important to highlight key limitations and related salient points. We discuss these points in the next several paragraphs.

First, it is possible for two sets of latent motifs to be equally effective at reconstructing the same network. Therefore, although one can interpret the structures in latent motifs as mesoscale structures of a network, one cannot conclude that other mesoscale structures (which are not in a given set of latent motifs) do not also occur in the network.

Second, our NDL algorithm approximately computes a network dictionary that successfully reconstructs the mesoscale patches of a network. It does not compute a network dictionary to reconstruct a network itself. Although our theoretical bound on the reconstruction error (see Theorem 1) implies that such a network dictionary should also be effective at reconstructing networks, it is still necessary to empirically verify the actual efficacy of doing so.

Third, our theoretical bound on the reconstruction error illustrates that it is possible to successfully reconstruct a very sparse network using latent motifs that one learns from a radically different but similarly sparse network at a given scale $k$ (see Fig. 5e). To better distinguish distinct sparse networks from each other, one can use a scale $k$ that is large enough so that $k$-node mesoscale patches have many off-chain edges and latent motifs at that scale are sufficiently different for different networks. For example, see the latent motifs at scale $k = 51$ in Supplementary Figs. 8–11. Naturally, using a larger scale $k$ increases the computational cost of our approach.

Fourth, although our method for network denoising is competitive (especially for anomalous-subgraph detection), it does not always outperform all existing methods, and some of those methods are much simpler than ours. For instance, for edge-prediction tasks, it seems that our method is often more conservative than the other examined methods at detecting unobserved edges. (See Supplementary Figs. 5–7.) Therefore, we recommend using our method in conjunction with existing methods for such tasks.

## Methods

We briefly discuss our algorithms for network dictionary learning (NDL) and network denoising and reconstruction (NDR). We also provide a detailed description of our real-world and synthetic networks.

We restrict our present discussion to networks that one can represent as a graph $G = (V, E)$ with a node set $V$ and an edge set $E$ without directed edges or multi-edges (but possibly with self-edges). In the SI, we give an extended discussion that applies to more general types of networks. Specifically, in that discussion, we no longer restrict edges to have binary weights; instead, the weights can have continuous nonnegative values. See Appendix A.1 of the SI.

## Motif sampling and mesoscale patches of networks

The connected $k$-node subgraphs of a network are natural candidates for the network's mesoscale patches. These subgraphs have $k$ nodes that inherit their adjacency structures from the original networks from which we obtain them. It is convenient to consider the $k \times k$ adjacency matrices of these subgraphs, as we can then perform computations on the space of subgraphs. However, to do this, we need to address two issues. First, because the same $k$-node subgraph can have multiple (specifically, $k!$) different representations as an adjacency matrix (depending on the ordering of its nodes), we need an unambiguous way to choose an ordering of its nodes. Second, because most real-world networks are sparse[1], independently choosing a set of $k$ nodes from a network may yield only a few edges and thus may often result in a disconnected subgraph. Therefore, we need an efficient sampling algorithm to guarantee that we obtain connected $k$-node subgraphs when we sample from sparse networks.

We employ an approach that is based on *motif sampling*[21] both to choose an ordering of the nodes of a $k$-node subgraph and to ensure that we sample connected subgraphs from sparse networks. The key idea is to consider the random $k$-node subgraph that we obtain by sampling a copy of a template subgraph uniformly at random from a network. (We sample the nodes uniformly at random and include all of the network's edges between those sampled nodes.) We suppose that such a template is a $k$-path. A sequence $\mathbf{x} = (x_1, \ldots, x_k)$ of $k$ (not necessarily distinct) nodes is a *$k$-walk* if $x_i$ and $x_{i+1}$ are adjacent for all $i \in \{1, \ldots, k-1\}$. A $k$-walk is a *$k$-path* if all nodes in the walk are distinct (see Fig. 2). For each $k$-path $\mathbf{x} = (x_1, \ldots, x_k)$, we define the corresponding mesoscale patch of a network to be the $k \times k$ matrix $A_\mathbf{x}$ such that $A_\mathbf{x}(i,j) = 1$ if nodes $x_i$ and $x_j$ are adjacent and $A_\mathbf{x}(i,j) = 0$ if they are not adjacent. This is the adjacency matrix of the $k$-node subgraph of the network with nodes $x_1, \ldots, x_k$. One can use one of the Markov-chain Monte Carlo (MCMC) motif-sampling algorithms of Lyu et al.[21] to efficiently and uniformly randomly sample a $k$-walk from a sparse network. By only accepting samples in which the $k$-walk has $k$ distinct nodes (i.e., so that it is $k$-path), we efficiently sample a uniformly random $k$-path from a network, as long as $k$ is not too large. If $k$ is too large, one has to reject too many samples of $k$-walks that are not $k$-paths. The expected number of rejected samples is approximately the number of $k$-walks divided by the number of $k$-paths. The number of $k$-walks on a network grows monotonically with $k$, but the number of $k$-paths can decrease with $k$. (See Appendix B of the SI for a detailed discussion.) Consequently, by repeatedly sampling $k$-paths $\mathbf{x}$, we obtain a data set of mesoscale patches $A_\mathbf{x}$ of a network.

## Algorithm for network dictionary learning (NDL)

We now present the basic structure of the algorithm that we employ for *network dictionary learning* (NDL)[13]. Suppose that we compute all possible $k$-paths $\mathbf{x}_1, \ldots, \mathbf{x}_M$ and their corresponding mesoscale patches $A_{\mathbf{x}_t}$ (which are $k \times k$ binary matrices), with $t \in \{1, \ldots, M\}$, of a network. We column-wise vectorize (i.e., we place the second column underneath the first column and so on; see Algorithm A4 in the SI) each of these $k \times k$ mesoscale patches to obtain a $k^2 \times M$ data matrix $X$. We then apply nonnegative matrix factorization (NMF)[25] to obtain a $k^2 \times r$ nonnegative matrix $W$ for some fixed integer $r \geq 1$ to yield an approximate factorization $X \approx WH$ for some nonnegative matrix $H$. From this procedure, we approximate each column of $X$ by a nonnegative linear combination of the $r$ columns of $W$; its coefficients are the entries of the corresponding column of $H$. If we let $\mathcal{L}_i$ be the $k \times k$ matrix that we obtain by reshaping the $i^{\text{th}}$ column of $W$ (using Algorithm A5 in the SI), then $\mathcal{L}_1, \ldots, \mathcal{L}_r$ are the learned latent motifs; they form a network dictionary. The set of these latent motifs is an approximate basis (but not a subset) of the set $\{A_{\mathbf{x}_1}, \ldots, A_{\mathbf{x}_M}\}$ of mesoscale patches. For instance, latent motifs have entries that take continuous values between 0 and 1, but mesoscale patches have binary entries. We can regard each $\mathcal{L}_i$ as the $k$-node weighted network with node set $\{1, \ldots, k\}$

and weighted adjacency matrix $\mathcal{L}_i$. See Fig. 4 for an illustration of latent motifs as weighted networks.

The scheme in the paragraph above requires us to store all possible mesoscale patches of a network, entailing a memory requirement that is at least of order $k^2 M$, where $M$ denotes the total number of mesoscale patches of a network. Because $M$ scales with the size (i.e., the number of nodes) of the network from which we sample subgraphs, we need unbounded memory to handle arbitrarily large networks. To address this issue, Algorithm NDL implements the above scheme in the setting of online learning, in which subsets (so-called minibatches) of data arrive in a sequential manner and one does not store previous subsets of the data before processing new subsets. Specifically, at each iteration $t \in \{1, 2, \ldots, T\}$, we process a sample matrix $X_t$ that is smaller than the full matrix $X$ and includes only $N \ll M$ mesoscale patches, where one can take $N$ to be independent of the network size. Instead of using a standard NMF algorithm for a fixed matrix[51], we employ an online NMF algorithm[13,52] that one can use on sequences of matrices. The intermediate dictionary matrices $W_t$ that we obtain by factoring the sample matrix $X_t$ typically improve as we iterate[13,52]. In Algorithm NDL in the SI, we give a complete implementation of the NDL algorithm.

## Algorithm for network denoising and reconstruction (NDR)

Suppose that we have an image patch $\gamma$ of size $k \times k$ pixels and a set of basis images $\beta_1, \ldots, \beta_r$ of the same size. We can reconstruct the image $\gamma$ using the basis images $\beta_1, \ldots, \beta_r$ by finding nonnegative coefficients $a_1, \ldots, a_r$ such that the linear combination $\hat{\gamma} = a_1 \beta_1 + \cdots + a_r \beta_r$ is as close as possible to $\gamma$. The basis images determine what shapes and colors of the original image to capture in the reconstruction $\hat{\gamma}$. In the standard pipeline for image denoising and reconstruction[22,23,53], one assumes that the size $k \times k$ of the image patches is much smaller than the size of the full image $\gamma$. One can then sample a large number of $k \times k$ overlapping patches $\gamma_1, \ldots, \gamma_M$ of the image $\gamma$ and obtain the best linear approximations $\hat{\gamma}_1, \ldots, \hat{\gamma}_M$ of them using the basis images $\beta_1, \ldots, \beta_r$. Because the $k \times k$ patches $\gamma_1, \ldots, \gamma_M$ overlap, each pixel $(I,J)$ of $\gamma$ can occur in multiple instances of $\gamma_1, \ldots, \gamma_M$. Therefore, we take the mean of the corresponding values in the mesoscale reconstructions $\hat{\gamma}_1, \ldots, \hat{\gamma}_M$ as the value of the pixel $(I,J)$ in the reconstruction $\hat{\gamma}$.

As an illustration, we reconstruct the color image in Fig. 9a in two ways to yield the images in Fig. 9b, c. In Fig. 9d, we show a dictionary with 25 basis images of size $21 \times 21$ pixels. We uniformly randomly choose the color of each pixel from all possible colors (which we represent as vectors in $[0, 256]^3$ for red–green–blue (RGB) weights). The basis images do not include any information about the original image in Fig. 9a, so the linear approximation of the $21 \times 21$ mesoscale patches of the image in Fig. 9a using the basis images in Fig. 9d may be inaccurate. However, when we reconstruct the entire image from Fig. 9a using the basis images in Fig. 9d, we do observe some basic geometric information of the original image. In Fig. 9b, we show the image that results from this reconstruction. Importantly, the image reconstruction in Fig. 9b uses both the basis images and the original image that one seeks to reconstruct. Unfortunately, the colors have averaged out and become neutral, so the reconstructed image is monochrome. Using a smaller (e.g., $5 \times 5$) randomly generated (with each pixel again taking an independently and uniformly chosen color) basis-image set for reconstruction results in a monochrome (but sharper) reconstructed image. Notably, one can learn the basis images in Fig. 9e from the image in Fig. 9f using nonnegative matrix factorization (NMF)[25]. The image in Fig. 9f is in black and white, so the images in Fig. 9e are also in black and white. The corresponding reconstruction in Fig. 9c has nicely captured shapes of the original image, although we have lost the color information of the original image in Fig. 9a and the reconstruction in Fig. 9c is thus in black and white.

A network analogue of the above patch-based image reconstruction proceeds as follows. Given a network $G = (V, E)$ and latent motifs

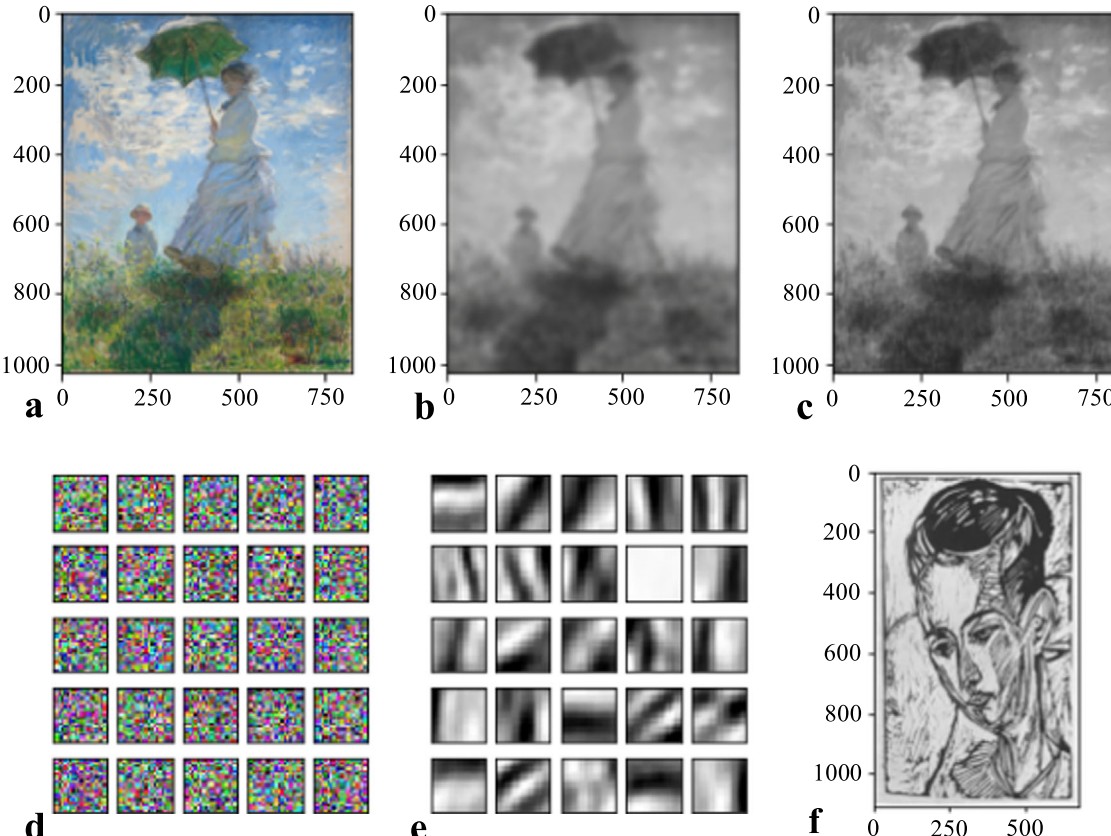

**Fig. 9 | Illustration of image reconstruction using two image dictionaries.** The (**a**) image `Woman with a Parasol – Madame Monet and Her Son` (Claude Monet, 1875). In (**b**, **c**), we show reconstructions of the image. The image in (**b**) is a reconstruction of the image in (**a**) using the dictionary with 25 basis images of size 21 × 21 pixels in (**d**). We uniformly randomly choose the color of each pixel from all possible colors (which we represent as vectors in [0, 256]³ for red–green–blue (RGB) weights). The image in (**c**) is a reconstruction of the image in (**a**) using the dictionary with 25 basis images of size 21 × 21 pixels in (**e**). We learn this basis from the image in (**f**) using NMF. The image in (**f**) is from the collection `Die Graphik Ernst Ludwig Kirchners bis 1924, von Gustav Schiefler Band I bis 1916` (Accession Number 2007.141.9, Ernst Ludwig Kirchner, 1926). We use the images in (**a**, **f**) with permission from the National Gallery of Art in Washington, DC, USA.

$\mathcal{L}_1, \ldots, \mathcal{L}_r$ (which we do not necessarily compute from $G$; see Fig. 9), we obtain a weighted network $G_{\text{recons}}$ using the same node set $V$ and a weighted adjacency matrix $A_{\text{recons}} : V^2 \to \mathbb{R}$. To do this, we first use a MCMC motif-sampling algorithm of Lyu et al.[21] with rejection sampling to sample a large number $T$ of $k$-paths $\mathbf{x}_1, \ldots, \mathbf{x}_T : \{1, \ldots, k\} \to V$ of $G$. (For details, see Algorithm IM in the SI.) We then determine the corresponding mesoscale patches $A_{\mathbf{x}_1}, \ldots, A_{\mathbf{x}_T}$ of $G$. We approximate each mesoscale patch $A_{\mathbf{x}_t}$, which is a $k \times k$ unweighted matrix, by a non-negative linear combination $\hat{A}_{\mathbf{x}_t}$ of the latent motifs $\mathcal{L}_i$. We seek to replace each $A_{\mathbf{x}_t}$ by $\hat{A}_{\mathbf{x}_t}$ to construct the weighted adjacency matrix $A_{\text{recons}}$. To do this, we define $A_{\text{recons}}(x, y)$ for each $x, y \in V$ as the mean of $\hat{A}_{\mathbf{x}_t}(a, b)$ over all $t \in \{1, \ldots, T\}$ and all $a, b \in \{1, \ldots, k\}$ such that $\mathbf{x}_t(a) = x$ and $\mathbf{x}_t(b) = y$. We state this network-reconstruction algorithm precisely in Algorithm NDR in the SI. See Appendix D of the SI for more details.

**A comparison of our work to the prior research by Lyu et al.[13]**
Recently, Lyu et al.[13] proposed a preliminary approach for the algorithms that we study in the present paper − the NDL algorithm with $k$-walk sampling and the NDR algorithm for network-reconstruction tasks − as an application to showcase a theoretical result about the convergence of online NMF for data samples that are not independently and identically distributed (IID)[13,Thm.1]. A notable limitation of the NDL algorithm in Lyu et al.[13] is that one cannot interpret the elements of a network dictionary as latent motifs and one thus cannot associate them directly with mesoscale structures of a network. Additionally, Lyu et al.[13] did not include any theoretical analysis of either the convergence or the correctness of network reconstruction, so it is unclear

from that work whether or not one can reconstruct a network using the low-rank mesoscale structures that are encoded in a network dictionary. Moreover, one cannot use the NDR algorithm of Lyu et al.[13] to denoise additive noise unless one knows in advance that the noise is additive (rather than subtractive) before denoising (see Lyu et al.[13,Rmk.4]). In the present paper, we build substantially on the research by Lyu et al.[13] and provide a much more complete computational and theoretical framework to analyze low-rank mesoscale structures in networks. In particular, we overcome all of the aforementioned limitations. In Supplementary Table 1, we summarize the key differences between our work and Lyu et al.[13].

The most significant theoretical advance of the present paper is a relationship between the reconstruction error and the error from approximating mesoscale patches by latent motifs, with an explicit dependence on the number $k$ of nodes in subgraphs. We state this result in Theorem F.10(**iii**) in the SI. Informally, Theorem F.10(**iii**) states that one can accurately reconstruct a network if one has a dictionary of latent motifs that can accurately approximate the mesoscale patches of a network. In Theorem 1, we stated this theoretical result in an informal mathematical style. See Theorem F.10 in the SI for a mathematically precise statement of this theorem. In Fig. 7, we showed supporting numerical experiments. To prove Theorem F.10(**iii**), we show that the sequence of weighted adjacency matrices of the reconstructed networks converges as the number of iterations that one uses for network reconstruction tends to infinity and that the limiting weighted adjacency matrix has an explicit formula. We state and prove these results in Theorem F.10(**i,ii**).

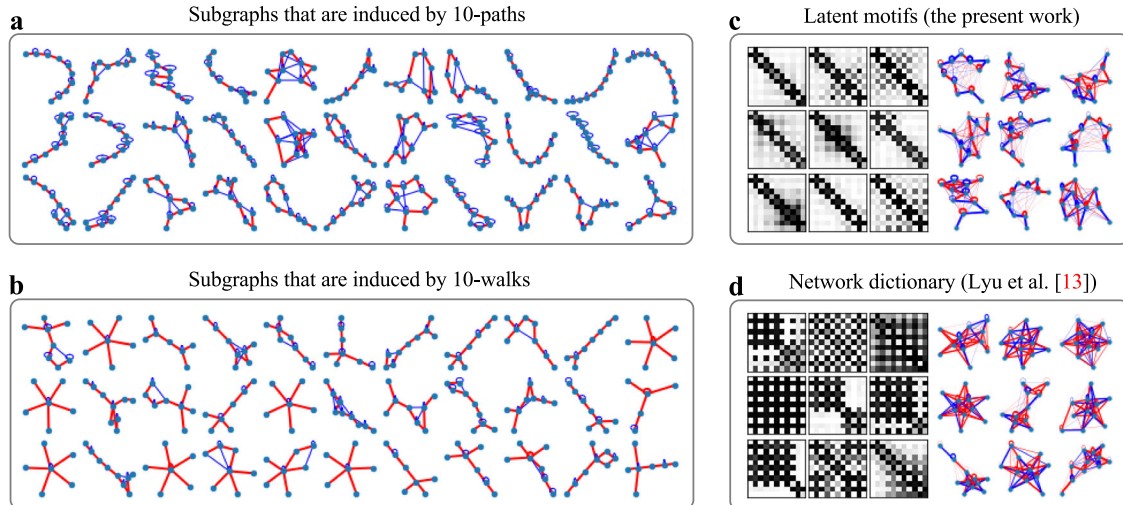

**Fig. 10 | Improved interpretability of latent motifs over Lyu et al.[13].** We compare subgraphs of CORONAVIRUS PPI that are induced by node sets that we sample using (**a**) uniformly random $k$-paths and (**b**) uniformly random $k$-walks with $k = 10$. We also compare the network dictionary with $r = 9$ latent motifs of CORONAVIRUS PPI that we determine using (**c**) the NDL algorithm (see Algorithm NDL in the SI) in the present paper to (**d**) the network dictionary that we determine using the NDL algorithm of Lyu et al.[13]. We also show the weighted adjacency matrices of the latent motifs. The 10-walks on the network tend to visit the same nodes many times. Consequently, one cannot regard the $10 \times 10$ mesoscale patches that correspond to those walks as the adjacency matrices of $k$-node subgraphs of the network. Additionally, the networks in the network dictionary in (**d**) have clusters of several large-degree nodes, even though the original network does not possess such mesoscale structures.

We now elaborate on the use of $k$-path sampling in our NDL algorithm to ensure that one can interpret the network-dictionary elements as latent motifs. The NDL algorithm of Lyu et al.[13] uses a $k$-walk motif-sampling algorithm of Lyu et al.[21]. That algorithm samples a sequence of $k$ nodes (which are not necessarily distinct) in which the $i$th node is adjacent to the $(i+1)$th node for all $i \in \{1, ..., k-1\}$. The $k$-walks that sample $k \times k$ subgraph adjacency matrices can have overlapping nodes, so some of the $k \times k$ adjacency matrices can correspond to subgraphs with fewer than $k$ nodes. If a network has a large number of such subgraphs, then the $k$-node latent motifs that one learns from the set of subgraph adjacency matrices can have misleading patterns that may not exist in any $k$-node subgraph of the network. This situation occurs in the network CORONAVIRUS PPI, where one obtains clusters of large-degree nodes from the learned latent motifs if one uses $k$-walk sampling. This misleading result arises from $k$-walks visiting the same large-degree node many times, rather than because $k$ distinct nodes of the network actually have such subgraph patterns (see Fig. 10). To resolve this issue, during the dictionary-learning phase, we combine MCMC $k$-walk sampling with rejection sampling so that we use only $k$-walks with $k$ distinct nodes (i.e., we use $k$-paths). Consequently, we learn $k$-node latent motifs only from $k \times k$ adjacency matrices that correspond to $k$-node subgraphs of a network. This guarantees that any network structure (e.g., large-degree nodes, communities, and so on) in the latent motifs must also exist in the network at scale $k$.

## Data sets

We use the following eight real-world networks:

(1) CALTECH: This connected network, which is part of the FACEBOOK100 data set[27] (and which was studied previously as part of the FACEBOOK5 data set[26]), has 762 nodes and 16651 edges. The nodes represent user accounts in the Facebook network of Caltech on one day in fall 2005, and the edges encode Facebook friendships between these accounts.

(2) MIT: This connected network, which is part of the FACEBOOK100 data set[27], has 6402 nodes and 251230 edges. The nodes represent user accounts in the Facebook network of MIT on one day in fall 2005, and the edges encode Facebook friendships between these accounts.

(3) UCLA: This connected network, which is part of the FACEBOOK100 data set[27], has 20453 nodes and 747604 edges. The nodes represent user accounts in the Facebook network of UCLA on one day in fall 2005, and the edges encode Facebook friendships between these accounts.

(4) HARVARD: This connected network, which is part of the FACEBOOK100 data set[27], has 15086 nodes and 824595 edges. The nodes represent user accounts in the Facebook network of Harvard on one day in fall 2005, and the edges represent Facebook friendships between these accounts.

(5) SNAP FACEBOOK (with the shorthand SNAP FB[33]): This connected network has 4039 nodes and 88234 edges. This network is a Facebook network that was used as an example in a study of edge inference[15]. The nodes represent user accounts in the Facebook network on one day in 2012, and the edges represent Facebook friendships between these accounts.

(6) ARXIV ASTRO-PH (with the shorthand ARXIV[15,34]): This network has 18722 nodes and 198110 edges. Its largest connected component has 17903 nodes and 197031 edges. We use the complete network in our experiments. This network is a collaboration network between authors of astrophysics papers that were posted on the arXiv preprint server. The nodes represent scientists and the edges indicate coauthorship relationships. This network has 60 self-edges; these edges encode single-author papers.

(7) CORONAVIRUS PPI (with the shorthand CORONAVIRUS): This connected network is curated by theBiogrid.org[30–32] from 142 publications and preprints. It has 1536 proteins that are related to coronaviruses and 2463 protein–protein interactions (in the form of physical contacts) between them. This network is the largest connected component of the coronavirus PPI network that we downloaded on 24 July 2020; in total, there are 1555 proteins and 2481 interactions. Of the 2481 interactions, 1536 are for SARS-CoV-2 and were reported by 44 publications and preprints; the rest are related to coronaviruses that cause Severe

Acute Respiratory Syndrome (SARS) or Middle Eastern Respiratory Syndrome (MERS).

(8) HOMO SAPIENS PPI (with the shorthand H. SAPIENS)[15,30,54]: This network has 24407 nodes and 390420 edges. Its largest connected component has 24379 nodes and 390397 edges. We use the complete network in our experiments. The nodes represent proteins in the organism *Homo sapiens*, and the edges encode physical interactions between these proteins.

We use the following eight synthetic networks:

(9) ER$_1$ and ER$_2$: An Erdős–Rényi (ER) network[1,35], which we denote by ER$(n, p)$, is a random-graph model. The parameter $n$ is the number of nodes and the parameter $p$ is the independent, homogeneous probability that each pair of distinct nodes has an edge between them. The network ER$_1$ is an individual graph that we draw from ER$(5000, 0.01)$, and ER$_2$ is an individual graph that we draw from ER$(5000, 0.02)$.

(10) WS$_1$ and WS$_2$: A Watts–Strogatz (WS) network, which we denote by WS$(n, k, p)$, is a random-graph model to study the small-world phenomenon[1,36]. In the version of WS networks that we use, we start with an $n$-node ring network in which each node is adjacent to its $k$ nearest neighbors. With independent probability $p$, we then remove and rewire each edge so that it connects a pair of distinct nodes that we choose uniformly at random. The network WS$_1$ is an individual graph that we draw from WS$(5000, 50, 0.05)$, and WS$_2$ is an individual graph that we draw from WS$(5000, 50, 0.10)$.

(11) BA$_1$ and BA$_2$: A Barabási–Albert (BA) network, which we denote by BA$(n, n_0)$, is a random-graph model with a linear preferential-attachment mechanism[1,37]. In the version of BA networks that we use, we start with $n_0$ isolated nodes and we introduce new nodes with $n_0$ new edges each that attach preferentially (with a probability that is proportional to node degree) to existing nodes until we obtain a network with $n$ nodes. The network BA$_1$ is an individual graph that we draw from BA$(5000, 25)$, and BA$_2$ is an individual graph that we draw from BA$(5000, 50)$.

(12) SBM$_1$ and SBM$_2$: We use stochastic-block-model (SBM) networks in which each block is an ER network[38]. Fix disjoint, finite sets $C_1, \ldots, C_{k_0}$ and a $k_0 \times k_0$ matrix $B$ whose entries are real numbers between 0 and 1. An SBM network, which we denote by SBM$(C_1, \ldots, C_{k_0}, B)$, has the node set $V = C_1 \cup \cdots \cup C_{k_0}$. For each unordered node pair $\{x, y\}$, there is an edge between $x$ and $y$ with independent probability $B[i_0, j_0]$, with indices $i_0, j_0 \in \{1, \ldots, k_0\}$ such that $x \in C_{i_0}$ and $y \in C_{j_0}$. If $k_0 = 1$ and $B$ has a constant $p$ in all entries, this SBM specializes to the Erdős–Rényi (ER) random-graph model ER$(n, p)$ with $n = |C_1|$. The networks SBM$_1$ and SBM$_2$ are individual graphs that we draw from SBM$(C_1, \ldots, C_{k_0}, B)$ with $|C_1| = |C_2| = |C_3| = 1000$, where $B$ is the $3 \times 3$ matrix whose diagonal entries are 0.5 in both cases and whose off-diagonal entries are 0.001 for SBM$_1$ and 0.1 for SBM$_2$. Both networks have 3000 nodes; SBM$_1$ has 752450 edges and SBM$_2$ has 1049365 edges.

## Types of noise

We now describe the three types of noise in our network-denoising experiments. (See Fig. 8 and Supplementary Fig. 4.) These noise types are as follows:

(1) (Noise type: −ER) Given a network $G = (V, E)$, we choose a spanning tree of $G$ (such a tree includes all nodes of $G$) uniformly at random from all possible spanning trees. Let $E_0$ denote the set of edges of $G$ that are not in the edge set of that spanning tree. We then obtain a corrupted network $G'$ by uniformly randomly removing half of the edges in $E_0$ from $G$. Note that $G'$ is guaranteed to be connected.

(2) (Noise type: +ER) Given a network $G = (V, E)$, we uniformly randomly choose a set $E_2$ of pairs of nonadjacent nodes of $G$ of size $|E_2| = \lfloor |E|/2 \rfloor$, where $\lfloor \cdot \rfloor$ denotes the floor function. The corrupted network is $G' = (V, E \cup E_2)$; note that 50% of the edges of $G'$ are new.

(3) (Noise type: +WS) Given a network $G = (V, E)$, fix integers $n_0 \in \{1, \ldots, |V|\}$ and $k \in \{1, \ldots, n_0\}$, and fix a real number $p \in [0, 1]$. We uniformly randomly choose a subset $V_0 \subseteq V$ (with $|V_0| = n_0$) of the nodes of $G$. We generate a network $H = (V_0, E_3)$ from the Watts–Strogatz model WS$(n_0, k_0, p)$ using the node set $V_0$. We then obtain the corrupted network $G' = (V, E \cup E_3)$, which has $|E_3| = n_0 \lfloor k/2 \rfloor$ new edges. When $G$ is CALTECH, SNAP FB, ARXIV, or CORONAVIRUS, we use the parameters $n_0 = 100$, $k = 20$, and $p = 0.3$. In this case, $G'$ has 1000 new edges. When $G$ is H. SAPIENS, we use the parameters $n_0 = 500$, $k = 120$, and $p = 0.3$. In this case, $G'$ has 30000 new edges.

## Reporting summary

Further information on research design is available in the Nature Portfolio Reporting Summary linked to this article.

## Data availability

The data sets that we generated in the present study are available in the repository https://github.com/HanbaekLyu/NDL_paper. In the Methods section (see the subsection Data sets), we give references for the examined real-world networks.

## Code availability

Our code for our algorithms and simulations is publicly available in the repository https://github.com/HanbaekLyu/NDL_paper. A permanant DOI for the repository is available at https://zenodo.org/badge/latestdoi/301965967. We also provide user-friendly versions of our algorithms at https://github.com/jvendrow/Network-Dictionary-Learning as a PYTHON package NDLEARN.

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

## Acknowledgements

H.L. was supported by the National Science Foundation through grants 2206296 and 2010035; J.V. was supported by the National Science Foundation through grant 1740325; and Y.H.K. and M.A.P. were supported by the National Science Foundation (via grant 1922952) through the Algorithms for Threat Detection (ATD) program.

## Author contributions

All authors contributed to developing the ideas for the project; H.L., Y.H.K., and J.V. designed the algorithms and wrote the code to implement them; H.L. wrote the first draft of the manuscript; and H.L. and M.A.P. edited the manuscript.

## Competing interests

The authors declare no competing interests.
