## [Peer Review File · Nature Communications]

REVIEWER COMMENTS

Reviewer #1 (Remarks to the Author):

In the manuscript entitled “Learning Low-Rank latent mesoscale structures in networks”, the authors proposed a new approach for describing low-rank mesoscale structures in networks. They claimed this approach can be applied in a wide range of network-analysis tasks. We have several comments on the current version of this manuscript.

Major:

(1) The key component of the proposed approach is the so-called network dictionary learning (NDL). Yet, how these learned latent motifs reflect the micro/macro structures in the original networks, especially for the latent motifs with different mesoscales/walk length k ? For example, the latent motif of the Watts–Strogatz (WS) network shows two modules at $k=11$. But the original WS network is not supposed to have such communities or modules. How do the authors reconcile this contradiction?

(2) To fully support the authors’ statement, we suggest a detailed comparison between latent motifs with the micro/macro structures in the original networks. Otherwise, how can we make sure this approach truly extracts the real motifs in networks, considering that (i) the latent motifs for BA networks only have one hub at $k=21$. (ii) the social networks: UCLA, MIT and Harvard do not have community structures.

(3) The authors evaluated the method in two network reconstruction scenarios via latent motifs. First, if we know the network exactly and the dictionary W is quite efficient, then the reconstructed graph should be very similar to the original graph, in the sense that the accuracy is much higher in a broad range of threshold. Second, it’s not a big surprise that the peaks of accuracy are around 0.3 to 0.6, since the number of reconstructed edges for small (or large) threshold is too small (or large), respectively. Why didn’t the authors just calculate the Accuracy? Third, how to explain the result that the performance of reconstructing UCLA using ER network can still achieve 90% accuracy?

(4) Regarding the application of their approach to network denoising. (i) As link prediction is typically dealing with a highly imbalanced dataset, it’s not impressive to only use the AUROC as the performance metric. (ii) There are so many existing link prediction methods. We suggest that the authors should compare their methods with some state-of-the-art link prediction methods. (iii) Since the AUROC of DNL is close to that of the node2vec, the authors need to report an average AUROC in many independent tests. (iv) There are many other link prediction methods that can be used in a unsupervised manner. So this is not a big advantage of the presented method.

Minor: Panel labels in Fig.3 are missing.

Reviewer #2 (Remarks to the Author):

This is an important paper that is clearly state-of-the-art in the exciting field of network analysis and network reconstruction techniques. The authors study takes advantage of complicated mathematical and computational tools of matrix factorization, image dictionary learning, graph embeddings, motif sampling techniques, advanced algorithmic developments, and machine learning. This goes beyond standard computational tools, and the authors use modern sophisticated MCMC techniques and provide theoretical guarantees for some of their key algorithms. They do not just use existing tools but develop their own, displaying substantive novel and technical developments. Even the motif sampling k-walk was quite cool. The paper studies various social networks (including some large-scale Facebook networks, as well as smaller sized), biological networks of Coronavirus and human protein networks, plus relevant networks ER, WS and BA that are essential benchmarks for such studies. The algorithms for network reconstruction under random edge deletion and addition of edges (denoising) appear to be very successful, and more accurate than other approaches to date, as the authors have taken pains to demonstrate. Altogether the paper reads as a tour-de-force of what should and could be done in deciphering complex networks of any type or size, up to the present limits of scientific knowledge. It is clearly the result of a huge amount of tedious work combined with a high-level knowledge of network science.

There are, however, several issues of concern that need to be dealt with.

1. It came to my attention at the very last minute that many of the concepts in the manuscript appear in earlier papers of the authors, especially the JML article Lyu, et al. 2020 "Online matrix factorization for Markovian data and applications to network dictionary learning". In: Journal of Machine Learning (JML) Research 21 (2020), pp. 1–49. The authors have of course cited their paper and been very open about this.

I had initially thought to ask the authors to demonstrate their method by providing a reconstruction of the Escher image. But to my surprise, they have actually done this already in their published JML paper. Still a cleverly thought out visualisation of a reconstruction could look good in this submission.

Clearly the present submission is a far more general and friendly version of JML article that Nature-Communication readers should appreciate, and devoid of the many technical complications found in the

JML piece. However, the authors need provide a strong justification as to why their submission should appear in Nat. Comm. if a major proportion of the ideas are already published, and a only repackaged in a more palatable form here, similar figures and all.

The authors need to identify and point out to Nature Communications editors and reviewers, what are the new contributions in this submission and how it differs from the JML paper. I notice the authors have done so in one or two places in the SI, but there is so much material it becomes difficult to assess this point.

2. There is some ambiguity as to whether and when the W motif matrix is constructed from G_{true} or from the observed network $G_{observed}$. This needs to be clarified (see below).

3. Some of the authors results border the miraculous and need further verification and explanation, if only to highlight further how remarkable their work is.

For example, when 50% of false edges were added to the Coronavirus network, their method was able to detect 90% of the 1232 false edges while misclassifying 10% of the 2463 true edges.

This might seem possible if the W motif matrix was constructed from G_{true} . (Was it?) But if so, that would seem unfair. On the other hand the authors may have constructed the W matrix from $G_{observed}$, in which case it is a remarkable result. So this needs to be clarified, and the authors need to provide more intuition as to why this level of accuracy is possible and ideally, demonstrate convincingly that their method is achieving this remarkable level of performance.

For example, readers could wonder whether the prediction achievement is a result of some sort of artefact in the index creating these good predictions, or some other possibility.

Thus the network denoising/reconstruction results need some sort of additional verification and explanation.

4. Also astonishing is that the authors ability to predict one network from motifs in another. In the extreme case, how is it possible to predict the Caltech network from a network dictionary based entirely on random ER networks. with >60% accuracy? If that is true, the authors should try and verify to the reader better that this is really happening, and explain in an intuitive way how a coin flip (ER edges) can

make such good predictions. And if that's the case, how could these motifs have meaning beyond their random structure, which would not on the face of it, explain this sort of accuracy.

Presumably there are simple answers to these questions, and maybe this reviewer has misunderstood the authors claims or setup. But the paper should be written in a way that will convince the reader without these sorts of questions being left hovering unresolved in the reader's mind.

5. The authors need to point out the differences and advantages of their method to the other related NDL methods of say Lee and Seung (Ref.5) and Mairal, for example, (as well as their own published work) that make their paper a major advance in the field.

Further remarks: In the Introduction, the authors emphasise some sort of connection with the very popular biological motifs, but the connection is a little tendentious. By the end of the paper the authors write: "In contrast to motifs [6], which have been used to capture common k-node subgraphs in a network, one can use weighted superpositions of latent motifs to build all such patterns. " It's not clear what the authors mean or whether they have a point here, as one can't use their latent motifs to explore the biological significance of anything the biological motif analyses were intended for, and vice versa. Moreover, it is not clear what the authors latent motifs tell us biologically and I would welcome the authors to suggest this as a future research direction.

The authors should briefly discuss the recent paper by Seshadhri S et al in PNAS (2020) The impossibility of low-rank representations for triangle-rich complex networks whose authors claim to be relevant for many real-world networks. This should be compared to p.9 of Lyu et al. "Our computational experiments.... Illustrate that various" real world nets "have low-rank mesoscale structures."

p.4 last line WS2 WS1

p.6 peak above 82% for $\theta \approx 0.6$. The peak is almost 90%

Finally, the paper's theoretical algorithmic developments and guarantees may need to be checked by a theoretician (computer scientist), although the bottom-line is that the algorithms appear to work well.

Reviewer #3 (Remarks to the Author):

LEARNING LOW-RANK LATENT MESOSCALE STRUCTURES IN NETWORKS

Single blind review of paper submitted to Nature Communications

SUMMARY AND POSITIONING OF THE WORK

The work presented in this paper sets out to detect so-called latent motifs in networks. These are patterns that are present at the meso level of a network; inbetween micro (the nodes and edges) and macro (the network as a whole). The meso level has, because of its use in better understanding complex network phenomena, been of great interest in the field of network science, where this paper is clearly positioned. The field itself is inherently interdisciplinary, as networks as a model can be useful in various domains, such as the social sciences (networks of people), biology (protein interactions) and science studies (collaboration networks). Overall, the paper is well-written, properly structured and adequately embedded within existing literature in network science.

One of the main promises of understanding meso level structure in networks, is that it can help understand what the building blocks of a network really are. Research thus far focused on finding these building blocks in the form of motifs: small subgraphs of a network that occur at (surprisingly) high rates. This paper seems to aim to advance this line of research by not only finding motifs (and interpreting their occurrence within some applied domain), but also showing how these motifs, the building blocks, can, after their discovery, be used for a number of relevant tasks, being 1) to compare networks based on their meso level structure, 2) reconstruct the original network from the meso level patterns, as well as 3) to solve downstream tasks such as predicting missing or spurious links in the network. This reuse of the discovered motifs is innovative.

METHODOLOGY

A big difference with respect to previous work is that the object of study is not just motifs in the form of frequent subgraphs, but so-called latent motifs. An original overall methodology, building on several previous works, is constructed that detects these latent motifs, ultimately returning a so-called dictionary of latent structures of the dataset. These are then used in the different experimental downstream tasks mentioned above. Below, two larger issues with respect to the methodology, in particular the intended interpretation of the latent motifs and the sampling approach, are sketched.

INTERPRETATION OF THE LATENT MOTIFS

It seems that the paper encounters a major difficulty in interpreting the “network dictionaries” obtained from NNMF as elementary subgraph patterns, i.e., latent motifs. The ordering of the indices of the induced subgraphs is strict according to the random walk used to sample it, and this is problematic for the interpretation of the low-rank approximations. Many of the latent motifs are difficult to interpret as an elementary subgraph pattern. Two examples:

In the WS networks, the blocks in the “network dictionary” reflect the number of steps the random walker took before traversing a re-wired link to a different area of the network. This has little (if any) relation to the characteristic size of communities in the network; indeed, WS is a modified lattice that

does not have distinct communities at all. In both the WS models used, all nodes remain densely connected with 90% or 95% of their original 50 (!) neighbors.

In the BA model, ordering the nodes in the induced subgraph by the progress along the random walk means that encountering a hub at step #6 creates a different matrix than encountering a hub at step #7. Or step #3. Or step #18. NMF sees these scenarios as distinct and so each would produce a different entry in the “network dictionary”. But a hub is a hub, so to interpret these as elementary subgraph patterns the index of that node should perhaps not play such a large role.

This leads to the general question of whether the matrices should be ordered according to the walk? Doing so is problematic in two ways: it affects the interpretation of several of the results and may introduce considerable amounts of “noise”. The authors should clarify that they are working with matrices included on k-chains, not k-node subgraphs.

SAMPLING

The effect of k-chain sampling on k-node subgraphs needs to be clarified and discussed. The first node of the k-node walks used to induce subgraphs is properly sampled, with solid mathematics behind the MCMC methodology. But it does not necessarily follow that you then have representative subgraphs. From Algorithm MP, it is clear that the walk used to select the remaining 20 nodes is a regular random walk. This would mean that in the subgraphs, nodes are going to be a part of these subgraphs proportional to their degree. That the nodes within the k-node subgraphs (just not the initial node) are sampled in this way may affect the downstream tasks. It is unclear how this affects for example the link prediction task.

A greater point of concern is that the walks used to select the k-node subgraphs are allowed to backtrack and loop. At various points in the paper it is claimed that $k = 21$ node subgraphs are what is studied. But in fact it is subgraphs induced from $k = 21$ node walks, which often also have fewer than 21 nodes in total. This is barely mentioned until deep in the supplementary material.

The solution to the issue of backtracks and loops, also, may not be sufficient. Specifically, the non-folding mask ensures that on-chain links do not appear multiple times in the resulting matrix, but also off-chain links can be duplicated. For instance, in the case of a backtrack along two edges of a triangle. Or a loop around a square. The possibility of duplicate off-chain links does not seem to be addressed and they would result in particular artefacts in the subgraph matrices. This would be a problem if they are consistent enough to be picked up on as “motifs” by NMF. For instance: the every-other pattern in Figure 2, Coronavirus, $k=11$: is this really a latent motif, or is it an artifact of backtracks? It is unclear which, if any, of the dominant latent motifs may be artifacts. If so, how does this influence the results?

EXPERIMENTS

The experiments conducted in the paper make use of some of the most common models used in network science, being the ER, BA and WS models. Moreover, several well-known and newer empirical network datasets from a range of domains (social, biological, etc.) are considered. The downstream tasks such as spurious/missing edge detection are compared to common existing approaches, such as graph embeddings. Again with the experiments, there are a number of concerns and questions that should be addressed.

One conclusion from the first set of experiments on the actual latent motifs is that using $k=21$ length walks, the Caltech dataset has communities with likely more than 6 nodes, and in the UCLA dataset this is unlikely (p. 3). Two things are a bit unclear here: First, is this really a finding about the underlying empirical data, or is it based on the chosen $k=21$ length walks? If so, can it be validated using for example a community detection algorithm, the de facto approach of the field for finding communities? Moreover, it is a bit unclear how this finding relates to the finding on p.7 where it is mentioned that 'Caltech has a lot more communities of size at least 10 than the other three universities'. This should be clarified.

The first set of experiments claims to "provide(s) a new lens on prior observations of such community structure" (p. 5), in particular drawing upon a few of the empirical networks and the WS models. The WS model is known to generate networks with a lot of local clustering, is that what is measured here as being community structure? In fact, in the WS experiments, a very dense network is created by initially connecting nodes to a large number of other nodes, and then rewiring with relatively low probability. To what extent is this really community structure, and to what extent is it just a slightly distorted dense network? Would a benchmark dataset known for having more realistic community structure, such as SBM or LFR, show the same results?

The first set of experiments is said to "reveal nodes that are adjacent to all other nodes". The word "reveal" here seems a bit misplaced as these hubs are the essence of the BA model which was used to generate the networks in which this pattern was revealed. This should at least be adequately rephrased.

The second set of experiments on network reconstruction aims to "reconstruct an entire network". The big problem here is the way in which it is validated whether the reconstruction was successful. This is done by taking the size of the intersection divided by that of the union of the original and reconstructed edge sets. This seems very crude, and appears to measure edge set overlap, and not network similarity. More common ways of assessing network similarity in the field, such as the similarity of the degree distribution or similarity of measures such as local clustering, average distance and/or diameter, are not reported on. This makes it hard to judge if the reconstructed networks are even remotely similar to the original networks, in terms of network structure.

Rather diverse accuracies of reconstruction (although the way of measuring that is perhaps problematic, see previous point) are reported on on page 6 / Figure 3. What makes that one empirical network is so

easy to reconstruct, and the other is not? Can any structural properties of these networks be derived that explain this?

The third set of experiments deals with predicting missing/spurious links, and the experimental setup is similar to that of link prediction, a well-known problem at the intersection of network analysis and machine learning. An important problem here is that the presented results in Figure 4f are subject to many many preconditions: there is intensive rebalancing of classes done to train a proper model (a common problem in this scenario due to the sparseness of the problem, and explained in the supplementary material). But moreover, a comparison is made with existing yet parameterized embedding methods. However, a detailed explanation of the exploration of the parameter space of these methods vs. the parameters of the NDL+NDR method, is not given. This makes the statement that it achieves state-of-the-art results difficult to validate.

The findings are all in all not extremely surprising or groundbreaking from a substantive perspective, and reconfirm either characteristics of the model that was used to generate the datasets, or reconfirm empirical findings about the considered networks that are already known about those types of networks. The conclusion of the paper is that the generality of the insights should be established, which is a valid one. Moreover, while clearly beyond the scope of the paper, slightly more attention could be given to the fact that in modern network science studies, the temporal, feature-rich and multi-layer aspect of a network plays a role. Existing methods for motif detection already incorporate these aspects, which could at least be reflected on to better position the work within the literature.

SMALLER POINTS:

p.2 weighted sum of small number -> a small number

use of ~ before `\ref{}` and `\cite{}` (rather than a space; occurs a few times)

on p.4, there is a "Section " without a `\ref{}` following it

on p.3, the sentence "We refer to the remaining entries as 'off-chain' entries; they represent the additional connection that may vary depending on the network structure" is a bit unclear; especially the part "the additional connection that may vary depending on..." is rather vague; what is the magnitude of this? In its current format, this paragraph raises more questions than it answers.

In the first set of experiments, the ER results are not really discussed, although their setup/parameters are mentioned.

It is interesting that ER shows higher than random accuracy in the second set of experiments, whereas it is a truly random graph. This warrants some explanation.

1. REVISION HIGHLIGHTS

Key changes in the revision:

(Key 1) [Detailed discussion of subgraph sampling.] The reviewers asked about (1) the interpretability of latent motifs (and in particular whether they truly represent mesoscale structures of networks) and also (2) how it is possible to reconstruct some networks (e.g., UCLA) with high accuracy by using latent motifs that one learns from a completely unrelated network (e.g., ER₁). In our previous manuscript, we neglected to discuss the actual subgraphs that we obtain using our motif-sampling algorithm and the fact that it is from these subgraphs that we learn latent motifs. To address these points (and to thereby address these questions from the reviewers), in the main text of the revised manuscript, we now give a detailed discussion of our k -path motif-sampling method. To help convey interpretability, we also show examples of actual subgraphs (see the new Figure 1 and the new Figure 11 in Methods) that we sample from various networks. This will help the readers to build intuition about the particular subgraph-based mesoscale structure that we investigate.

For example, k -node subgraphs that one samples from the sparse Erdős–Rényi network ER₁ tend to have k -node paths and a few extra off-chain edges (see Figure 1 in the main manuscript). A similarly sparse network, such as UCLA, has similar subgraph patterns, even if the off-chain edges are not independently random as they are in subgraphs of ER₁. This is why we can reconstruct some networks with high accuracy by using latent motifs that we learn from a completely unrelated network. We discuss this point in the sixth paragraph (starting with “We also comment briefly about the cross-reconstruction experiments.”) of Section “Network reconstruction using latent motifs” of the revised manuscript.

Additionally, note that the ‘mesoscale community structure’ that we observe in these sampled subgraphs is a different mesoscale feature from the familiar ‘global community structure’ from networks such as stochastic block models. We now comment on this in the revised manuscript.

(Key 2) [Improved Interpretability of latent motifs.]

We now elaborate on the use of k -path sampling in our new network dictionary learning (NDL) algorithm to ensure interpretability of the network dictionary elements as latent motifs. The previous NDL algorithm (both the original one from [10] and the improved one from our initial submission) had an issue with respect to the interpretation of latent motifs as elementary k -node subgraphs. This is due to the fact that a $k \times k$ binary matrix that is induced by a uniformly randomly sampled k -walk during dictionary learning does not always correspond to the adjacency matrix of an actual k -node subgraph in a network, because the k -walk may have fewer than k distinct nodes (e.g., it can alternate between two adjacent nodes). To resolve this issue in the revised manuscript, during the dictionary-learning phase of our NDL algorithm, we now combine Markov-chain Monte Carlo k -walk sampling with rejection sampling so that we use only k -walks with k distinct nodes (i.e., we use k -paths). Consequently, we now learn the k -node latent motifs only from $k \times k$ adjacency matrices that correspond to k -node subgraphs of a network. This guarantees that any network structure (e.g., large-degree nodes, communities, and so on) in the latent motifs must also exist in the network at scale k .

In the revised manuscript, because of this change in our sampling approach, we have revised our theoretical analysis to our modified NDL algorithm and have proven convergence results for this modified algorithm. This modification improves the interpretability of the latent motifs (e.g., see the new results for the network **CORONAVIRUS PPI**), as we illustrate in a new experiment. (See Figure 11 in the Methods section of the manuscript.) Accordingly, we have changed all of the figures that used the previous version of the NDL algorithm. This includes all figures that report results of network-reconstruction and network-denoising experiments. Specifically, we have revised Figures 2, 4, 5, and 9 (in the main manuscript), 11 (in the Methods section of the main manuscript), and 12 and 14–21 (in the Supplementary Information) In the revised manuscript, we show the latent motifs as weighted networks, rather than as weighted adjacency matrices. This helps convey the structures of the latent motifs and connects them directly with subgraph samples. (For example, see Figure 1 in the main manuscript.)

(Key 3) [New theoretical bound on the network-reconstruction error] In the revised manuscript, we establish a novel result about the relationship between the global reconstruction error and an approximation error of mesoscale patches by latent motifs, with an explicit dependence on the number of nodes in subgraphs at a chosen mesoscale. We state this result in Theorem G.10(iii) in our Supplementary Information. Informally, Theorem G.10(iii) states that one can accurately reconstruct a network if one has a dictionary of latent motifs that can accurately approximate a network’s mesoscale patches. See Eq. (33) in Theorem G.10(iii) in our SI.

(Key 4) [Explanation of high cross-reconstruction accuracy] In Figure 10 in the Methods section of the revised manuscript, we include an instructive example of image reconstruction, which includes reconstructing an image from randomized dictionary. This new example helps illustrate that reconstruction depends both on the original object and on the dictionary, so even when a dictionary is uninformative, one can still recover some structure of the original object.

As we have noted in key change (Key 1) above, sparse networks tend to have similar subgraph patterns that are induced by k -paths. Specifically, they consist of a

k-path that is used to sample those subgraphs and a few additional off-chain edges. (See the examples in Figure 1 in the manuscript.) Therefore, for sparse networks, we tend to obtain a very high reconstruction accuracy even if the latent motifs are learned from a different sparse network from the one that is being reconstructed, as we see in cross-reconstruction accuracy for the network **UCLA** in Figure 5 in the main manuscript.

(Key 5) [Revised network-denoising experiments.] We have thoroughly revised our network-denoising experiments in the revised manuscript. We have implemented several recent network-embedding methods using **DEEPWALK** and **NODE2VEC** and several more traditional approaches using **JACCARD INDEX**, **PREFERENTIAL ATTACHMENT**, the **ADAMIC-ADAR INDEX**, and **SPECTRAL EMBEDDING**. (We give implementation details in Section F.9 of the Supplementary Information.) We now compare our network-denoising method to all of these methods in three different settings: (1) we add 1,000 (except 30,000 for **H. SAPIENS**) false edges that we randomly generated using a Watts–Strogatz model; (2) we add 50% (of the number of original edges) of false edges that we randomly generated using an Erdős–Rényi model; (3) we remove 50% of uniformly randomly chosen edges while maintaining the fact that our network consists of one connected component. As suggested by the reviewers, we now report the mean of five independent runs of each experiment. The experiment in (3) is identical to the one in [Grover and Leskovec, 2016]. The implementation of and comparison against the benchmark methods in (2) are new to the revised manuscript and experiment setting (1) is entirely new to the revised manuscript. We used settings (2) and (3) in the original manuscript, and we have added setting (1) in the revision. We report the result in terms of area under the curve (AUC) in the main manuscript (see Figure 9). As requested by the first reviewer, we also provide results in all three settings in terms of accuracy, precision, and recall in the Supplementary Information.

The reviewers suggested that we need to explain why our method achieves such a high accuracy for some cases, such as the network **CORONAVIRUS** with the addition of 50% false ER edges. Our new experiments indicate that network denoising for settings (2) and (3) are not particularly difficult tasks because some of the existing benchmark methods (specifically, **JACCARD INDEX**, **PREFERENTIAL ATTACHMENT**, **ADAMIC-ADAR INDEX**, and **SPECTRAL EMBEDDING**) achieve high accuracy on them.

For instance, the high performance of **PREFERENTIAL ATTACHMENT** for **CORONAVIRUS** with noise type +ER implies that the product of the degrees between the nodes that are attached to a true edge is likely to be larger than the product of the degrees of nodes that are attached to false edges. For example, consider a star network with one central node and five leave nodes. In this network, a uniformly randomly chosen non-edge always consists of two degree-1 nodes, but a uniformly randomly chosen edge always has one degree-5 node and one degree-1 node. As in this star example, the real-world networks that we consider in our paper are sparse (with a maximum density of at most 0.057 for **CALTECH**, which is our densest network). One can choose a non-edge uniformly at random from such sparse networks approximately by choosing two nodes uniformly at random. However, as in the star example, uniformly randomly choosing an edge may lead to a bias that favors choosing at least one large-degree node.

From our results, we see that our method has the best performance of all tested methods in setting (1). The difference in performance gain in terms of AUC against the second-best method is as large as 35% (for the network **CORONAVIRUS**). Our

intuition behind the better performance of our approach is that false edges that are generated by a small WS network have a distinctive mesoscale structure, and it can thus be detected more effectively using our approach.

(Key 6) [Explanation of our high denoising accuracy] In our revised manuscript, we report a new phenomenon that contributes to the high reconstruction accuracy of our method. (See the new experiments in Figure 8 in the main manuscript.) Namely, when a real-world network has false edges, the Markov-chain Monte Carlo (MCMC) sampling algorithm (which incrementally evolves a specified k -walk) that we use throughout the denoising process tends to connect false edges using a smaller number of edges than it uses to connect true edges. Therefore, false edges are more likely to be off-chain edges than on-chain edges when we sample a k -walk by evolving our MCMC algorithm. (See the histogram in Figure 8a in the main manuscript.) Consequently, of the edges between nodes in a uniformly sampled k -walk in an additively corrupted network, false edges are more likely to appear as on-chain edges than as off-chain edges. When we consider subtractive noise, an analogous observation holds for true non-edges and false non-edges.

We elaborate on the above point in the section on “Network denoising using latent motifs” in the main manuscript.

2. ITEMIZED RESPONSES TO REVIEWER 1

2.1. Detailed responses.

1. Summary: In the manuscript entitled Learning Low-Rank latent mesoscale structures in networks, the authors proposed a new approach for describing low-rank mesoscale structures in networks. They claimed this approach can be applied in a wide range of network-analysis tasks. We have several comments on the current version of this manuscript.

Response: Thank you for your comments.

2. The key component of the proposed approach is the so-called network dictionary learning (NDL). Yet, how do these learned latent motifs reflect the micro/macro structures in the original networks, especially for the latent motifs with different mesoscales/walk length k ? For example, the latent motif of the WattsStrogatz (WS) network shows two modules at $k=11$. But the original WS network is not supposed to have such communities or modules. How do the authors reconcile this contradiction?

Response: This is not a contradiction, as we now explain explicitly in the revised manuscript. See key changes (Key 1) and (Key 2) in Section 1.

The WS networks have locally densely connected nodes on a ring of 5,000 nodes and also have randomly added shortcut edges across the ring. Therefore, when one samples a k -walk uniformly at random, it is unlikely that the walk does not use any shortcut edge. When a k -walk uses a shortcut edge, we expect the resulting sampled induced subgraph to have two distinct densely connected communities.

For example, with $k = 6$, the walk length is short enough that the associated most-dominant latent motifs consist of a single small community. As one increases k to 11 and beyond, we start to see multiple communities (for the reason that we just indicated above). This type of ‘module’ is not what one expects from a WS network when one looks at it globally, because its associated adjacent matrix doesn’t have an associated block structure (as Reviewer 1 correctly pointed out). Importantly, however, latent motifs reflect a particular mesoscale structure of networks (that is, subgraph structures that are induced on k -paths) of a very different type than standard block structure, and such a mesoscale structure does not have to be the same as what one observes in network structure.

3. To fully support the authors statement, we suggest a detailed comparison between latent motifs with the micro/macro structures in the original networks. Otherwise, how can we make sure this approach truly extracts the real motifs in networks, considering that (i) the latent motifs for BA networks only have one hub at $k=21$. (ii) the social networks: UCLA, MIT, and Harvard do not have community structures.

Response: It seems that the reviewer’s comment is based on comparing the second most-dominant latent motif in Figure 2 in the initial submission to some known mesoscale structure (so-called ‘real motifs’) of networks. While a single latent motif in a network dictionary (the full set of r latent motifs) may reflect some known mesoscale structure of networks to some extent, it does not have to align with any ‘real motif’ (such as a clique, a community, or something else) in the structure of a network. The full set of r latent motifs represent ‘latent structure’ of k -node sub-graphs. Analogously, the top r principal components may be able to approximate a data point by their linear span, but any single principal component will miss some features.

In this regard, we address the reviewer’s comment as follows: (i) Latent motifs with single hubs (i.e., nodes that are adjacent to many other nodes in the same latent motif)

at different locations can represent k -node subgraphs with multiple hubs (i.e., nodes that are adjacent to many other nodes in the same subgraph) by taking a nonnegative linear combination of them (analogous to taking linear combinations of principal components); and (2) several of the latent motifs in the full network dictionary (see Figure 14 in the revised manuscript) do have community structure. For instance, the entire set of $r = 25$ latent motifs of the networks `UCLA` and `CALTECH` in Figure 2**b,c** in the revised manuscript do show community structure. In our revised manuscript, we have also added a new experiment (see Figure 3 in the main manuscript) that compares community sizes in 10,000 network subgraphs with those in $r = 25$ latent motifs. In most cases, they show excellent agreement.

4. The authors evaluated the method in two network reconstruction scenarios via latent motifs. First, if we know the network exactly and the dictionary W is quite efficient, then the reconstructed graph should be very similar to the original graph, in the sense that the accuracy is much higher in a broad range of thresholds. Second, it's not a big surprise that the peaks of accuracy are around 0.3 to 0.6, since the number of reconstructed edges for small (or large) thresholds is too small (or large), respectively. Why didn't the authors just calculate the Accuracy? Third, how to explain the result that the performance of reconstructing `UCLA` using ER network can still achieve 90% accuracy?

Response: Thank you for raising these important points.

Point 1: We agree with this point. In our original submission, this reasonable claim was not justified rigorously. In the revised manuscript, we rigorously derive a quantitative version of this observation. See key point (Key 3) in Section 1.

Point 2: We believe that the reviewer is suggesting that we compared weighted reconstructed networks directly with unweighted original networks. In the revised manuscript, we define a notion of 'Jaccard reconstruction error' between a weighted reconstructed network and an unweighted original network (see Section G.3 of the Supplementary Information) and prove (in Theorem G.10 of the SI) that this quantity is upper bounded by $E[\text{loss}]/2(k - 1)$, where k denotes the number of nodes in subgraphs that correspond to a chosen mesoscale and 'loss' measures the approximation error of mesoscale patches of a network by the latent motifs that we use for reconstruction. See key point (Key 3) in the revision highlights. We illustrate and computationally verify this new result with a variety of examples. (See Figure 6 in the main manuscript.)

Point 3: In the main manuscript, we now comment on the cross-reconstruction experiments in Figure 5 that use latent motifs that we learn from ER networks. For instance, when reconstructing `MIT`, `HARVARD`, and `UCLA` using latent motifs that we learn from `ER2`, we obtain reconstruction accuracies of at least 72%. This may seem unreasonable at first glance because the latent motifs that we learn from `ER2` should not have any information about the Facebook networks. However, all of these networks are sparse (with edge densities of at most 0.02) and we are sampling subgraphs using k -paths. The k -node subgraphs that are induced by uniformly random k -paths in these sparse networks have only a few off-chain edges. (See Figure 1 in the main manuscript.) For example, the k -node subgraphs that we sample from the sparse ER network `ER2` tend to have k -node paths and a few extra off-chain edges. (See Figure 1 in the main manuscript.) A similarly sparse or sparser network, such as `UCLA` (which has an edge density of about 0.0036) has similar subgraph patterns (despite the fact

that its off-chain edges are not independent, unlike the subgraphs of ER_2). This is the reason that we can reconstruct some networks with high accuracy by using latent motifs that we learn from a completely unrelated network.

We also added new results (and figures to demonstrate them) to help address Point 3. We proved a new theoretical result about a Jaccard reconstruction error bound (see key point (Key 3) in the revision highlights) and presented new experiments in Figure 6 in the main manuscript. As a corollary of our new result, we deduce that the expected Jaccard reconstruction accuracy of reconstructing an Erdős–Rényi random network with expected edge density p is at most $kp/2$ when the network dictionary consists of a single k -path. For instance, if $k = 20$ and $p = 0.01$, this upper bound for the expected Jaccard reconstruction error is 0.1. Therefore, the expected Jaccard reconstruction accuracy in this case is at least 0.9. See the section on “Network reconstruction” using latent motifs” in the main manuscript.

5. Regarding the application of their approach to network denoising. (i) As link prediction is typically dealing with a highly imbalanced dataset, its not impressive to only use the AUROC as the performance metric. (ii) There are so many existing link prediction methods. We suggest that the authors should compare their methods with some state-of-the-art link prediction methods. (iii) Since the AUROC of DNL is close to that of the node2vec, the authors need to report an average AUROC in many independent tests. (iv) There are many other link prediction methods that can be used in a unsupervised manner. So this is not a big advantage of the presented method.

Response:

- (a) We used AUC and ROC to do a direct comparison with NODE2VEC and DEEP-WALK results, as was done in [Grover and Leskovec, 2016]. In the revised manuscript, we now also report the performance in terms of accuracy, precision, and recall. See Figure 13 in the Supplementary Information.
- (b) In the revised manuscript, we directly implement several additional classical and recent network-denoising approaches and compare their performance with that of our new approach on multiple experimental runs. We give a detailed response to this in key change (Key 5) in Section 1.
- (c) We agree that the fact that our network-denoising method is unsupervised is not one of its main advantages, as classical denoising algorithms (such as JAC-CARD INDEX, THE ADAMIC–ADAR INDEX, and PREFERENTIAL ATTACHMENT) are also unsupervised and directly give a confidence score for each edge. The unsupervised nature of our method is only an advantage when compared to node-embedding methods (such as SPECTRAL EMBEDDING, DEEPWALK, and NODE2VEC), which output feature vectors for the edges and then need to subsequently use some other classification algorithm (e.g., logistic regression). In the revised manuscript, we have thoroughly revised the section on network denoising and have added new experiments and discussions. See the section on “Network denoising using latent motifs” in the main manuscript.

6. Minor: Panel labels in Fig.3 are missing.

Response: We fixed this. Thank you for pointing it out.

3. ITEMIZED RESPONSES TO REVIEWER 2

- 1. Summary:** This is an important paper that is clearly state-of-the-art in the exciting field of network analysis and network reconstruction techniques. The authors study

takes advantage of complicated mathematical and computational tools of matrix factorization, image dictionary learning, graph embeddings, motif sampling techniques, advanced algorithmic developments, and machine learning. This goes beyond standard computational tools, and the authors use modern sophisticated MCMC techniques and provide theoretical guarantees for some of their key algorithms. They do not just use existing tools but develop their own, displaying substantive novel and technical developments. Even the motif sampling k-walk was quite cool. The paper studies various social networks (including some large-scale Facebook networks, as well as smaller sized), biological networks of Coronavirus and human protein networks, plus relevant networks ER, WS, and BA that are essential benchmarks for such studies. The algorithms for network reconstruction under random edge deletion and addition of edges (denoising) appear to be very successful, and more accurate than other approaches to date, as the authors have taken pains to demonstrate. Altogether the paper reads as a tour-de-force of what should and could be done in deciphering complex networks of any type or size, up to the present limits of scientific knowledge. It is clearly the result of a huge amount of tedious work combined with a high-level knowledge of network science. There are, however, several issues of concern that need to be dealt with.

Response: Thank you for your many kind comments about the contributions of our paper. Thank you also for expressing your concerns, which we have worked hard to address in the revised manuscript.

2. It came to my attention at the very last minute that many of the concepts in the manuscript appear in earlier papers of the authors, especially the JML article Lyu, et al. 2020 Online matrix factorization for Markovian data and applications to network dictionary learning. In: Journal of Machine Learning (JML) Research 21 (2020), pp. 149. The authors have of course cited their paper and been very open about this.

I had initially thought to ask the authors to demonstrate their method by providing a reconstruction of the Escher image. But to my surprise, they have actually done this already in their published JML paper. Still a cleverly thought out visualization of a reconstruction could look good in this submission.

Response: The image reconstruction example in the JMLR paper was to demonstrate a different point: that independent and identically distributed (IID) sampling and Markovian sampling of image patches can both result in an accurate reconstruction.

In the revised manuscript, we now include a more instructive example of image reconstruction. This example includes reconstructing an image from a completely randomized dictionary. (See Figure 10 in the Methods section of the main manuscript.) This example illustrates that reconstruction depends both on the original object and on the dictionary, so even if a dictionary is uninformative, one can still recover some structure of the original object. Also see key point (Key 4) in Section 1.

Clearly the present submission is a far more general and friendly version of JML article that Nature-Communication readers should appreciate, and devoid of the many technical complications found in the JML piece. However, the authors need provide a strong justification as to why their submission should appear in Nat. Comm. if a major proportion of the ideas are already published, and a only repackaged in a more palatable form here, similar figures and all.

The authors need to identify and point out to Nature Communications editors and reviewers, what are the new contributions in this submission and how it differs from

the JML paper. I notice the authors have done so in one or two places in the SI, but there is so much material it becomes difficult to assess this point.

Response: Thank you very much for raising this concern. We emphasize that the current work under consideration for publication in *Nature Communications* is not just a simple repackaging of our previous work published in JMLR [10], and we agree that it is important to openly allay any potential concerns of our manuscript’s readers. While we did report some preliminary ideas of NDL and NDR in [10], there are a number of major advances that we report in this work. In the revised manuscript, we now include a subsection in Methods section that compares the main contribution of our work to the preliminary approach that was proposed in [10]. See Table 1 and the discussion in the subsection of Methods with the heading “Comparison of our work with the prior research in [10]”). Furthermore, we also now comment in the introduction of the main manuscript that “We give a **thorough** comparison of the present work and the prior work [10] in Methods as well as in our SI.”.

The most significant theoretical advance in the present paper, which is a new result in the revised manuscript, concerns the relationship between global reconstruction error and an approximation error of mesoscale patches by latent motifs. This new result includes an explicit dependence on the number of nodes in subgraphs at a chosen mesoscale. We state this result as Theorem G.10(iii) in the Supplementary Information. See key point (Key 3) in Section 1.

3. There is some ambiguity as to whether and when the W motif matrix is constructed from G_{true} or from the observed network $G_{observed}$. This needs to be clarified (see below).

Some of the authors results border the miraculous and need further verification and explanation, if only to highlight further how remarkable their work is.

For example, when 50% of false edges were added to the Coronavirus network, their method was able to detect 90% of the 1232 false edges while misclassifying 10% of the 2463 true edges. This might seem possible if the W motif matrix was constructed from G_{true} . (Was it?) But if so, that would seem unfair. On the other hand the authors may have constructed the W matrix from $G_{observed}$, in which case it is a remarkable result. So this needs to be clarified, and the authors need to provide more intuition as to why this level of accuracy is possible and ideally, demonstrate convincingly that their method is achieving this remarkable level of performance.

For example, readers could wonder whether the prediction achievement is a result of some sort of artifact in the index creating these good predictions, or some other possibility.

Thus the network denoising/reconstruction results need some sort of additional verification and explanation.

Response: We confirm that the network dictionary W was learned from $G_{observed}$ in all cases. In the revised manuscript, we thoroughly revised both our network-denoising experiments and our associated discussions. We responded to the reviewer’s comments in this item in key points (Key 5) and (Key 6) in Section 1.

In point (Key 5), we discuss that our previous result on denoising CoRoNAvIRus with 50% false edges added uniformly at random is in fact not a difficult problem, as we show that simple benchmark methods for network denoising (e.g., one that uses the product of the degrees of two nodes in an edge as a confidence score for classifying that edge as positive) performs very well. As an illustrative example, consider a star network with one central node and five leaf nodes. In this network,

a uniformly randomly chosen non-edge always consists of two degree-1 nodes, but a uniformly randomly chosen edge always has one degree-5 node and one degree-1 node. Therefore, the two nodes in a false edge always have degree 1. We demonstrate the effectiveness of our method in a new network denoising setting, where the false edges are not added uniformly at random, but instead in a structured way according to the Watts–Strogatz (WS) model. Our intuition behind the better performance of our approach in this situation is that false edges that are generated by a small WS network have a distinctive mesoscale structure, which we can detect effectively using our approach.

In point (Key 6), we report a new phenomenon that contributes to the high reconstruction accuracy of our method. (See the new experiments in Figure 8 in the main manuscript.) Namely, when a real-world network has false edges, the Markov-chain Monte Carlo (MCMC) sampling algorithm (which incrementally evolves a specified k -walk) that we use throughout the denoising process tends to connect false edges using a smaller number of edges than it uses to connect true edges. Therefore, false edges are more likely to be off-chain edges than on-chain edges when we sample a k -walk by evolving our MCMC algorithm. See the histogram in Figure 8a in the main manuscript.

4. Also astonishing is that the authors ability to predict one network from motifs in another. In the extreme case, how is it possible to predict the Caltech network from a network dictionary based entirely on random ER networks. with $> 60\%$ accuracy? If that is true, the authors should try and verify to the reader better that this is really happening, and explain in an intuitive way how a coin flip (ER edges) can make such good predictions. And if thats the case, how could these motifs have meaning beyond their random structure, which would not on the face of it, explain this sort of accuracy. Presumably there are simple answers to these questions, and maybe this reviewer has misunderstood the authors claims or setup. But the paper should be written in a way that will convince the reader without these sorts of questions being left hovering unresolved in the readers mind.

Response: We appreciate this comment, and we agree that this needs to be conveyed to the readers of our papers. Indeed, there is a simple explanation that various real-world networks can be reconstructed with decent accuracy using latent motifs that we learn from completely unrelated networks (including statistically unstructured ones, such as ER networks). We gave a detailed response to this point in key changes (Key 1) and (Key 4) in Section 1. The short answer is that for very sparse networks, regardless of the nature of those networks (a social network, a randomly-generated network, or something else), the subgraphs that are induced by k -paths tend to have similar structures. Namely, when a network is very sparse, conditional on sampling a sufficiently long k -path, the chance of observing any off-chain edges is very small. Therefore, the sampled subgraphs look like they have one long path and a few additional edges. For ER networks, the on-chain edges in a sampled k -path are *not* random, whereas the additional off-chain edges are random. It is these additional edges that reflect the subgraph structure of a particular network, but the mismatch in their pattern has little effect on the overall reconstruction accuracy for sparse networks, which do not have many off-chain edges in sampled subgraphs. For example, this is the case for UCLA.

Additionally, it is important to realize that one can successfully reconstruct some basic features of networks using a completely random dictionary. To illustrate this,

we have added an image-reconstruction example that uses a completely randomized dictionary to the revised manuscript. (See Figure 10 in the Methods section of our revised manuscript.) This example also illustrates that reconstruction depends both on the original object and on the dictionary. Therefore, even with an uninformative dictionary, one can still recover some structure of an original object. See also key change (Key 4) in Section 1, as well as the example of denoising CALTECH with additive noise from a randomly generated dictionary in which we draw each entry of each latent motif's $k \times k$ weighted adjacency matrix independently and uniformly from $[0, 1]$. See Figure 8 in the main manuscript.

5. The authors need to point out the differences and advantages of their method to the other related NDL methods of say Lee and Seung (Ref.5) and Mairal, for example, (as well as their own published work) that make their paper a major advance in the field.

Response: Thank you very much for this comment. We have significantly revised our manuscript in order to best respond to this comment. In the revision, we now include a subsection in Methods that compares the main contribution of the current work in comparison to a preliminary approach that was proposed in the prior work [10] (see Table 1 as well as the discussion in the section of Methods with heading “Comparison of our work with the prior research in [10]”).

We comment further on how our NDL method compares to other nonnegative matrix factorization (NMF) methods in the literature. The NMF methods in [2] and [12] are generic dictionary-learning algorithms that are not related to network analysis. A primary contribution of [10] was to apply such dictionary-learning algorithms (which are usually applied to image data and text data) to network subgraph samples by using a k -walk motif-sampling MCMC algorithm and to establish convergence of such an algorithm despite the Markovian dependence that occurs during data sampling. The preliminary method in [10] had issues in interpreting latent motifs because of an artifact in their sampling process and did not have any theoretical analysis for network denoising and reconstruction (NDR) algorithms. In the present work, we address all of these issues and also conduct a new theoretical analysis for network reconstruction. (See key point (Key 3) in Section 1.) Furthermore, we make a new scientific claim that many real-world networks have such low-rank subgraph mesoscale structure, and we support this claim with extensive computational experiments.

6. Further remarks: In the Introduction, the authors emphasise some sort of connection with the very popular biological motifs, but the connection is a little tendentious. By the end of the paper the authors write: In contrast to motifs [6], which have been used to capture common k -node subgraphs in a network, one can use weighted superpositions of latent motifs to build all such patterns. Its not clear what the authors mean or whether they have a point here, as one cant use their latent motifs to explore the biological significance of anything the biological motif analyses were intended for, and vice versa. Moreover, it is not clear what the authors latent motifs tell us biologically and I would welcome the authors to suggest this as a future research direction.

Response: Thank you for this comment. Our intention is only to compare the concept of our latent motif to the traditional network motifs (which are common in studies of biological networks). We do not intend to claim that one can use our latent motifs to directly infer network motifs. To clarify our point, we revised the Conclusions of our manuscript so that we say “ In contrast to motifs [3], which are

used frequently to capture overrepresented k -node subgraphs in a network, nonnegative linear combinations of our latent motifs approximate k -node subgraphs that are induced by uniformly random k -paths in a network.”

7. The authors should briefly discuss the recent paper by Seshadhri S et al in PNAS (2020) The impossibility of low-rank representations for triangle-rich complex networks whose authors claim to be relevant for many real-world networks. This should be compared to p.9 of Lyu et al. “Our computational experiments. Illustrate that various real world nets have low-rank mesoscale structures.” p.4 last line WS2 WS1 p.6 peak above 82% for theta 0.6. The peak is almost 90%.

Response: Thank you for this excellent comment. We added a discussion in the introduction of the main text on this point. The main impossibility result in [11] assumes a null model in the form of a random dot-product graph (RDPG), in which each node is represented as a feature vector in a Euclidean space and each node pair has an incident edge that is chosen independently at random with a probability that is proportional to the dot product of the feature vectors of the two nodes in the pair. The main result in [11] states that, for n -node graphs with roughly cn triangles that consist only of nodes with a degree of at most c' , one finds that as n grows, one cannot model the associated sequence of graphs by an RDPG in which the embedding dimension of each node is sublinear in n .

In stark contrast, in our paper, we analyze the space of k -node subgraph adjacency matrices and claim that various real-world networks have low-rank mesoscale structures that involve k -node adjacency matrices in the sense that one can have a few latent motifs and still can reconstruct a network with high accuracy. Our analysis does not assume any null model, and in particular it does not assume that one has a RDPG. Therefore, our claim of low-rankness of mesoscale structures in real-world networks does not contradict the main result in [11].

In the introduction of the revised manuscript, we write: “Our claim of the low-rank nature of a network’s mesoscale structure concerns the space of certain subgraph patterns, rather than the embedding of an entire network into a low-dimensional Euclidean space (as considered in spectral-embedding and graph-embedding methods [4, 5]). It is impossible to obtain such a low-dimensional graph embedding for networks with small mean degrees and large clustering coefficients [11].”.

For the last point, we believe that the reviewer meant arXiv and SNAP FB instead of WS1 and WS2. This is a correct point in our previous submission. However, in the revision, we have revised Figure 5 for our network-reconstruction experiments so that it shows results from the new NDL algorithm, which uses k -path sampling instead of k -walk sampling. (See key point (Key 1) in Section 1.) These results in the revised manuscript are thus slightly different than the corresponding results in our original submission. In the revision, we now state “... the accuracies for arXiv and SNAP FB peak above 88% and 70%, respectively, for $\theta \approx 0.6$.” See the third paragraph of the section in the main manuscript with the heading “Network reconstruction using latent motifs”.

8. Finally, the papers theoretical algorithmic developments and guarantees may need to be checked by a theoretician (computer scientist), although the bottom-line is that the algorithms appear to work well.

Response: We have carefully checked our theoretical analysis.

4. ITEMIZED RESPONSES TO REVIEWER 3

1. **SUMMARY AND POSITIONING OF THE WORK:** The work presented in this paper sets out to detect so-called latent motifs in networks. These are patterns that are present at the meso level of a network; in between micro (the nodes and edges) and macro (the network as a whole). The meso level has, because of its use in better understanding complex network phenomena, been of great interest in the field of network science, where this paper is clearly positioned. The field itself is inherently interdisciplinary, as networks as a model can be useful in various domains, such as the social sciences (networks of people), biology (protein interactions), and science studies (collaboration networks). Overall, the paper is well-written, properly structured and adequately embedded within existing literature in network science.

Response: Thank you for the nice summary of our work and for your overall positive conclusion about our paper.

One of the main promises of understanding meso level structure in networks, is that it can help understand what the building blocks of a network really are. Research thus far focused on finding these building blocks in the form of motifs: small subgraphs of a network that occur at (surprisingly) high rates. This paper seems to aim to advance this line of research by not only finding motifs (and interpreting their occurrence within some applied domain), but also showing how these motifs, the building blocks, can, after their discovery, be used for a number of relevant tasks, being 1) to compare networks based on their meso level structure, 2) reconstruct the original network from the meso level patterns, as well as 3) to solve downstream tasks such as predicting missing or spurious links in the network. This reuse of the discovered motifs is innovative.

Response: Thank you for your kind comment about our innovative use of motifs.

2. **METHODOLOGY:** A big difference with respect to previous work is that the object of study is not just motifs in the form of frequent subgraphs, but so-called latent motifs. An original overall methodology, building on several previous works, is constructed that detects these latent motifs, ultimately returning a so-called dictionary of latent structures of the dataset. These are then used in the different experimental downstream tasks mentioned above. Below, two larger issues with respect to the methodology, in particular the intended interpretation of the latent motifs and the sampling approach, are sketched.

Response: Thank you for this summary. We respond to the larger issues below.

3. **INTERPRETATION OF THE LATENT MOTIFS:** It seems that the paper encounters a major difficulty in interpreting the network dictionaries obtained from NNMF as elementary subgraph patterns, i.e., latent motifs. The ordering of the indices of the induced subgraphs is strict according to the random walk used to sample it, and this is problematic for the interpretation of the low-rank approximations. Many of the latent motifs are difficult to interpret as an elementary subgraph pattern. Two examples: In the WS networks, the blocks in the network dictionary reflect the number of steps the random walker took before traversing a re-wired link to a different area of the network. This has little (if any) relation to the characteristic size of communities in the network; indeed, WS is a modified lattice that does not have distinct communities at all. In both the WS models used, all nodes remain densely connected with 90% or 95% of their original 50 (!) neighbors.

Response: Thank you for this comment. In the main text of the revised manuscript, we now provide examples and an extended discussion of our motif-sampling

algorithm and subgraph samples from various networks. See the first section of the main manuscript and the examples in Figure 1. (See also key points (Key 1) in Section 1.) In our revised discussion in the manuscript, we explicitly illustrate the subgraph patterns that are induced by k -paths.

We also note that latent motifs (1) are supposed to capture the structure of such subgraphs as the mesoscale structure of a network and that (2) such mesoscale structures can be rather different from global mesoscale 'block' structures (e.g., in the form of densely-connected communities). In other words, the latent motifs convey elementary subgraph patterns through the perspective of a randomly sampled k -path. From this perspective, it is natural to regard the two-block structure that is captured in some latent motifs as elementary subgraph patterns at mesoscale k , because they convey that a network is composed of densely connected local communities that are connected directly by some shortcut edges. We revised the manuscript to clarify this issue, and in particular we point out that the type of community structure that we examine is very different from typical network community structure. See our detailed discussion in the third paragraph in the section "Latent motifs of networks at various mesoscales" in the main manuscript.

See key points (Key 1) and (Key 2) in Section 1.

In the BA model, ordering the nodes in the induced subgraph by the progress along the random walk means that encountering a hub at step #6 creates a different matrix than encountering a hub at step #7. Or step #3. Or step #18. NNMF sees these scenarios as distinct and so each would produce a different entry in the network dictionary. But a hub is a hub, so to interpret these as elementary subgraph patterns the index of that node should perhaps not play such a large role.

Response: We appreciate this concern from the reviewer.

The reviewer's concern is certainly valid when one learns a network dictionary from a small number of sampled subgraphs. However, it is not an issue when one learns a network dictionary from a large number of subgraphs. Our NDL algorithm is an iterative algorithm that is guaranteed to converge to the stationary points of the corresponding expected loss function asymptotically (see Theorems G.4 and G.7 in the Supplementary Information) when the number of iterations tends to infinity. Importantly, this objective function is independent of the particular samples that one encounters.

Suppose, for instance, that we have a single network G and have two sets $\{H_1, \dots, H_n\}$ and $\{F_1, \dots, F_m\}$ of k -node subgraphs, where both n and m are significantly smaller than the total number of all k -node subgraphs in G . Suppose that we sample these two sets from two disjoint parts of G , where one part is densely connected and the other part is not. One can learn a dictionary from each of these two sets, and they may capture distinctive subgraph structures in this scenario. However, by our proven theoretical guarantee, the discrepancy between these two dictionaries vanishes as we let n and m tend to infinity.

Now suppose that v is a hub node in G and that we have sampled a set $\{H_1, \dots, H_n\}$ of subgraphs. For each H_i , suppose that we first sample a k -path (according to one of our MCMC algorithms) and then take the subgraph that is induced by it. As the reviewer correctly pointed out, v will appear in a different location along the k -path with respect to the node ordering from traversing a walk. If n is small, then v may occur at different indices along the k -paths; this can then induce some bias on the location of a hub node in the latent motifs. However, as $n \rightarrow \infty$, this statistical error

will vanish, because due to the correctness of the MCMC sampling algorithm (see Proposition G.3 in the Supplementary Information), the uniformly random k -path converges to the uniform distribution over all k -paths in G , so v will appear in all possible locations in realizations of the k -path without sampling bias.

This leads to the general question of whether the matrices should be ordered according to the walk? Doing so is problematic in two ways: it affects the interpretation of several of the results and may introduce considerable amounts of noise. The authors should clarify that they are working with matrices included on k -chains, not k -node subgraphs.

Response: Thank you for bringing up this important point. We understand the concern of the reviewer that we are only choosing one node ordering (namely, the one that is given by k -chain sampling) and use only the corresponding $k \times k$ adjacency matrix to represent one ‘ k -node subgraph’ out of $k!$ possible ones. (Our previous k -chain sampling can return fewer than k distinct nodes; we discuss this point later in this response.) We first mention the relevant changes in our revised manuscript, and we then further discuss these points in this response.

First, as we elaborated in key points (Key 1) and (Key 2) in Section 1, in the revised manuscript, we only use k -paths to sample k -node subgraphs (with k distinct nodes), rather than k -walks, which can potentially use fewer than k nodes. Once we sample a k -node subgraph, the k -path that is sampled first is a Hamiltonian path of the k -node subgraph that is induced by the nodes of in the k -path, so it is natural to use the ordering from the Hamiltonian path to order the nodes in the subgraph. (Note that we do not use the term ‘Hamiltonian path’ in the revised manuscript.) With our use of k -paths instead of k -walks, all of the $k \times k$ matrices that we input into NDL algorithm correspond to real k -node subgraphs in a network. Therefore, one can regard the latent motifs that approximately generate those $k \times k$ matrices as elementary k -node subgraphs. Additionally, we regard a latent motif (which we compute initially as a weighted $k \times k$ matrix) as a weighted network with node set $\{1, \dots, k\}$ and edge weights that are given by the weighted adjacency matrix. In the revised manuscript, we represent all latent motifs as weighted networks in this way.

Second, we do not agree with the reviewer that using the node ordering that is induced by a k -path (previously a k -chain) sampling has the two potential issues that they pointed out. It is true that we are making some choices in our methodology and that one may thus suspect that some types of interpretability can be compromised. However, through extensive experiments (e.g., see Figures 1, 2, and 4 in the main manuscript), we find that (1) the structure that we observe in latent motifs also occurs in real k -node subgraphs and that (2) the community-size statistics in the latent motifs are in excellent agreement with those of many (specifically, 10^4) actual k -node subgraphs that are induced by uniformly randomly sampled k -paths in a network. See Figure 3 in the main manuscript.

We conducted multiple additional experiments with alternative node ordering (e.g., choosing one out of $k!$ node orderings uniformly at random and even looking at all $k!$ of them) and alternative subgraph sampling (e.g., repeatedly initializing random walks at the same node until collecting k distinct nodes). In all of these variants, we find that the learned latent motifs are much noisier and it is hard to see clear hub nodes (i.e., nodes that are adjacent to many nodes in the same motif) or community structure. We believe that using k -path ordering is likely a key choice to improve the interpretability of latent motifs. When one samples all subgraphs in a similar manner,

they all share the same Hamiltonian-path structure. The goal of dictionary learning from these subgraphs is to learn some typical patterns to subsequently use them to approximately generate all sampled subgraphs. Subgraphs from sparse networks have few off-chain edges; in most cases, there are fewer off-chain edges than on-chain edges. Therefore, by matching the Hamiltonian-path structure across all sampled subgraphs, one can learn a small number of generating patterns for the off-chain edges.

4. **SAMPLING:** The effect of k -chain sampling on k -node subgraphs needs to be clarified and discussed. The first node of the k -node walks used to induce subgraphs is properly sampled, with solid mathematics behind the MCMC methodology. But it does not necessarily follow that you then have representative subgraphs. From Algorithm MP, it is clear that the walk used to select the remaining 20 nodes is a regular random walk. This would mean that in the subgraphs, nodes are going to be a part of these subgraphs proportional to their degree. That the nodes within the k -node subgraphs (just not the initial node) are sampled in this way may affect the downstream tasks. It is unclear how this affects for example the link prediction task.

Response: Thank you for this comment. We presented three MCMC algorithms to sample a k -walk from a network with a desired target distribution (see Eq. (1) in the Supplementary Information (SI)), which is the uniform distribution over all possible k -walks when a network is symmetric and unweighted. We assume this throughout the main text; our theoretical analysis in the SI also considers symmetric and weighted networks. Two of the three MCMC algorithms — Algorithm MP (the pivot chain) with `AcceptProb = Exact` and Algorithm MG (the Glauber chain) — converge to this target distribution exponentially fast.

It seems that the reviewer has missed our particular use of the Metropolis–Hastings algorithm in Eq. (8) in Algorithm MP to correct the bias that may be introduced in the subsequent node sampling in Eq. (9). Namely, the particular choice of our acceptance probability in Eq. (9) in the SI involves probabilities of choosing the subsequent nodes, and the entire k -walk converges to the correct target distribution. (This is shown in [9, Thm. 5.7, 5.8].) To clarify this point in the revised manuscript, we have revised Appendix G.1 and added a new Proposition G.1 with a sketch of the relevant argument.

For the approximate pivot chain (see Algorithm MP with `AcceptProb = Approx`), we discussed how the use of the approximate acceptance probability in Eq. (9) of the SI affects the stationary distribution in detail at the end of Appendix C, with a rigorous justification of the convergence to the ‘tilted’ stationary distribution $\pi^{\square}_{F \rightarrow G}$ in Eq. (12) of the SI in Proposition G.2 (which was previously numbered as Prop. G.1 in our original manuscript).

Finally, we added Proposition G.3, which guarantees that our new k -path sampling algorithm (which consists of the usual MCMC k -walk sampling algorithms that we discussed above along with a rejection-sampling algorithm) converges to the correct target distribution (namely, the uniform distribution) as in Eq. (3) in the SI.

A greater point of concern is that the walks used to select the k -node subgraphs are allowed to backtrack and loop. At various points in the paper it is claimed that $k = 21$ node subgraphs are what is studied. But in fact it is subgraphs induced from $k = 21$ node walks, which often also have fewer than 21 nodes in total. This is barely mentioned until deep in the supplementary material.

The solution to the issue of backtracks and loops, also, may not be sufficient. Specifically, the non-folding mask ensures that on-chain links do not appear multiple

times in the resulting matrix, but also off-chain links can be duplicated. For instance, in the case of a backtrack along two edges of a triangle. Or a loop around a square. The possibility of duplicate off-chain links does not seem to be addressed and they would result in particular artefacts in the subgraph matrices. This would be a problem if they are consistent enough to be picked up on as motifs by NNMF. For instance: the every-other pattern in Figure 2, Coronavirus, $k=11$: is this really a latent motif, or is it an artifact of backtracks? It is unclear which, if any, of the dominant latent motifs may be artifacts. If so, how does this influence the results?

Response: Thank you for this important comment. We agree with the reviewer about the potential issues with allowing k -walks to backtrack, as they can substantially affect the interpretability of latent motifs. In the revised manuscript, we have resolved this issue by only using k -node walks with distinct nodes. Namely, instead of considering k -node subgraphs that are induced by any k -walk in a network, we only consider those that are induced by k -walks with distinct nodes. Consequently, the possible k -node subgraphs that we consider are exactly the ones that contain a Hamiltonian path. See key changes (Key 1) and (Key 2) in Section 1 for more details.

5. EXPERIMENTS: The experiments conducted in the paper make use of some of the most common models used in network science, being the ER, BA and WS models. Moreover, several well-known and newer empirical network datasets from a range of domains (social, biological, etc.) are considered. The downstream tasks such as spurious/missing edge detection are compared to common existing approaches, such as graph embeddings. Again with the experiments, there are a number of concerns and questions that should be addressed.

One conclusion from the first set of experiments on the actual latent motifs is that using $k=21$ length walks, the Caltech dataset has communities with likely more than 6 nodes, and in the UCLA dataset this is unlikely (p. 3). Two things are a bit unclear here: First, is this really a finding about the underlying empirical data, or is it based on the chosen $k=21$ length walks? If so, can it be validated using for example a community detection algorithm, the de facto approach of the field for finding communities? Moreover, it is a bit unclear how this finding relates to the finding on p.7 where it is mentioned that Caltech has a lot more communities of size at least 10 than the other three universities. This should be clarified.

The first set of experiments claims to provide(s) a new lens on prior observations of such community structure (p. 5), in particular drawing upon a few of the empirical networks and the WS models. The WS model is known to generate networks with a lot of local clustering, is that what is measured here as being community structure? In fact, in the WS experiments, a very dense network is created by initially connecting nodes to a large number of other nodes, and then rewiring with relatively low probability. To what extent is this really community structure, and to what extent is it just a slightly distorted dense network? Would a benchmark dataset known for having more realistic community structure, such as SBM or LFR, show the same results?

Response: Importantly, we are interested in a particular type of mesoscale structure in networks namely, the structure of subgraphs that are induced by a uniformly random k -path that depends both on a network and on the mesoscale k . Therefore, our findings concern networks at each mesoscale k . For a given network, one can observe different structures at different mesoscales. See, e.g., Figure 4 in the main manuscript.

The community structure that we have in mind pertains to communities that we observe in k -node subgraphs that are induced by uniformly random k -paths. We emphasize that we do not try to infer any kind of global network structure (e.g., global block structures, as in community structure) from the latent motifs. Instead, the latent motifs reflect a particular mesoscale structure of networks (that is, sub-graph structures that are induced by k -paths), and such mesoscale structures can be rather different from global block structures. In the revised manuscript, we give some examples of such k -node subgraphs from various networks to illustrate the type of mesoscale structure that we see in such subgraphs. Additionally, see key changes (Key 1) and (Key 2) in Section 1.

In the revised manuscript, we have added a new experiment to give a quantitative validation that the community sizes that we observe in latent motifs agree with our observations on actual subgraphs. In Figure 3 in the main text of the revised manuscript, we compare community sizes (from communities that we obtain from Louvain algorithm for modularity maximization) in actual 10,000 subgraphs with those in $r = 25$ latent motifs. As suggested by the reviewer, in the revised manuscript, we have also added two SBM networks with three global communities. For most networks that we consider, there is excellent agreement between the community-size statistics in actual subgraphs (with 10,000 checked in each case) and $r = 25$ latent motifs.

The first set of experiments is said to reveal nodes that are adjacent to all other nodes. The word reveal here seems a bit misplaced as these hubs are the essence of the BA model which was used to generate the networks in which this pattern was revealed. This should at least be adequately rephrased.

Response:

- (a) We have revised the sentence “However, the latent motifs for the two BA networks at scale $k = 21$ reveal nodes that are adjacent to all other nodes. Such ‘hubs’ are characteristic of BA networks, which have a heavy-tailed degree distribution [8].” to “In SNAP_FB, CALTECH, and MIT at scales $k \in \{6, 11, 21\}$ and for the BA networks at scales $k \in \{11, 21, 51\}$, the two most dominant latent motifs in Figure 3 have nodes that are adjacent to many other nodes in the latent motif. Hubs (i.e., nodes that are adjacent to many other nodes) are characteristic of both BA networks (which have heavy-tailed degree distributions) [1] and most social networks (which typically have heavy-tailed degree distributions) [8].” In the revision, we now say “latent motifs have ..” rather than “latent motifs reveal ..”.

The second set of experiments on network reconstruction aims to reconstruct an entire network. The big problem here is the way in which it is validated whether the reconstruction was successful. This is done by taking the size of the intersection divided by that of the union of the original and reconstructed edge sets. This seems very crude, and appears to measure edge set overlap, and not network similarity. More common ways of assessing network similarity in the field, such as the similarity of the degree distribution or similarity of measures such as local clustering, average distance and/or diameter, are not reported on. This makes it hard to judge if the reconstructed networks are even remotely similar to the original networks, in terms of network structure.

Response:

- (a) We understand the concern of the reviewer. We first note that the Jaccard distance, which is equal to 1 minus the Jaccard coefficient, is a proper metric (in

the mathematical sense) on the space of networks with the same set of nodes. Therefore, if the Jaccard coefficient takes the value 1 in a comparison between the original and the (truncated) reconstructed network on the same node set, then those networks have identical adjacency matrices and are thus identical networks. Therefore, network statistics (such as degree distribution, mean local clustering coefficient, and so on) of the original and the reconstructed network will be identical.

We have added new experiments that compare the degree distributions and the mean local clustering coefficients of the original and the thresholded reconstructed networks. (See Figure 6 of the main manuscript. We observe excellent agreement between the original and reconstructed networks, and we also observe interesting behavior in the degree distributions of the reconstructed networks as we increase the ‘rank’ r of the network dictionary that we use for reconstruction. We discuss these points in the fourth paragraph in the section “Network reconstruction using latent motifs” in the main manuscript. We did not add experiments with mean path lengths or diameter because they take an extremely long time to compute for large networks.

Rather diverse accuracies of reconstruction (although the way of measuring that is perhaps problematic, see the previous point) are reported on on page 6 / Figure 3. What makes that one empirical network is so easy to reconstruct, and the other is not? Can any structural properties of these networks be derived that explain this?

Response:

- (a) Thank you for this comment. We briefly discussed this point in key points (Key 3) and (Key 4) in Section 1. We include two additional detailed comments here.

First, for very sparse networks, regardless of the nature of such a network (whether it is a social network, a randomly-generated network, or something else), the subgraphs that are induced by k -paths tend to have similar structures. For example, k -node subgraphs that one samples from the sparse Erdős–Rényi network ER_1 tend to have k -node paths and a few extra off-chain edges. (See Figure 1 in the main manuscript.) A similarly sparse network, such as UCLA, has similar subgraph patterns, even though its off-chain edges are not independently random as they are in subgraphs of ER_1 . This is why we can reconstruct some networks with high accuracy by using latent motifs that we learn from a completely unrelated network. We discuss this point in the sixth paragraph (starting with “We also comment briefly about the cross-reconstruction experiments.”) of the section “Network reconstruction using latent motifs”.

Second, in the revised manuscript, we added a novel theoretical result (see Theorem G.10(iii) in our Supplementary Information) about the relationship between the global reconstruction error and an approximation error of mesoscale patches by latent motifs. This result includes an explicit dependence on the number of nodes in subgraphs at a chosen mesoscale. Informally, Theorem G.10(iii) states that one can accurately reconstruct a network if one has a dictionary of latent motifs that can accurately approximate the mesoscale patches of a network. By using a suboptimal network dictionary that consists of only a single k -path, our theoretical bound yields the following simple but insightful result:

$$\text{Jaccard reconstruction error} \leq \frac{1}{k-1} \quad \left(\begin{array}{l} \text{mean \# of off-chain edges} \\ \text{in mesoscale patches} \end{array} \right)$$

For any Erdős–Rényi network with expected edge density p , the bound in the right-hand side above equals $kp/2$ in expectation. At scale $k = 20$ for ER_2 , this value is 0.5. Consequently, we expect a reconstruction accuracy of at least 50% when reconstructing ER_2 using latent motifs that have large on-chain entries and small off-chain entries. Substituting the edge density of UCLA for p , we expect a reconstruction accuracy of at least 94%. We give a detailed discussion on this point in Figure 7 in the main manuscript. We give a proof of Theorem G.10 in Section G.3 in the SI.

The third set of experiments deals with predicting missing/spurious links, and the experimental setup is similar to that of link prediction, a well-known problem at the intersection of network analysis and machine learning. An important problem here is that the presented results in Figure 4f are subject to many many preconditions: there is intensive rebalancing of classes done to train a proper model (a common problem in this scenario due to the sparseness of the problem, and explained in the supplementary material). But moreover, a comparison is made with existing yet parameterized embedding methods. However, a detailed explanation of the exploration of the parameter space of these methods vs. the parameters of the NDL+NDR method, is not given. This makes the statement that it achieves state-of-the-art results difficult to validate.

Response:

- (a) We have thoroughly revised our network-denoising experiments in the revised manuscript. We have implemented recent network-embedding methods (specifically, DEEPWALK and NODE2VEC) and several more traditional approaches (specifically, the JACCARD INDEX, PREFERENTIAL ATTACHMENT, the ADAMIC-ADAR INDEX, and SPECTRAL EMBEDDING). (We give implementation details in Section F.9 in the Supplementary Information.)

In the revised manuscript, we compare our network-denoising method to all of these methods in three different settings: (1) we add 1,000 (except 30,000 for H. SAPIENS) false edges that we randomly generated using a Watts–Strogatz model; (2) we add 50% (of the number of original edges) of false edges that we randomly generated using an Erdős–Rényi model; (3) we remove 50% of uniformly randomly chosen edges while maintaining the fact that our network consists of one connected component. We provide full implementation details as well as hyperparameter choices in Section F.9 in the SI. See key change (Key 5) in Section 1 for more details.

For NDL and NDR, we use $k = 21$ and $r \in \{2, 25\}$. However, we agree that it is difficult to make the strong claim that our method is state-of-the-art. Therefore, we have revised our phrasing in the main text of the manuscript to instead claim that our method achieves particularly good performance for denoising structured and additive noise.

The findings are all in all not extremely surprising or groundbreaking from a substantive perspective, and reconfirm either characteristics of the model that was used to generate the datasets, or reconfirm empirical findings about the considered networks that are already known about those types of networks. The conclusion of the paper is that the generality of the insights should be established, which is a valid one. Moreover, while clearly beyond the scope of the paper, slightly more attention could

be given to the fact that in modern network science studies, the temporal, feature-rich and multi-layer aspect of a network plays a role. Existing methods for motif detection already incorporate these aspects, which could at least be reflected on to better position the work within the literature.

Response:

- (a) Thank you for suggesting potential extensions of our work to temporal networks. In the conclusion, we have added the following paragraph: “Notions of motifs have been extended to more complicated network structures, such as temporal networks (in which nodes, edges, and edge weights can change with time) [7] and multilayer networks (in which, e.g., nodes can be adjacent via multiple types of relationships) [6]. We did not consider latent motifs in these generalized network structures in our work, but it is clearly of interest to extend our approach and algorithms to these situations.”

6. SMALLER POINTS:

- (a) p.2 weighted sum of small number \rightarrow a small number

Response:

“..weighted sum of small number..”: We have changed this to “.. one can successfully approximate their k-node subgraph patterns as a weighted sum of a small number of latent motifs.”

- (b) use of \sim before `\ref{}{} and \cite{}{} (rather than a space; occurs a few times)`

Response:

Thanks for pointing this out. In the revision we globally use \sim `\ref{}{} and \sim \cite{}{}.`

- (c) “on p.4, there is a Section without a ref following it”

Response:

We have fixed this typo.

- (d) on p.3, the sentence We refer to the remaining entries as off-chain entries; they represent the additional connection that may vary depending on the network structure is a bit unclear; especially the part the additional connection that may vary depending on is rather vague; what is the magnitude of this? In its current format, this paragraph raises more questions than it answers.;

Response:

In the revised manuscript, we now state that “ A latent motif is a k-node weighted network with nodes $\{1, \dots, k\}$ and edges that have weights between 0 and 1. We use the term ‘on-chain edges’ for the edges of a latent motif between nodes i and $i + 1$ for $i \in \{1, \dots, k - 1\}$; we use the term ‘off-chain edges’ for all other edges.” Additionally, in the revision, we give subgraph examples in Figure 1 in the main manuscript. In that example, we show on-chain edges in red and off-chain edges in blue.

- (e) In the first set of experiments, the ER results are not really discussed, although their setup/parameters are mentioned. It is interesting that ER shows higher than random accuracy in the second set of experiments, whereas it is a truly random graph. This warrants some explanation.

Response:

Thank you for pointing this out. In the main manuscript, we now comment on the cross-reconstruction experiments in Figure 5 that use latent motifs that we learn from ER networks. We also have added a new theoretical result on an upper bound on cross-reconstruction error (and figures to

demonstrate them) to help address this point. See response (5.a) to one of the reviewer's main comments.

REFERENCES

- [1] Albert-László Barabási and Réka Albert. “Emergence of scaling in random networks”. *Science* 286.5439 (1999), pp. 509–512.
- [2] Daniel D. Lee and H. Sebastian Seung. “Learning the parts of objects by non-negative matrix factorization”. *Nature* 401.6755 (1999), pp. 788–791.
- [3] Ron Milo, Shai Shen-Orr, Shalev Itzkovitz, Nadav Kashtan, Dmitri Chklovskii, and Uri Alon. “Network motifs: Simple building blocks of complex networks”. *Science* 298.5594 (2002), pp. 824–827.
- [4] Bryan Perozzi, Rami Al-Rfou, and Steven Skiena. “DeepWalk: Online learning of social representations”. *Proceedings of the 20th ACM SIGKDD International Conference on Knowledge Discovery and Data Mining*. 2014, pp. 701–710.
- [5] Aditya Grover and Jure Leskovec. “NODE2VEC: Scalable feature learning for networks”. 2016, pp. 855–864.
- [6] Federico Battiston, Vincenzo Nicosia, Mario Chavez, and Vito Latora. “Multilayer motif analysis of brain networks”. *Chaos: An Interdisciplinary Journal of Nonlinear Science* 27.4 (2017), p. 047404.
- [7] Ashwin Paranjape, Austin R Benson, and Jure Leskovec. “Motifs in temporal networks”. *Proceedings of the tenth ACM international conference on web search and data mining*. 2017, pp. 601–610.
- [8] Mark E. J. Newman. *Networks*. Second. Oxford, UK: Oxford University Press, 2018.
- [9] Hanbaek Lyu, Facundo Memoli, and David Sivakoff. “Sampling random graph homomorphisms and applications to network data analysis”. *arXiv:1910.09483* (2019).
- [10] Hanbaek Lyu, Deanna Needell, and Laura Balzano. “Online matrix factorization for Markovian data and applications to network dictionary learning”. *Journal of Machine Learning Research* 21 (2020), pp. 1–49.
- [11] C. Seshadhri, Aneesh Sharma, Andrew Stolman, and Ashish Goel. “The impossibility of low-rank representations for triangle-rich complex networks”. *Proceedings of the National Academy of Sciences of the United States of America* 117.11 (2020), pp. 5631– 5637.
- [12] Julien Mairal, Francis Bach, Jean Ponce, and Guillermo Sapiro. “Online learning for matrix factorization and sparse coding”. *Journal of Machine Learning Research* 11 (2010), pp. 19–60.

REVIEWER COMMENTS

Reviewer #1 (Remarks to the Author):

In the revised manuscript “LEARNING LOW-RANK LATENT MESOSCALE STRUCTURES IN NETWORKS”, the authors did additional analysis to address our previous comments. Yet, we still have some concerns on the revised version.

1. In Figure 3, the authors concluded that the community-size statistics of the learned motifs are close approximations of the corresponding statistics for the subgraph samples of the networks. It would be nice to show the p-values to see whether it does not have significant difference between two cases. In addition, why are the median values of some networks are not seen in the box plot? Is this because the median is equal to either the lower or the upper quartile?

2. As mentioned in our previous reviewer report, Accuracy is not a good metric for imbalanced dataset. In particular, for link prediction, the number of true negative links is significantly higher, a model that predicting all links as negative can also achieve much higher accuracy.

3. We suggest the authors also show the AUPRC in the Figure 13, rather than showing the Precision and Recall at a particular threshold. In addition, the author should explain why the accuracy of Adamic-Adar index was missing in most of networks and noise type. At least for the noise type -ER, we expect the accuracy cannot be zero.

4. The authors performance additive noise as: “In our experiments with additive noise, we corrupt a network by uniformly randomly adding 50% of the number of its edges or 1,000 random edges for all but one network (we add 30,000 random edges for H. sapiens) that we generated using the WS model. We seek to classify the edges and non-edges in the resulting corrupted network as ‘negative’ (i.e., false edges) or ‘positive’ (i.e., true edges)”. Do the edges refer to the original edges in the network and the additionally introduced edges and non-edges refer to the non-existing edges in the original network? Wouldn't the application be identifying those additional links (positive) from the links already existed in original networks (negative), which we considered as a denoise scenario?

5. The comparison with the existing methods was shown in Figure 9. However, we do not see strong evidence that the proposed method outperforms existing ones.

Reviewer #2 (Remarks to the Author):

See attached

Reviewer #3 (Remarks to the Author):

Comments regarding interpretation of the latent motifs and sampling have been addressed. Most notably, the sampling method is much improved. Artefacts from backtracks no longer appear in the network dictionaries and this is now an area where the paper improves upon prior work. The aim of the MCMC procedure is also much clearer: to sample from the set of all subgraphs with a hamiltonian walk. This and the new section on size distributions greatly help with the interpretation of the latent motifs, and we compliment the authors on the improved visualizations. Also experiments were adjusted, with a more clear perturbation scheme.

The paper is now clear about what it does: decompose a network using NNMF onto a dictionary built out of a large sample of hamiltonian-walk subgraphs from that same network.

It remains unclear why this is a useful thing to do. Initially, the stated motivation was that the new method beats the state-of-the-art on particular tasks. But the downstream tasks highlighted in the paper are, in the end, not so difficult (as noted in the response to several of the reviewers). And anyways the classic approaches perform about as well, with much less conceptual and algorithmic complexity. While the framing of the paper has been appropriately adjusted, it is now missing a compelling motivation.

Perhaps our collective curiosity about new methods is motivation enough; the approach is an interesting one. But the paper would be stronger with a straightforward discussion about the promises and pitfalls of adding so much methodological complexity.

Perhaps the promise of this method, in future iterations, is so bright that the added complexity is worth it. It is a major weakness of the paper that such a discussion does not appear.

1. MAIN CHANGES

1. All reviewers pointed out that our previous network-denoising experiments did not give a strong justification of the proposed method for a downstream task. More specifically, they noted that $-ER$ and $+ER$ denoising are not very challenging even for the classical methods. To address this issue, in the newly revised manuscript (R2), we have replaced the $-ER$ and $+ER$ denoising experiments with a more challenging ‘localized noise’ setting, where we remove edges from or add edges to a selected subset of the nodes. Additionally, while addressing this point, we found an error in our code implementation in the previous revision (R1) that significantly degraded denoising performance of our method for $-ER$ case. This then yielded a lower denoising accuracy for our approach than what we had reported in the initial submission. Our new results shows that our method is competitive with the other methods for denoising additive and localized noise.
2. As suggested by Reviewer 2, we have moved Figure 4 and associated text in (R1) to the Supplementary Information.
3. Reviewer 2 pointed out that the main text needs further clarification and a self-contained description of the network-denoising algorithm. In the new revision (R2), we have added a new illustrative figure (see Figure 4) to do this.

2. ITEMIZED RESPONSES TO REVIEWER 1

Summary: In the revised manuscript LEARNING LOW-RANK LATENT MESOSCALE STRUCTURES IN NETWORKS, the authors did additional analysis to address our previous comments. Yet, we still have some concerns on the revised version.

2.1. Detailed responses.

1. In Figure 3, the authors concluded that the community-size statistics of the learned motifs are close approximations of the corresponding statistics for the subgraph samples of the networks. It would be nice to show the p-values to see whether it does not have significant difference between two cases. In addition, why are the median values of some networks are not seen in the box plot? Is this because the median is equal to either the lower or the upper quartile?

Response: As suggested by the referee, we now compute p-values (using Mood’s median test) to determine whether or not there is a statistically significant difference between the community-size samples from the subgraphs and the latent motifs. See

the revised Figure 3 in the main text. Based on these calculations, we find that there is not a statistically significant difference between the medians of the two samples, except for `SBM2` at a significance level of 0.1.

Additionally, the referee is correct in their speculated answer about the median values. Moreover, this occurs more frequently for latent motifs than for subgraphs because we compute the community-size box plots for latent motifs using 25 such motifs, whereas the box plots for subgraphs use 10,000 subgraphs.

2. As mentioned in our previous reviewer report, Accuracy is not a good metric for imbalanced dataset. In particular, for link prediction, the number of true negative links is significantly higher, a model that predicting all links as negative can also achieve much higher accuracy. We suggest the authors also show the AUPRC in the Figure 13, rather than showing the Precision and Recall at a particular threshold. In addition, the author should explain why the accuracy of Adamic-Adar index was missing in most of networks and noise type. At least for the noise type -ER, we expect the accuracy cannot be zero.

Response: As suggested, we have added AUPRC to Figures 15 and 16 (in the Supplementary Information). Below Figure 9 in our previous version (R1), we noted that “The ADAMIC-ADAR INDEX is not defined for nodes with self-edges, so we do not use it for networks with self-edges (such as `ARXIV`, `CoRoNAvIRuS`, and `H. SAPIENS`).” In the new revision (R2), we revised our experiments so that we no longer consider self-edges in those data sets. This allows us to now report the Adamic-Adar index for all data sets.

3. The authors performance additive noise as: In our experiments with additive noise, we corrupt a network by uniformly randomly adding 50% of the number of its edges or 1,000 random edges for all but one network (we add 30,000 random edges for `H. sapiens`) that we generated using the WS model. We seek to classify the edges and non-edges in the resulting corrupted network as negative (i.e., false edges) or positive (i.e., true edges). Do the edges refer to the original edges in the network and the additionally introduced edges and non-edges refer to the non-existing edges in the original network? Wouldnt the application be identifying those additional links (positive) from the links already existed in original networks (negative), which we considered as a denoise scenario?

Response: That is precisely how we perform denoising with additive noise (except that we label the added edges as “negative” and the original edges as “positive”). It seems that our prior description was not very clear and thus caused the reviewer’s confusion. To address this issue, in our new revision (R2), we now describe this setting more clearly in the main text of our manuscript. We have also revised the caption of Figure 9 to clarify this point: “We seek to classify the edges in the resulting corrupted network as true edges (i.e., original edges) and false edges (i.e., added edges).”

4. The comparison with the existing methods was shown in Figure 9. However, we do not see strong evidence that the proposed method outperforms existing ones.

Response: We have revised our denoising experiments and have made them harder. In most cases, our proposed method outperforms the existing methods. The performance gain is especially noticeable for denoising additive and localized noise.

3. ITEMIZED RESPONSES TO REVIEWER 2

Summary. We appreciate the authors thoroughness in addressing all the reviewers comments. The revised manuscript has definitely improved. The manuscript is well written, and the contributions of the manuscript are sufficient to warrant publication in *Nature Communications*. There are several issues that might impede the readers understanding of the material presented, which we would like to convey to the authors.

Response: Thank for for your positive view of our work and that you feel it warrants publication in *Nature Communications*. Thank you as well for your additional suggestions, which we have addressed to further improve our paper.

3.1. Major comments.

1. In the network construction task, symbolized by $Y \leftarrow X$, the motifs (L_1, \dots, L_r) are being extracted from X . Reconstructing Y apparently means taking k -path samples of Y , and trying to fit the r motifs to these samples, replacing the original k -paths (in a copy of Y) with the best fits to obtain an estimated Y at the end. X and Y may be the same or different graphs.

This procedure is critical for understanding the manuscript, yet in our opinion, it is being conveyed (almost) obscurely at different points in the main text. (E.g., mentioning network G and W that may or may not be driven from G , or this brief message in Page 8: “(The reconstruction also uses the original network Y .)”). We strongly recommend adding a simple, clear, and complete explanation in the main text, similar to the one provided in the last paragraph of Page 21. Such an explanation will definitely help the reader to comprehend the methodology and results better.

In short, the paper could be improved further by outlining the overall logic in a simpler fashion. From the main text its not all that clear which network the motifs are derived from, apart from a short phrase in a Figure legend. The authors write: “*In the other scenario, we suppose that G is a noisy version of some true network G_{true} and that W is faithful in the sense that it can well-approximate mesoscale patches of G_{true} at scale k .*” Apart from figure legends, this is perhaps the best indication of where W is derived from, yet it is ambiguous. Of course with $Y \leftarrow X$ the W are supposed to be derived from X , but this should be stated directly in the main text. Perhaps even worse, the authors dont seem to mention in the main text that Y is made using numerous patches sampled from Y .

Response: Thank you for this comment. We wholeheartedly agree that it is important to make this clear in the main text of our manuscript, so we are grateful for this feedback that we did not succeed in doing so in our most recent submission (R1). Accordingly, in the new revision (R2), we have added a new Figure 4 to illustrate the network-reconstruction algorithm. Additionally, as the reviewer suggested, we added a complete description of the network-reconstruction algorithm, which we previously gave in the “Methods” section (in the last paragraph of page 21 in (R1)).

In (R1), below Figure 5, we noted that “*We label each subplot of Figure 5 with $Y \leftarrow X$ to indicate that we are reconstructing network Y using a network dictionary that we learn from network X . (The reconstruction also uses the original network Y .)*”. But we agree with the reviewer that the description of the experiment $X \leftarrow Y$ could be further clarified. In (R2), below Figure 5, we now write “*we label each subplot of Figure 5 with $Y \leftarrow X$ to indicate that we are reconstructing network Y by*

approximating mesoscale patches of Y using a network dictionary that we learn from network X .”

2. We urge authors to provide the same clear definition for the network denoising task, in the main text. When denoising a given G to estimate the unknown G , are motifs derived from G ? It is very hard to tell from the main text.

Response: This is correct: We always learn the motifs for denoising tasks from a corrupted network (without knowing the true original network). In the new revision (R2), we have clarified the specification of the network-denoising tasks. At the beginning of the third paragraph with the heading “Network denoising using latent motifs” on p.11 of (R2), we now write “*For each observed network G_{Obs} , we apply NDL to learn a network dictionary W_{Obs} and then use NDR to reconstruct a network G_{Recons} by approximating mesoscale patches of G_{Obs} using latent motifs in W_{Obs} . (We compute G_{Recons} without using any information on G_{True} .)*”

3. Please comment on the efficacy of the method for the purpose of network comparison which is being advocated as a potential application for NDL and NDR. In Fig. 5e, it can be seen that the UCLA network can be reconstructed with more accuracy by using the Harvard network than by using the UCLA network itself. How could this be, unless error fluctuations are the cause for this discrepancy?

Response: In Figure 5 in the new revision (R2), we have changed these experiments so that the results are now means over 5 trials. The revised results show that UCLA is best reconstructed by using latent motifs that are learned from UCLA itself.

Also, motifs from widely different network structures lead to a rather similar (and high) network reconstruction accuracy. Does this not suggest that the application for network comparison might be problematical given it may be difficult to discern different networks.

Response: Thank you for raising this important point. This point can be understood by noting that subgraphs from different networks that are induced by k -paths can be similar if the networks are very sparse. Consequently, the latent motifs that are learned from such networks will be similar and may yield high cross-reconstruction accuracies. This indicates that such networks are likely to have similar mesoscale structures that are based on k -path-induced subgraphs. Therefore, to better distinguish between different sparse networks, it may be desirable to use a scale k that is large enough so that k -node mesoscale patches have many off-chain edges. Naturally, doing so comes with added computational cost. We have added a relevant discussion to the Conclusions section of the main text.

3.2. minor comments.

1. We did not find Figure 3 (and the associated explanations) of the revised manuscript essential to the message being delivered by the main text. Authors may consider moving it to the SI.

Response: We believe that the reviewer meant Figure 4 in (R1), which shows the two most-dominant latent motifs for all networks at various scales ($k = 6$, $k = 11$, $k = 21$, and $k = 51$). As suggested, we have moved Figure 4 and the associated explanations to the Supplementary Information. This allowed us to incorporate the additional figures and discussions that the other referees suggested for the main text.

2. Authors have replaced the k -walk sampling method to obtain k -paths from observed networks. In many places, it seems that the k -walk terminology is still being used by mistake instead of k -path.

Response: We clarify that, despite our use of k -path sampling instead of k -walk sampling for the NDL algorithm in our last manuscript revision (R1), there are still instances of using k -walk sampling for NDR algorithm. Namely, in the statement of the NDR algorithm (in Appendix F of the Supplementary Information), the option $\text{InjHom} = \text{T}$ indicates that only k -paths (injective homomorphisms) are used for the network reconstruction, whereas $\text{InjHom} = \text{F}$ indicates that all sampled k -walks are used for the network reconstruction. Using all sampled k -walks for reconstruction has a computational advantage over only using k -paths. Accordingly, we used the option $\text{InjHom} = \text{T}$ throughout the network-reconstruction experiments in (R2) (as well as in (R1)).

Despite the computational gain in using the option $\text{InjHom} = \text{T}$, there is a potential theoretical issue. Namely, because we use network dictionary W that we learn from mesoscale patches on k -paths for the reconstruction, our reconstruction bound (on the right-hand side of equation (33) in the Supplementary Information in (R2)) is better optimized when we use only k -paths for network reconstruction. Nevertheless, we did not find any significant difference in the results of reconstruction experiments for the two options $\text{InjHom} \in \{\text{T}, \text{F}\}$. We elaborate on this point in the third and the fourth paragraph in Section 4.1 of the Supplementary Information in the new revision (R2).

In the main text in (R2), for clarity, we state the network-reconstruction algorithm for the version that uses only k -paths. Accordingly, we have changed all instances of k -walks to k -paths in the main text in sections for network reconstruction and denoising.

3. In several instances, there is inconsistency between the text and presented results regarding the number of networks used (15 or 16 graphs).

Response: Thank you for pointing this out. We have checked this and fixed all instances.

4. In figure 10 legend the panels are mixed up in several places. The image in panel d is a reconstruction of the image in panel a is incorrect. Panel c is not mentioned. Please check the text referring to these panels as well.

Response: Thank you. We have fixed this error.

5. p.5 carries over →carry over

Response: We have fixed this.

4. ITEMIZED RESPONSES TO REVIEWER 3

1. Comments regarding interpretation of the latent motifs and sampling have been addressed. Most notably, the sampling method is much improved. Artefacts from backtracks no longer appear in the network dictionaries and this is now an area where the paper improves upon prior work. The aim of the MCMC procedure is also much clearer: to sample from the set of all subgraphs with a hamiltonian walk. This and

the new section on size distributions greatly help with the interpretation of the latent motifs, and we compliment the authors on the improved visualizations. Also experiments were adjusted, with a more clear perturbation scheme.

Response: Thank you for your positive feedback on our revision.

2. The paper is now clear about what it does: decompose a network using NNMF onto a dictionary built out of a large sample of hamiltonian-walk subgraphs from that same network. It remains unclear why this is a useful thing to do. Initially, the stated motivation was that the new method beats the state-of-the-art on particular tasks. But the downstream tasks highlighted in the paper are, in the end, not so difficult (as noted in the response to several of the reviewers). And anyways the classic approaches perform about as well, with much less conceptual and algorithmic complexity. While the framing of the paper has been appropriately adjusted, it is now missing a compelling motivation.

Response: As the reviewer pointed out, $-ER$ and $+ER$ denoising are not very challenging tasks even for the classical methods. To address this issue, in the newly revised manuscript (R2), we have replaced the $-ER$ and $+ER$ denoising experiments with a more challenging ‘localized noise’ setting, where we remove edges from or add edges to a selected subset of the nodes. In both of these new experiments in (R2), our method significantly outperforms all benchmark methods.

3. Perhaps our collective curiosity about new methods is motivation enough; the approach is an interesting one. But the paper would be stronger with a straightforward discussion about the promises and pitfalls of adding so much methodological complexity. Perhaps the promise of this method, in future iterations, is so bright that the added complexity is worth it. It is a major weakness of the paper that such a discussion does not appear.

Response: We have extended our Conclusions section to summarize our contribution, the motivation behind, when our method works well, when it does not, and what should be done in the future. We certainly agree that such a discussion should appear in our paper, and now it does in the new revision.

REVIEWER COMMENTS

Reviewer #1 (Remarks to the Author):

In the revised manuscript, the authors did additional analyses to address our previous comments. However, we still have some concerns about the revised version.

1. The authors calculated the Precision-Recall curve, as well as the NPV-specificity curve in the revised version. However, the performance of the proposed method NDL+NDR in the scenario of -ER (randomly removing half of existing links, which is more realistic than +WS and +ER) is even lower than the baseline method Jaccard index and Adamic-Adar index (see Figs.15, 16). The authors should explain the poor performance of their method in those cases and discuss the limitations of their method.

2. As AUPRC is a metric to quantify the ability of a prediction method to identify positive instances from all unobserved instances, it should be much lower in link prediction due to the number of unobserved links being significantly larger than existing links. Therefore, identifying positive links from numerous unobserved links is quite challenging. However, the AUPRCs of many methods in multiple networks in Figure 15 are very close to 1, which is unexpected. The authors should double-check if they have inadvertently flipped the labels of positive and negative links. More importantly, the authors should make all their source code/data available so that readers can reproduce their results.

Reviewer #2 (Remarks to the Author):

see attached file

Reviewer #3 (Remarks to the Author):

The first version of this paper had what appeared to be larger conceptual errors, and the second version had a minor coding error. These errors have now been fixed, and the authors have improved the description and explanation of much of the methodology. Figure 3 is a useful addition. At the same time, the motivation of the paper remains rather poor, so there is a need for further clarification in that regard.

The experiments have been updated to better suit the methodology presented. In the narrow sense of these examples, the paper can claim improvement over prior work. However, performance on contrived examples is enough to say only that the method has *potential* usefulness on important tasks. We cannot know if and when real noise is 'localised' in a way that suits this methodology. To say that the *potential* usefulness implies *actual* usefulness requires addressing lingering concerns around interpretability and applicability.

For example, further methodological clarification regarding on-chain and off-chain links is needed to improve interpretability. Some of the findings hinge on parameter ξ , but it is not clear how the removal of on-chain links from the latent motifs works its way through NDR. Without a clear explanation there is no principled way to set ξ in scenarios without ground truth. The mapping back from X to A in the reconstruction must be clarified---here are two ideas for how to go about it:

A) Provide an explanation of NDR using a dictionary of size 1, with only on-chain edges. This is alluded to as a limiting case in the discussion of Theorem 1 on page 10. Explaining the reconstructed network that on-chain edges create should help clarify when it is important to use only off-chain edges.

B) Walks of length 1 starting from a random node is a random sample of the edges of a graph, and the corresponding dictionary is a single on-chain edge. This is an even simpler limiting case that should help clarify the value-added of using k -paths that allow for learning off-chain edges.

Overall, I remain convinced that, should it be published in Nature Communications, the paper needs a discussion about the pitfalls of using a really really heavy hammer when simpler tools are available. For example, it is more difficult to find and correct errors when using complicated methods. This hurts the applicability of this method as compared to simpler methods. If the argument is that this method is better, not just new or interesting, then the specialised expertise that it takes to apply it correctly cannot be ignored.

Summary: In the revised manuscript, the authors did additional analyses to address our previous comments. However, we still have some concerns about the revised version.

1.1. Detailed responses.

1. The authors calculated the Precision-Recall curve, as well as the NPV-specificity curve in the revised version. However, the performance of the proposed method NDL+NDR in the scenario of -ER (randomly removing half of existing links, which is more realistic than +WS and +ER) is even lower than the baseline method Jaccard index and Adamic-Adar index (see Figs.15, 16). The authors should explain the poor performance of their method in those cases and discuss the limitations of their method.

Response: Thank you for the comment. We have added relevant comments to the manuscript about the method’s performance in these cases. We have also added a discussion of the limitations of our method. Specifically, we have added the following text to the manuscript:

In the Limitations and further discussion: Fourth, although our method for network denoising is competitive especially for the anomalous-subgraph-detection problem it does not always outperform all existing methods, and some of those methods are much simpler than ours. For instance, for edge-prediction tasks, it seems that our method is often more conservative than the other examined methods at detecting unobserved edges (See Figures 16, and 17 in the SI). Therefore, we recommend using our method in conjunction with existing methods for such tasks.

In Appendix E.15 in the SI for Fig. 16: For denoising +WS and +ER noise, our approach yields larger AUCs for the precision–recall curves than all of the other examined network-denoising methods. Our approach performs competitively for denoising –ER noise, except for the network H. SAPIENS.

In Sec. E.14 in the SI for Fig. 17: In Figure 17, we show the dependence of the NPV and specificity scores of the network-denoising experiments in Figure 8 on the threshold θ . Our approach yields larger AUCs for the NPV–specificity curves than all of the other examined methods for denoising +WS and +ER noise, except for +ER for the network CALTECH. For denoising –ER noise, our approach does not seem to be particularly effective at detecting unobserved edges. It is outperformed by other methods for SNAP FB (by all of them except PREFERENTIAL ATTACHMENT), ARXIV (by all of them except PREFERENTIAL ATTACHMENT and

SPECTRAL EMBEDDING), and H. SAPIENS (by all of them except PREFERENTIAL ATTACHMENT and SPECTRAL EMBEDDING).

Additionally, we have added a new Figure 1 to discuss the motivating application of anomalous-subgraph detection and how to apply our method to study this problem. More specifically, anomalous-subgraph detection seeks to algorithmically detect a subset of the nodes of a network whose connection patterns are anomalous with respect to the rest of the network. This anomalous-subgraph-detection problem is identical to the network-denoising problem with additive noise in Figure 8 of the main text.

We consider the following simple conceptual framework for anomalous-subgraph detection:

- We learn “normal subgraph patterns” in an observed network and then seek to detect subgraphs in the observed network that deviate significantly from these patterns.

By studying low-rank mesoscale structures in networks, we can turn this high-level idea for anomalous-subgraph detection into a concrete approach, which we illustrate in the new Figure 1.

2. As AUPRC is a metric to quantify the ability of a prediction method to identify positive instances from all unobserved instances, it should be much lower in link prediction due to the number of unobserved links being significantly larger than existing links. Therefore, identifying positive links from numerous unobserved links is quite challenging. However, the AUPRCs of many methods in multiple networks in Figure 15 are very close to 1, which is unexpected. The authors should double-check if they have inadvertently flipped the labels of positive and negative links. More importantly, the authors should make all their source code/data available so that readers can reproduce their results.

Response: Thank you for the comment. The reviewer is correct in that identifying positive instances of unobserved edges among all nonedges is a challenging task because of the very large number of negative instances. However, the reviewer might have missed that in our experimental setting for -ER denoising, we choose a random subset of negative instances with a matching size to the number of positive instances prior to the classification task. We clarified this point further in the revision at the end of the first paragraph in the section “Network denoising experiments” with the following text:

“There are many more positives than negatives because G_{true} is sparse (i.e., the edge density is low), so we restrict the classification task to a subset E_{nonedge} of E_{obs} that includes all negatives and an equal number of positives. We will discuss shortly how we choose E_{nonedge} .”

Code for all our experiment is already public in our GitHub repository, as stated in “Data availability” section in Methods. This URL is also provided in a footnote on page 1 of the paper. (We also note that this was already the case, including the explicit hyperlink to the code in the footnote on page 1, in our prior submission.)

2. ITEMIZED RESPONSES TO REVIEWER 2

Summary. The paper presents an innovative application of Machine Learning techniques to Network Science. In particular, the efforts to use network motifs to reconstruct unknown

or corrupted networks, as based on the analogue to existing image reconstruction techniques, makes this work attractive. The authors responses to the reviewers comments are adequate.

We also hope that this new approach is worth the added complexity it entails relative to existing methods, as other reviewers seem to agree as well. Overall, given the interesting idea which is novel, the immense technical efforts on the authors side, and the prospect of opening new avenues in network science, publication of the paper in Nature Communications should be worthwhile.

We encourage authors to work more on clarifying the method and experiments, discuss/justify the worth of methodological and computational complexity of the proposed processes, and better interpret/discuss the results (especially those that appear counter intuitive).

2.1. Detailed responses.

1. It would be very helpful if authors use a simple template explanation of the process, and consistently describe all experiments using that simple template. We believe that the proposed process (the core idea and excluding the mathematics) can be simply put as something more or less like: Given a network G and motifs W from G . We can sample a large number of mesoscale patches A_i from G , and for each A_i , independently, find the best nonnegative linear combination of W giving the closest approximation \hat{A}_i . Comparing A_i s and their corresponding \hat{A}_i s demonstrates how well the given motifs W (from G) represent the network G . This is carefully framed but as late as p.7, and gets lost in the wash.

Response: Thank you for the comment. In the revised manuscript, we revised Figure 3 in the previous version as a new Figure 3. The new figure demonstrates the key idea of approximating sampled subgraphs by latent motifs, as the reviewer suggests. Readers can also conveniently refer to the pipeline in Figure 3 throughout the paper. We have added several sentences to the last paragraph of the ‘‘Motivating Applications’’ section about Figure 3 to illustrate the key idea of our paper.

2. It is possible to provide a better interpretation of the results, especially when we observe that $X \pm Y$ is still more accurate than $X \pm X$ ($X \neq Y$) in a number of your analyses for many r values, as was mentioned in our last review. We dont see how your new analysis of the mean changes this. Why are the results so good? Is it due to similarity of X and Y motifs and a bit of luck (freedom given by nonnegative linear combination of W)? Can this be eliminated by increasing the number of realizations in the experiments? Further explanation is needed.

Response: Thank you for the comment. In our current reconstruction experiment (see Figure 5), the only two instances in which $X \pm Y$ for $Y \neq X$ gives a more accurate reconstruction of X than $X \pm X$ is in panel (d), where (X, Y, r) is either (HARVARD, MIT, 25) or (HARVARD, CALTECH, 25). In these cases, the cross-reconstruction accuracies $X \pm Y$ are only slightly more accurate (by at most 2%) than the self-reconstruction accuracy $X \pm X$. Nevertheless, we agree with the reviewer that we should discuss these instances to provide a better interpretation of our results, as it is natural for readers to expect self-reconstruction to be more accurate than any cross-reconstruction.

Although averaging over multiple reconstruction experiments (as we did in the previous revision) can remove statistical errors from the randomness of the Markov chains that we use in the experiment, we missed addressing an important point in our previous revision. Specifically, *the latent motifs are learned so that they maximize the*

accuracy of reconstructing mesoscale patches (i.e., k -node subgraphs), rather than of the entire network. Therefore, although it is natural to expect the self-reconstruction of mesoscale patches to be more accurate than the cross-reconstruction of mesoscale patches indeed, our experiments in Figure 7b–e confirm this intuition it is not entirely clear whether or not this intuition automatically holds for network reconstruction. This is because network reconstruction involves averaging over multiple reconstructions of the same edge from several mesoscale patches that contain that edge. Nevertheless, we are able to provide a theoretical guarantee for network reconstruction (see Figure 7a) to assure that an accurate mesoscale patch reconstruction also yields an accurate network reconstruction.

In the revised manuscript (above Figure 7), we have added the following paragraph to explain this:

“One learns latent motifs by maximizing the accuracy of reconstructions of mesoscale patches using them, rather than by maximizing the network-reconstruction accuracy. In Figure 7b–e, we illustrate that self-reconstruction of mesoscale patches is more accurate than cross-reconstructions of mesoscale patches. However, because network reconstruction involves taking the mean of the reconstructed weights of an edge from multiple mesoscale patches that include that edge, an accurate reconstruction of mesoscale patches need not always entail accurate network reconstruction. In Figure 5, we see that the self-reconstruction $X \leftarrow X$ is more accurate than the cross-reconstructions $X \leftarrow Y$ for $Y \neq X$ for almost all choices of networks X and Y and the parameter r . The two exceptions are $(X, Y, r) = (\text{HARVARD}, \text{MIT}, 25)$ and $(X, Y, r) = (\text{HARVARD}, \text{CALTECH}, 25)$, although the cross-reconstruction accuracies in these cases are at most 2% larger than the self-reconstruction accuracy.”

Additionally, we have added new concluding comments on the interpretability of our method at the end of the main text.

3. Motifs (W_{obs}) are extracted from the intensely noisy network G_{obs} (50% noise). Presumably these high levels of noise can distort or destroy some motifs in a way that they are no longer good representatives of G_{true} . It would be great to add some explanations why things are working at such high noise levels in the main text and allow for better interpretation of results showing high performance of the algorithm. That is, it wouldnt hurt to emphasise more the underlying idea of the low-rank approximations of the original matrix being robust to noise. When you reconstruct an observed network, you are hoping to reconstruct the low-rank motifs of the observed network. The hope is that they will also correspond to the low-rank motifs of the original matrix. Even though you may consider it trivial, simple explanations explaining the key idea could make life easier for the reader in this difficult paper.

Response: Thank you for the comment. We indeed have a similar remark at the end of the main text when we discuss two confounding factors for the high denoising accuracy. In the revised manuscript, we have added this rationale just below the new Figure 1 for why our method for network denoising using low-rank mesoscale structures in networks should work well. We include it in our new discussion in which we motivate our approach using the anomalous-subgraph-detection problem. We have also discussed this point once more at the end of the section “Network-denoising experiments”.

3.

ITEMIZED RESPONSES TO REVIEWER 3

Summary. The first version of this paper had what appeared to be larger conceptual errors, and the second version had a minor coding error. These errors have now been fixed, and the authors have improved the description and explanation of much of the methodology. Figure 3 is a useful addition. At the same time, the motivation of the paper remains rather poor, so there is a need for further clarification in that regard.

The experiments have been updated to better suit the methodology presented. In the narrow sense of these examples, the paper can claim improvement over prior work. However, performance on contrived examples is enough to say only that the method has *potential* usefulness on important tasks. We cannot know if and when real noise is 'localised' in a way that suits this methodology. To say that the *potential* usefulness implies *actual* usefulness requires addressing lingering concerns around interpretability and applicability.

Response: Thank you for this comment. We have made further revisions to the manuscript to address this remaining concern. More specifically, we now motivate our study of low-rank mesoscale structures in networks by the practically important application of anomalous-subgraph detection and we provide a new experiment where our method can successfully detect anomalous edges that are not necessary localized (see Figure 1). We believe that this discussion strengthens the motivation and potential usefulness of our work. We also give more thorough discussions of the interpretability and limitations of our work. This new text will help readers without special expertise to use our method on their domain problems and to interpret the results of doing so in a scientifically correct manner.

We state our changes below in our responses to Reviewer 3's specific points that elaborate on the above concern.

3.1. Detailed responses.

1. Figure 3 is a useful addition. At the same time, the motivation of the paper remains rather poor, so there is a need for further clarification in that regard.

Response: Thank you for the comment. In the revised manuscript, we motivate our study of low-rank mesoscale structure of networks by using the example application of anomalous-subgraph detection. (See the example in the new Figure 1.) The problem is to detect an anomalous subgraph from an observed network. This problem is equivalent to the 'network denoising with additive noise' that we present later in the paper. This problem naturally motivates the learning of latent motifs as 'normative subgraph patterns' and then to reconstruct the observed network by using the latent motifs to determine which edges deviate from the normative patterns.

2. For example, further methodological clarification regarding on-chain and off-chain links is needed to improve interpretability. Some of the findings hinge on parameter ξ , but it is not clear how the removal of on-chain links from the latent motifs works its way through NDR. Without a clear explanation there is no principled way to set ξ in scenarios without ground truth.

Response: Thank you for the comment. We use the parameter $\in [0, 1]$ in the main text only for the illustrative purpose of the key idea of thinning on-chain edges of latent motifs. In all of our network-denoising experiments, we use only the two extreme values $= 0$ and $= 1$. Our NDR algorithm does not have an option to tune and only implements these extreme cases, with $= 0$ corresponding to **Denoising = T** and $= 1$ corresponding to **Denoising = F**. After the reviewers' comment, we decided that the detailed discussion of on-chain thinning and the additional experiment in

Figure 8 with the CALTECH data set are mostly technical details that are not necessary for readers to understand the main text. Therefore, in the new revision, we have moved these items into the “Methods” section. Figure 8 in the previous manuscript is now Figure 11 in “Methods”. In the main text (above Figure 8 in the revised manuscript), we have added the following explanation of the application of NDR for network denoising:

“For each observed network G_{obs} , we apply NDL at scale $k = 21$ with $r \in \{2, 25\}$ to learn a network dictionary W_{obs} . We create another network dictionary W_{obs} by removing the on-chain edges from all latent motifs of W_{obs} . (See the Methods for more discussion.) This gives total four network dictionaries, corresponding to the two values of r and whether we keep the on-chain edges of the latent motifs. With each of the network dictionaries, use NDR to reconstruct a network G_{recons} .”

The mapping back from X to A in the reconstruction must be clarified—here are two ideas for how to go about it:

A) Provide an explanation of NDR using a dictionary of size 1, with only on-chain edges. This is alluded to as a limiting case in the discussion of Theorem 1 on page 10. Explaining the reconstructed network that on-chain edges create should help clarify when it is important to use only off-chain edges.

B) Walks of length 1 starting from a random node is a random sample of the edges of a graph, and the corresponding dictionary is a single on-chain edge. This is an even simpler limiting case that should help clarify the value-added of using k -paths that allow for learning off-chain edges.

Response: Thank you for the suggestions. In the revised manuscript, we have revised Figure 3 from the previous version as the new Figure 4. In this figure, we illustrate the network reconstruction algorithm more clearly for the following two scenarios: (1) one in which we use four latent motifs with both on-chain and off-chain edges; and (2) one in which we use a single latent motif, which specifically is a k -path and hence only has on-chain edges. These illustrations help clarify the need to use more than one latent motif with off-chain edges to obtain more accurate network reconstruction. In the text, we also added an explanation of how network reconstruction with a single 2-path trivially returns the original network. The new Figure 4 also more clearly demonstrate how the sampled k -path $x = (x_1, \dots, x_k)$ maps to the adjacency matrix A_x of the subgraph that is induced by the nodes of x (i.e., $A_x(i, j) = 1$ if x_i and x_j are adjacent in the network and $A_x(i, j) = 0$ otherwise).

3. Overall, I remain convinced that, should it be published in Nature Communications, the paper needs a discussion about the pitfalls of using a really really heavy hammer when simpler tools are available. For example, it is more difficult to find and correct errors when using complicated methods. This hurts the applicability of this method as compared to simpler methods. If the argument is that this method is better, not just new or interesting, then the specialised expertise that it takes to apply it correctly cannot be ignored.

Response: Thank you for the comment. We believe the revised manuscript (including, e.g., with the new guiding Figure, as suggested by other reviewers) gives a much clearer exposition of our method and motivation. We also believe that our method is an exciting new approach in network science, and we have tried our best to demonstrate the key idea, motivation, significance, and usefulness in applications such as network comparison and network denoising. Additionally, to the referee’s summarizing point (in the locations that we indicated in our responses to comments from

the other referees), we have added additional emphasis of the method's limitations and pitfalls.

REVIEWERS' COMMENTS

Reviewer #1 (Remarks to the Author):

The authors have addressed our previous comments. We have no further concerns.

Reviewer #2 (Remarks to the Author):

The authors have appropriately revised the manuscript in response to our comments. The revised introduction works well. Fig. 1&3 of the latest version, present the ideas and processes in a simple and clear fashion as requested. Additional interpretations, explanation of non-trivial results, and discussion of limitations in this round of revision has made the manuscript ready for publication, in our opinion.

We appreciate the authors' explanation regarding cross-reconstruction sometimes beating self-reconstruction of networks in accuracy. However, the authors explained in their response that this only occurs in the reconstruction of Harvard from MIT/Caltech networks (Fig. 5d $r=25$). In fact, the Harvard self-reconstruction is clearly doing better here, as seen if you zoom in on the results. hence the authors seem to have unknowingly fixed the problem here.

Second, we would like to point out that we could not figure out the exact cause of subtle changes in the accuracy of these recent reconstructions. For example, in the previous version UCLA<-UCLA was more accurate than UCLA<-Harvard (Fig. 5e $r=9,16$) but now it is not, and this is the case for some other data points although most of the data points seem to be the same as previous versions. In any event, all is in order now. We are just pointing these things out to the authors for their own interest.

1. ITEMIZED RESPONSES TO REVIEWER 1

Reviewer 1: The authors have addressed our previous comments. We have no further concerns.

Response: Thank you for your positive feedback. We appreciate your insightful comments on the earlier versions of our manuscript.

2. ITEMIZED RESPONSES TO REVIEWER 2

Reviewer 2: The authors have appropriately revised the manuscript in response to our comments. The revised introduction works well. Fig. 1&3 of the latest version, present the ideas and processes in a simple and clear fashion as requested. Additional interpretations, explanation of non-trivial results, and discussion of limitations in this round of revision has made the manuscript ready for publication, in our opinion.

We appreciate the authors explanation regarding cross-reconstruction sometimes beating self-reconstruction of networks in accuracy. However, the authors explained in their response that this only occurs in the reconstruction of Harvard from MIT/Caltech networks (Fig. 5d $r=25$). In fact, the Harvard self-reconstruction is clearly doing better here, as seen if you zoom in on the results. hence the authors seem to have unknowingly fixed the problem here.

Second, we would like to point out that we could not figure out the exact cause of subtle changes in the accuracy of these recent reconstructions. For example, in the previous version UCLA<-UCLA was more accurate than UCLA<-Harvard (Fig. 5e $r=9,16$) but now it is not, and this is the case for some other data points although most of the data points seem to be the same as previous versions. In any event, all is in order now. We are just pointing these things out to the authors for their own interest.

Response: Thank you for your positive feedback. We appreciate your additional comments, as well as your previous insightful comments on the earlier versions of our manuscript.

For the first point, we were incorrect when we referred to the two exceptions in which self-reconstruction is less accurate than cross-reconstruction. The two instances are $(X, Y, r) = (\text{MIT}, \text{HARVARD}, 25)$ and $(X, Y, r) = (\text{MIT}, \text{UCLA}, 25)$. In the previous version of the manuscript (Revision 3), we stated incorrectly that these exceptions are $(\text{HARVARD}, \text{MIT}, 25)$ and $(\text{HARVARD}, \text{CALTECH}, 25)$. We have fixed our statements in the main text.

For the second point, we confirm that, between Revision 2 (i.e., R2) and Revision 3 (i.e., R3), we made only cosmetic changes to Figure 5 for the reconstruction experiments and that all of the numerical values and Figure captions are the same. For convenience (including for

any of our paper's readers¹, in Figure 1 of this letter, we include a side-by-side comparison of these plot from the past three revisions.

FIGURE 1. A comparison between Figures 5c–e in revisions R1, R2, and R3/R4 of our manuscript.

¹Hello readers! We hope that you are enjoying our paper.